# Privacy Breach Detection by Non-Parametric Two-Sample Tests

## Abstract

With the proliferation of machine learning services, the risk of privacy breaches has never been higher, owing to the need for collecting – sometimes by any means necessary – valuable, yet sensitive training data. When an unsanctioned data access occurs, it may become apparent after the fact, in the predictive models that have been trained on compromised data. This calls for effective membership inference methods, enabling an evaluator to identify privacy breaches. Distinct from traditional *membership inference attacks* (MIAs), which focus on determining whether individual data records were used in training, this study centers on the evaluation of sets of records, particularly when only a small proportion of the set are training members. In this scenario, traditional MIAs often suffer from non-ideal evaluation reliability. To address this issue, from a privacy evaluator's perspective, we propose a novel approach for membership inference, applicable not to individual records but to sets thereof. It relies on a non-parametric two-sample test, which leverages the differences between high-level representation to infer membership. Based on extensive experiments, our proposed *High-level Representation-based MMD* (HR-MMD) test exhibits high sensitivity in distinguishing between the training and non-training sets, with ideal type I error, making it a powerful membership detection tool. Our study offers insights into an alternative privacy breach detection scenario and opens up a promising avenue for privacy evaluation based on membership inference tests.

## 1 Introduction

Machine learning (ML) models have become indispensable in critical domains such as healthcare, genomics, and image recognition, playing a pivotal role in tasks like disease diagnosis, genetic pattern analysis, and image classification (Xiong et al., 2015; He et al., 2016; Miotto et al., 2018). These models, particularly deep neural networks, excel at extracting complex patterns and correlations from diverse datasets, enabling them to make accurate predictions (Song et al., 2017; Carlini et al., 2019; Zhang et al., 2021). However, this proficiency comes with a potential drawback – the models can inadvertently memorize training data, introducing a potential vulnerability to various privacy attacks.

As a vulnerability, this propensity of models to memorize data facets has been explored extensively, with studies revealing that attackers can exploit model memorization through several methodologies, including model extraction, attribute inference, property inference, and, crucially, *membership inference* (whether a given data record may have been used to train a model or not) (Fredrikson et al., 2015; Tramèr et al., 2016; Shokri et al., 2017b; Ganju et al., 2018). The last one, being the most related to the focus of our discussion, stands out for specifically targeting the confidentiality of training data points (Shokri et al., 2017b; Yeom et al., 2018b; Salem et al., 2019b; Chen et al., 2021).

But what is mostly regarded as a vulnerability in this context can also be exploited as an opportunity for privacy evaluators to detect unsanctioned data accesses *in hindsight*, from predictive models that have been trained on compromised data. For example, a clinical trial for a common disease may be conducted across multiple hospitals, with each hospital recruiting its patients from a large population, following a uniform recruitment strategy. Although clinical records are highly sensitive, different hospitals may be more or less able to protect them. Some data may be exposed to data theft, and may then make its way into training datasets for third-party predictive models (for that disease). By

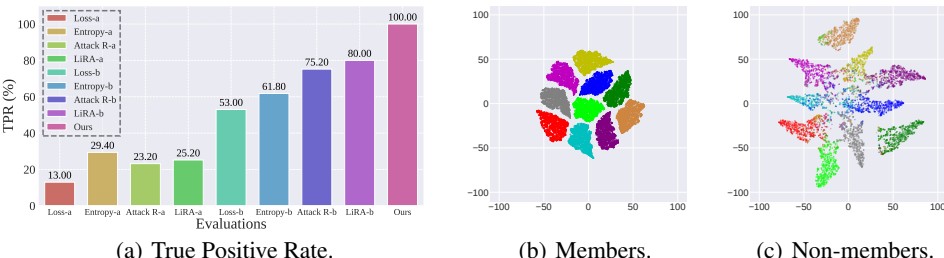

(a) True Positive Rate.    (b) Members.    (c) Non-members.

Figure 1: Motivation figures. Subfigure (a) compares the true positive rates between prior MIA methods and our proposed HR-MMD test. We evaluate a scenario where the testing subjects consist of 10% training members and 90% non-training members, aiming for an ideal true positive rate of 100%. Thresholds for prior methods were set at different false positive rates (1% for group a, e.g., Loss-a; 10% for group b, e.g., Loss-b). The false positive rate of our proposed HR-MMD test is 0%. The results demonstrate that our method provides a clearer distinction between cases where $S_Y$ contains members and those where it does not. Note that the experimental settings in Subfigure (a), detailed in Appendix G.7, differ from those used in the main experiments comparing with standard statistical baselines. Subfigures (b) and (c) use t-SNE (Maaten & Hinton, 2008) to visualize high-level representations in ResNet-18, where different colors represent different semantics (i.e., classes in *CIFAR-10*).

identifying whether a particular hospital's records have been used to train such models, a privacy evaluator could infer that those records were exposed to attackers.

The breach of privacy is a general and profound concern in the current machine learning landscape, prompting urgent discussions around user data protection. Recognizing this, legislative bodies have enacted robust privacy protection laws such as the *general data protection regulation* (GDPR) in the European Union, the *california consumer privacy act* (CCPA) in California, and the *personal information protection and electronic documents act* (PIPEDA) in Canada, which have legally consolidated the privacy protection. In particular, these laws provide individuals with the legal framework to protect their *membership privacy*, recognizing the risk of breaches through membership inference from ML models, calling for privacy-aware learning approaches. However, when unsanctioned data accesses do occur, and when compromised data does make its way, later on, into training datasets for third-party predictive models, membership inference can also be an effective tool for privacy auditors / evaluators, allowing them to detect data breaches from such models.

There are however essential differences between the well-studied membership inference problem, viewed from an attacker's perspective, and a membership inference problem reframed from a privacy evaluator's perspective, in terms of objectives and data granularity, as discussed next.

Membership inference attacks (MIAs) are commonly used to discern whether a given data record is used to train a target model (member) or not (non-member) (Hayes et al., 2017; Pyrgelis et al., 2017; Nasr et al., 2018; Rahman et al., 2018; Salem et al., 2018; Yeom et al., 2018b; Truex et al., 2018; Jia et al., 2019; Sablayrolles et al., 2019; Song & Marn, 2020; Leino & Fredrikson, 2020b; Rahimian et al., 2020; Song & Mittal, 2021; Choquette-Choo et al., 2021; Li et al., 2021; Liu et al., 2022; Ye et al., 2022; Carlini et al., 2022). However, by their positioning and objectives, traditional MIAs often suffer from high false positive rate when aiming for high true positive rate, even with the currently considered state-of-the-art *Likelihood Ratio Attack* (LiRA) (Carlini et al., 2022). This drawback persists in scenarios where the subject being tested are sets of records (Maini et al., 2021; Kandpal et al., 2023; Maini et al., 2024), yet only a small proportion are training members (as seen in Figure 1 (a)). This renders them unreliable as a tool serving a privacy evaluator – i.e., using them directly to evaluate third-party models that are potentially trained on compromised (leaked) data. An evaluation's reliability when using MIAs approaches directly would therefore not be sufficient in practice, as it would return a prediction with a high false positive rate or low true positive rate. Such an evaluation may not provide enough evidence to impose penalties, and the third-party owning the model could argue that their samples were *mis-classified* by the evaluation.

To address the issue of non-ideal evaluation reliability, we turn to the domain of two-sample tests and we adopt a membership test working at a coarser granularity, for sets of records. Two-sample tests are hypothesis tests that determine if two sets of samples come from the same distribution. Traditional methods such as $t$-tests and Kolmogorov-Smirnov tests are effective only on data in extremely low-dimensional spaces and require strong parametric assumptions about the distributions being

studied. Recent work in statistics and machine learning has focused on relaxing these assumptions, resulting in methods either generally applicable or specific to various complex domains (Gretton et al., 2012a; Székely & Rizzo, 2013; Heller & Heller, 2016; Jitkrittum et al., 2016; Chen & Friedman, 2017; Ghoshdastidar et al., 2017; Ramdas et al., 2017; Li & Wang, 2018; Gao et al., 2018).

A popular class of non-parametric two-sample tests is based on *kernel methods* (Smola & Schölkopf, 2001). Such tests construct a *kernel mean embedding* (Berlinet & Thomas-Agnan, 2004; Muandet et al., 2017) for each distribution, and measure the difference in these embeddings. For any *characteristic* kernel, two distributions are the same if and only if their mean embeddings are the same, and the distance between mean embeddings is the *maximum mean discrepancy* (MMD) (Gretton et al., 2012a). Tests based on checking for differences in mean embeddings evaluated at specific locations (Chwialkowski et al., 2015; Jitkrittum et al., 2016) and the kernel Fisher discriminant analysis (Harchaoui et al., 2007) are also related methods. These non-parametric tests work well for samples from simple distributions when using appropriate kernels and can be applied to machine learning problems such as domain adaptation, generative modeling, and causal discovery (Gong et al., 2016; Lopez-Paz & Oquab, 2017; Binkowski et al., 2018; Stojanov et al., 2019).

The detection problem we address can be framed as the *statistical membership inference for privacy breach detection*. In short, given a well-trained classifier $\hat{f}$ on $S_T$, the problem we are concerned with is determining if an incoming data set $S_Y$ has the same distribution as a non-training set $S_X$ and contains no elements from the training set $S_T$. Although previous two-sample tests can readily solve the former challenge, they are incapable of addressing the latter. This is because the raw features of both the non-training and training sets originate from the same underlying distribution. **Therefore, in this paper, we focus on solving the latter issue.** Our motivation stems from studies on the overfitting phenomenon (Shokri et al., 2017a; Long et al., 2018; Leino & Fredrikson, 2020a), which reveal that neural networks inevitably capture noise, leading to varying degrees of overfitting. This, in turn, results in the generalization gap between non-training and training data. Through our investigation, we have discovered that the high-level representations extracted by the well-trained classifier $\hat{f}$ exhibit significant differences between the non-training and training data (e.g., as graphically illustrated in Figure 1 (b-c)). This inherent disparity in high-level representations forms the foundation of our proposed method, which leverages these differences to effectively distinguish between members and non-members. In this study, we broaden the scope of our analysis on high-level representations by incorporating a wider array of statistics derived from these representations, including confidence scores and several widely-recognized metrics within the MIA literature, such as loss value (Yeom et al., 2018b), entropy (Salem et al., 2019a), and likelihood ratio (Carlini et al., 2022).

In summary, we propose a novel and effective test that employs a high-level representation-based kernel, termed *High-level Representation-based MMD* (HR-MMD), to assess whether a given privacy-evaluation dataset contains elements from the training set. Unlike existing MIAs, which focus on distinguishing individual data records as members or non-members and often exhibit non-ideal evaluation reliability, our approach is designed to operate at a broader granularity, distinguishing between entire sets of records. Our objective is to achieve high test power and an ideal type I error rate. By developing a non-parametric two-sample test for membership inference, our goal is to establish a robust statistical methodology capable of accurately detecting privacy breaches in hindsight when compromised records have been used to train a third-party model. Extensive experiments across image and non-image datasets demonstrate that our HR-MMD test successfully meets these objectives, exhibiting high sensitivity in distinguishing between training and non-training records.

Our contribution lies in being the first to introduce a non-parametric two-sample test for membership inference, and we provide theoretical analysis on the asymptotics of HR-MMD to further substantiate its robustness. We emphasize that existing two-sample test methods cannot effectively address the problem of statistical membership inference for privacy breach detection. The effectiveness of our HR-MMD test stems from two key aspects: the superior performance of high-level representations and the novel deep kernel design. Our utilization of these high-level representations and the incorporation of a high-level representation-based deep kernel in the context of MIAs distinguishes our work from existing literature. As membership privacy is one of the most established methods for quantifying privacy risks in ML models (Murakonda & Shokri, 2020), we believe our contribution offers valuable insights into an alternative scenario for privacy breach detection and, in the long run, opens up a promising avenue for privacy evaluation based on membership inference tests.

## 2 PRELIMINARIES

### 2.1 CLASSIFICATION

In this paper, we focus on classification, one of the most common machine learning tasks. Let $f : \mathbb{R}^n \to [0,1]^K$ denote a neural network and let $K$ denote the number of classes. The output of the network is computed using the softmax function, which ensures that the output is a proper probability vector. Namely, given an input $x \in \mathbb{R}^n$, we have $f(x) = [p_1, \ldots, p_K] = p$, where $\sum_{i=1}^K p_i = 1$ and $p_i$ is the probability that input $x$ belongs to class $i$. Before softmax, the output of the network represents the logits $z$, i.e., $p = \text{softmax}(z)$. The classifier assigns the label $y = \arg\max_i f(x)_i$. To construct a machine learning model, one needs to collect a set of data samples, which is referred to as the training set $S_T$. The model is then built through a training process that aims to minimize a predefined loss function using optimization algorithms, such as SGD.

### 2.2 NON-PARAMETRIC TWO-SAMPLE TEST

Let $\mathcal{X} \subset \mathbb{R}^d$ and $\mathbb{P}$, $\mathbb{Q}$ be Borel probability measures on $\mathcal{X}$. Given IID samples $S_X = \{\boldsymbol{x}_i\}_{i=1}^n \sim \mathbb{P}^n$ and $S_Y = \{\boldsymbol{y}_j\}_{j=1}^m \sim \mathbb{Q}^m$, in the two-sample test problem, we aim to determine if $S_X$ and $S_Y$ come from the same distribution, i.e., if $\mathbb{P} = \mathbb{Q}$.

We use the null hypothesis testing framework, where the null hypothesis $H_0 : \mathbb{P} = \mathbb{Q}$ is tested against the alternative hypothesis $H_1 : \mathbb{P} \neq \mathbb{Q}$. We perform a two-sample test in four steps: (i) select a significance level $\alpha \in [0,1]$, (ii) compute a test statistic $\hat{t}(S_{\mathbb{P}}, S_{\mathbb{Q}})$, (iii) compute the $p$-value $\hat{p} = \text{Pr}_{H_0}(T > \hat{t})$, i.e., the probability of the two-sample test returning a statistic as large as $\hat{t}$ when $H_0$ is true, and (iv) finally, reject $H_0$ if $\hat{p} < \alpha$.

### 2.3 MAXIMUM MEAN DISCREPANCY

The *maximum mean discrepancy* (MMD) aims to measure the closeness between two distributions $\mathbb{P}$ and $\mathbb{Q}$:

$$\text{MMD}(\mathbb{P}, \mathbb{Q}; \mathcal{F}) := \sup_{f \in \mathcal{F}} |\mathbb{E}[f(X)] - \mathbb{E}[f(Y)]|, \tag{1}$$

where $\mathcal{F}$ is a set containing all continuous functions (Gretton et al., 2012a). To obtain an analytic solution regarding the sup in Eq.(1), Gretton et al. (2012a) restricts $\mathcal{F}$ to be a unit ball in the *reproducing kernel Hilbert space* (RKHS) and obtains the kernel-based MMD defined as follows:

$$\text{MMD}(\mathbb{P}, \mathbb{Q}; \mathcal{H}_k) := \sup_{f \in \mathcal{H}, \|f\|_{\mathcal{H}_k} \leq 1} |\mathbb{E}[f(X)] - \mathbb{E}[f(Y)]|$$

$$= \|\mu_{\mathbb{P}} - \mu_{\mathbb{Q}}\|_{\mathcal{H}_k}, \tag{2}$$

where $k$ is a bounded kernel w.r.t. a RKHS $\mathcal{H}_k$ (i.e., $|k(\cdot,\cdot)| < +\infty$), $X \sim \mathbb{P}$, $Y \sim \mathbb{Q}$ are two random variables, and $\mu_{\mathbb{P}} := \mathbb{E}[k(\cdot, X)]$ and $\mu_{\mathbb{Q}} := \mathbb{E}[k(\cdot, Y)]$ are kernel mean embeddings of $\mathbb{P}$ and $\mathbb{Q}$, respectively (Gretton et al., 2005; 2012a; Jitkrittum et al., 2016; 2017; Sutherland et al., 2017; Liu et al., 2020b). From Eq.(1), it is clear that MMD equals zero *if and only if* $\mathbb{P} = \mathbb{Q}$ (Gretton et al., 2008). As for the MMD defined in Eq.(2), Gretton et al. (2012a) also proves this property, which means that we could *in principle* use the MMD to test whether two distributions are the same.

In the MMD-based test, we are given two samples observed from $\mathbb{P}$ and $\mathbb{Q}$, and we aim to check whether these two samples come from the same distribution. Specifically, we first *estimate* the MMD value from two samples, and then compute the $p$-value corresponding to the estimated MMD value (Sutherland et al., 2017). If the $p$-value is above a given threshold $\alpha$, then two samples are considered to come from the same distribution. In the last decade, MMD-based tests have been used to detect the distributional discrepancy in several application scenarios, including high-energy physics data (Chwialkowski et al., 2015), amplitude modulated signals (Gretton et al., 2012b), and challenging image datasets, e.g., the *MNIST* and the *CIFAR-10* (Liu et al., 2020b).

### 2.4 ESTIMATING MMD

There are several natural estimators of the MMD from samples. We will assume n = m and estimate the MMD (Eq.(2)) using the $U$-statistic estimator, which is unbiased for $\text{MMD}^2$ and has nearly

minimal variance among unbiased estimators (Gretton et al., 2012a):

$$\widehat{\text{MMD}}_u^2(S_X, S_Y; k) = \frac{1}{n(n-1)} \sum_{i \neq j} H_{ij}, \tag{3}$$

$$H_{ij} = k(\boldsymbol{x}_i, \boldsymbol{x}_j) + k(\boldsymbol{y}_i, \boldsymbol{y}_j) - k(\boldsymbol{x}_i, \boldsymbol{y}_j) - k(\boldsymbol{y}_i, \boldsymbol{x}_j),$$

where $\boldsymbol{x}_i, \boldsymbol{x}_j \in S_X$ and $\boldsymbol{y}_i, \boldsymbol{y}_j \in S_Y$.

## 3 PROBLEM SETTING

We aim to address the following problem of *statistical membership inference for privacy breach detection*.

**Problem 1.** *Let $\mathcal{X}$ be a subset of $\mathbb{R}^d$ and $\mathbb{P}$ be a Borel probability measure on $\mathcal{X}$, and let $S_T = \{\boldsymbol{x}_i'\}_{i=1}^n \sim \mathbb{P}^n$ be IID observations from $\mathbb{P}$. Assume that the privacy evaluator can obtain (i) a well-trained classifier $\hat{f}$ trained on $S_T$, and (ii) IID observations $S_X$ from $\mathbb{P}$, where $S_T \cap S_X = \emptyset$ (a validation dataset for $\hat{f}$). For reasonably large $m$ values, we aim to determine if a given privacy-evaluation dataset $S_Y = \{\boldsymbol{y}_i\}_{i=1}^m$ contains elements from the training set $S_T$.*

In Problem 1, if $S_Y$ does not contain elements from the $S_T$, given a significance level $\alpha$, we aim to accept the null hypothesis $H_0$ (i.e., $S_Y$ are not the members of the training set of the classifier $\hat{f}$) against the alternative hypothesis $H_1$ (i.e., $S_Y$ contains members of the training set of the classifier $\hat{f}$) with the probability $1 - \alpha$. Conversely, if $S_Y$ contains elements from the $S_T$, we aim to reject the null hypothesis $H_0$ with a probability near to 1. The probability of not committing a type II error is called the power of the hypothesis test. Achieving a high test power requires sufficient samples, which can accurately reflect the sample distribution. Unsurprisingly, the test power is influenced by the sample size (denoted as $m$ in Problem 1). A minimum sample size $M$ can be set to prevent an inadequately small sample size, leading to unreliable hypothesis testing. More details on this aspect can be found in our analysis on the test power of our hypothesis under varying sample sizes. Additionally, we discussed the scenario where $S_Y$ contains *out-of-distribution* samples in Appendix D. The used important notations can also refer to Table 1.

In our framework, we rely on the following assumptions: **I.** The evaluator has access to the target model, which essentially follows the setting of black-box MIAs (see Appendix A for a more detailed literature review). We believe this is a realistic assumption, whenever the target black-box model can be queried freely, e.g., in the setting of *Machine Learning as a Service* (MLaaS), where query results can be stored locally by the evaluator; **II.** The evaluator has a validation set ($S_X$ in Problem 1), which is also a realistic assumption as validation sets are widely used in machine learning (e.g., the local shadow dataset in (Chen et al., 2021)); **III.** A *generalization gap* (test error minus training error) holds. Note that a generalization gap has been observed by practitioners during the training of deep learning models (Song et al., 2017; Carlini et al., 2019; Murakonda & Shokri, 2020; Zhang et al., 2021). It is a common assumption in MIAs and many works (Shokri et al., 2017b; Yeom et al., 2018b; Salem et al., 2019b; Leino & Fredrikson, 2020b; Chen et al., 2020) have pointed out that overfitting of the target ML models is the main factor contributing to the success of MIAs; **IV.** All the data mentioned in Problem 1 are *independent and identically distributed* (IID) observations.

## 4 LIMITATIONS OF PRIOR RESEARCH FOR PRIVACY BREACH DETECTION

There is a consensus in established computer security practices (Lazarevic et al., 2003; Vangelis et al., 2006; Kolter & Maloof, 2006; Kantchelian et al., 2015; Ho et al., 2017) that designing methods should focus on achieving low false positive rate (FPR). This viewpoint is also supported by previous state-of-the-art MIAs (Carlini et al., 2022), which argue that membership inference should prioritize the true positive rate (TPR) at exceedingly low FPRs, often approaching zero. Such stringent requirements are essential to ensure the reliability and credibility of privacy breach detections, thereby preventing unjustified accusations that could have severe repercussions for stakeholders.

In alignment with these security standards, our research is premised on the assertion that any detection method exhibiting a non-ideal FPR cannot be deemed successful for evaluating privacy breaches. Our objective is to develop privacy breach detection techniques that achieve an ideal TPR of $100\%$ while

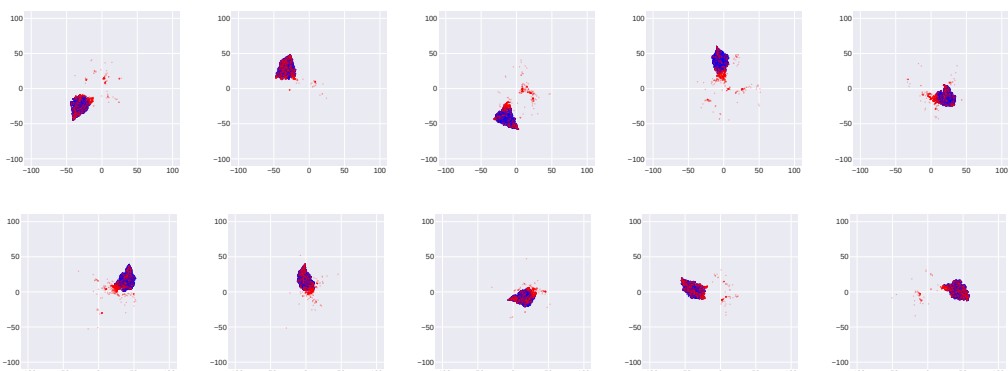

Figure 2: Visualization of high-level representation in ResNet-18, by t-SNE (Maaten & Hinton, 2008). Each subfigure corresponds to a *CIFAR-10* class, with red dots representing the confidence scores of non-members and blue dots representing the confidence scores of members. The high-level representation of members appears different from the one of non-members in each class.

maintaining an FPR of $0\%$, thereby adhering to the rigorous demands of computer security protocols. However, existing MIAs and two-sample tests fall short of meeting these stringent criteria. Traditional MIAs are fundamentally designed to assess individual instances, often leading to a compromised balance between TPR and FPR. And the drawback persists in scenarios where the subject being tested are sets of records (Maini et al., 2021; Kandpal et al., 2023; Maini et al., 2024), yet only a small proportion are training members. For instance, an intuitive application of traditional MIAs involves computing average statistics such as loss and setting a threshold to predict the presence of training members within a dataset. Nonetheless, this approach proves unreliable because these methods are constrained by their heuristic foundations, which lack the robust statistical guarantees necessary for high-stakes privacy evaluations.

Moreover, the inherent design limitations of existing MIAs render them incapable of providing the ideal FPR required for dependable privacy breach detection. These methods, which rely heavily on heuristic motivations—such as the assumption that samples with lower loss values are more likely to be members—fail to offer the same level of statistical rigor as our proposed HR-MMD test. Consequently, they invariably result in non-ideal FPRs, which undermine their utility in practical scenarios where false positives must be minimized. Empirical evaluations, detailed in Appendix H.1, demonstrate that existing MIAs including Loss (Yeom et al., 2018b), Entropy (Salem et al., 2019a), Attack R (Ye et al., 2022), and Likelihood Ratio Attack (LiRA) (Carlini et al., 2022) fail to simultaneously achieve the desired TPR and FPR under conditions where non-training members predominate the dataset. Specifically, as illustrated in Figure 1 (a), these traditional MIA methods are unable to attain an ideal TPR of $100\%$ and an FPR of $0\%$, primarily due to their disproportionate focus on the characteristics of non-training members, who constitute the majority in such settings.

Additionally, while two-sample tests provide a robust statistical framework for assessing whether two datasets originate from the same distribution and for maintaining an ideal FPR, they are unsuitable for addressing the specific challenge of statistical membership inference in privacy breach detection. This limitation is particularly pronounced when both members and non-members are drawn from identical distributions. Such inadequacies reveal a significant gap in existing privacy breach detection methodologies, underscoring the necessity for innovative approaches that can operate effectively at the granularity of record sets while ensuring both high sensitivity and reliability.

## 5 A HIGH-LEVEL REPRESENTATION-BASED MMD TEST

In response to above significant limitations, we introduce the *High-level Representation-based MMD* (HR-MMD) test. HR-MMD leverages high-level representations extracted from well-trained classifiers to enhance the discriminative power between training and non-training sets. By integrating novel deep kernel designs specifically tailored to capture membership-induced variations, HR-MMD overcomes the inherent deficiencies of prior MIAs and existing two-sample tests. This innovative

approach offers a robust and statistically rigorous solution for privacy breach detection, capable of meeting the stringent requirements of ideal FPR and ideal TPR in complex real-world scenarios.

**High-level Representation.** As outlined in Section 3, a generalization gap, defined as the difference between test error and training error ($g = e_{\text{test}} - e_{\text{train}} > 0$), is commonly observed in real-world machine learning models. Formally, it is expressed as:

$$g = \frac{1}{N_{\text{train}}} \sum_{i=1}^{N_{\text{train}}} \mathcal{L}\left(f\left(x_i^{\text{train}}\right), y_i^{\text{train}}\right) - \frac{1}{N_{\text{test}}} \sum_{j=1}^{N_{\text{test}}} \mathcal{L}\left(f\left(x_j^{\text{test}}\right), y_j^{\text{test}}\right), \tag{4}$$

where $\mathcal{L}(\cdot, \cdot)$ denotes the loss function. The condition $g > 0$ signifies that there is a discrepancy in the error distributions between the training set (members) and the test set (non-members).

Based on how machine learning models work, this divergence in error distributions implies that the difference may also exist in other high-level representations. Empirical evidence supporting this phenomenon is illustrated in Figures 1 (b-c) and 2, where t-SNE (Maaten & Hinton, 2008) visualizations of the high-level representations extracted from a well-trained deep neural network (e.g., ResNet-18) reveal distinct clustering patterns between members and non-members. Notably, while the raw input features of these datasets may appear similarly distributed within their respective semantic classes, the high-level representations exhibit clear separations attributable to the underlying generalization gap. Additional visualizations across different network layers are provided in Appendix F.

Motivated by these observations, we propose to develop a statistical testing method to distinguish members from non-members by leveraging the differences in high-level representations. Unlike traditional MIAs that rely on specific metrics such as loss values, our high-level representation-based approach explores a broader spectrum of high-level representations to enhance the discriminative power between training and non-training sets, and previous statistics like loss, entropy, and likelihood ratio can also be easily included in our framework. This flexibility is particularly advantageous in real-world scenarios where certain information may be unavailable or deliberately obfuscated. For instance, in settings where true labels are inaccessible, making loss-based metrics infeasible, our framework can leverage alternative high-level representations like confidence scores. Furthermore, adversarial strategies such as MemGuard (Jia et al., 2019) introduce perturbations to confidence scores to thwart membership inference, rendering loss-based and confidence score-based methods less effective. In such cases, our HR-MMD test can employ other high-level representations, such as those from the penultimate layer, to maintain robust membership inference capabilities. By accommodating a diverse array of high-level representations, HR-MMD extends the evaluative toolkit for privacy breach detection, ensuring adaptability and resilience across varying operational conditions.

**High-level Representation-based MMD.** Based on the high-level representation, we consider the following deep kernel $k_\omega(\boldsymbol{x}, \boldsymbol{y})$ to measure the feature similarity:

$$k_\omega(\boldsymbol{x}, \boldsymbol{y}) = \left[(1 - \epsilon_0)s_{\hat{f}}(\boldsymbol{x}, \boldsymbol{y}) + \epsilon_0\right]q(\boldsymbol{x}, \boldsymbol{y}), \tag{5}$$

where $s_{\hat{f}}(\boldsymbol{x}, \boldsymbol{y}) = \kappa(\phi_p(\boldsymbol{x}), \phi_p(\boldsymbol{y}))$ is a deep kernel function that measures the feature similarity between $\boldsymbol{x}$ and $\boldsymbol{y}$ using high-level representations extracted by $\hat{f}$. We use $\phi_p$ to extract high-level representations, which can include features from any higher layer, confidence scores, loss value, entropy (Salem et al., 2019a), and likelihood ratio (Carlini et al., 2022). $\kappa$ is the Gaussian kernel (with bandwidth $\sigma_{\phi_p}$), $\alpha \in [0, 1]$ is the weight for $s_\phi(\boldsymbol{x}, \boldsymbol{y})$, while $\epsilon_0 \in (0, 1)$ and $q(\boldsymbol{x}, \boldsymbol{y})$ (the Gaussian kernel with bandwidth $\sigma_q$) are key components to ensure that $k_\omega(\boldsymbol{x}, \boldsymbol{y})$ is a characteristic kernel (i.e., ensuring that HR-MMD equals zero if and only if two distributions are the same (Liu et al., 2020b)).

Since $\hat{f}$ is fixed, the set of parameters of $k_\omega$ is $\omega = \{\epsilon_0, \sigma_{\phi_p}, \sigma_q\}$. Based on $k_\omega(\boldsymbol{x}, \boldsymbol{y})$ in Eq.(5), HR-MMD$(\mathbb{P}, \mathbb{Q})$ is

$$\sqrt{\mathbb{E}\left[k_\omega(X, X') + k_\omega(Y, Y') - 2k_\omega(X, Y)\right]},$$

where $X, X' \sim \mathbb{P}, Y, Y' \sim \mathbb{Q}$. We can estimate HR-MMD$(\mathbb{P}, \mathbb{Q})$ using the $U$-statistic estimator, which is unbiased for HR-MMD$^2(\mathbb{P}, \mathbb{Q})$:

$$\widehat{\text{HR-MMD}}_u^2(S_X, S_Y; k_\omega) = \frac{1}{n(n-1)} \sum_{i \neq j} H_{ij}, \tag{6}$$

---

**Algorithm 1** High-level Representation-based MMD Test

---

**Input:** $S_X$, $S_Y$, $\phi$, various hyperparameters used below;

$\omega \leftarrow \omega_0$; $\lambda \leftarrow 10^{-8}$; $\beta \leftarrow 0.5$;         ▷ *Initialize parameters*

Split the data as $S_X = S_X^{tr} \cup S_X^{te}$ and $S_Y = S_Y^{tr} \cup S_Y^{te}$;   ▷ *Divide data into training and testing sets*

\# *Phase 1: training the kernel parameters $\omega$ on $S_X^{tr}$ and $S_Y^{tr}$*

**for** $T = 1, 2, \ldots, T_{max}$ **do**

    $S_X' \leftarrow$ minibatch from $S_X^{tr}$;               ▷ *Sample a minibatch from $S_X^{tr}$*

    $S_Y' \leftarrow$ minibatch from $S_Y^{tr}$;               ▷ *Sample a minibatch from $S_Y^{tr}$*

    $k_\omega \leftarrow$ kernel function with parameters $\omega$ using Eq.(5); ▷ *Define kernel function based on current $\omega$*

    $M(\omega) \leftarrow \widehat{\text{HR-MMD}}_u^2(S_X', S_Y'; k_\omega)$ using Eq.(6); ▷ *Compute the HR-MMD statistic for minibatch*

    $V_\lambda(\omega) \leftarrow \hat{\sigma}_{H_1,\lambda}^2(S_X', S_Y'; k_\omega)$ using Eq.(9);     ▷ *Estimate variance under $H_1$ with regularization*

    $\hat{J}_\lambda(\omega) \leftarrow M(\omega)/\sqrt{V_\lambda(\omega)}$ using Eq.(8);     ▷ *Compute the objective function for optimization*

    $\omega \leftarrow \omega + \eta \nabla_{\text{Adam}} \hat{J}_\lambda(\omega)$;     ▷ *Update kernel parameters via gradient ascent to maximize $\hat{J}_\lambda(\omega)$*

**end for**

\# *Phase 2: testing with $k_\omega$ on $S_X^{te}$ and $S_Y^{te}$*

$est \leftarrow \widehat{\text{HR-MMD}}_u^2(S_X^{te}, S_Y^{te}; k_\omega)$     ▷ *Compute the HR-MMD statistic on the test data*

**for** $i = 1, 2, \ldots, n_{perm}$ **do**

    Shuffle $S_X^{te} \cup S_Y^{te}$ into $X$ and $Y$     ▷ *Shuffle data to create permutation samples*

    $perm_i \leftarrow \widehat{\text{HR-MMD}}_u^2(X, Y; k_\omega)$     ▷ *Compute the HR-MMD statistic for each permutation*

**end for**

**Output:** $p$-value: $\frac{1}{n_{perm}} \sum_{i=1}^{n_{perm}} \mathbb{1}(perm_i \geq est)$     ▷ *Return the p-value*

---

where $H_{ij} = k_\omega(\boldsymbol{x}_i, \boldsymbol{x}_j) + k_\omega(\boldsymbol{y}_i, \boldsymbol{y}_j) - k_\omega(\boldsymbol{x}_i, \boldsymbol{y}_j) - k_\omega(\boldsymbol{y}_i, \boldsymbol{x}_j)$.

**Asymptotics and test power of HR-MMD.** We analyze next the asymptotics of HR-MMD, when $S_Y$ represents the *target data*. Based on the asymptotics of HR-MMD, we can estimate its test power and use it to optimize HR-MMD (i.e., optimize the parameters in $k_\omega(\boldsymbol{x}, \boldsymbol{y})$).

**Theorem 1** (Asymptotics under $H_1$ – simplified). *Under the alternative $H_1 : S_Y$ are from a stochastic process $\{Y_i\}_{i=1}^{+\infty}$, under mild assumptions, we have*

$$\sqrt{n}(\widehat{\text{HR-MMD}}_u^2 - \text{HR-MMD}^2) \xrightarrow{d} \mathcal{N}(0, C_1^2 \sigma_{H_1}^2),$$

*where $Y_i = \mathcal{G}_{\ell, \hat{f}}(\mathcal{B}_\epsilon[X_i]) \sim \mathbb{Q}$, $X_i \sim \mathbb{P}$, $\sigma_{H_1}^2 = 4(\mathbb{E}_Z[(\mathbb{E}_{Z'} h(Z, Z'))^2] - [(\mathbb{E}_{Z,Z'} h(Z, Z'))^2])$, $h(Z, Z') = k_\omega(X, X') + k_\omega(Y, Y') - k_\omega(X, Y') - k_\omega(X', Y)$, $Z := (X, Y)$ and $C_1 < +\infty$ is a constant for a given $\omega$.*

The detailed formulation and proof of Theorem 1 can be found in Appendix C.

Using Theorem 1, we have

$$\text{Pr}_{H_1, r}^{\text{HR-MMD}} \rightarrow \Phi\Big(\frac{\sqrt{n}\text{HR-MMD}^2}{C_1 \sigma_{H_1}} - \frac{r}{\sqrt{n} C_1 \sigma_{H_1}}\Big), \tag{7}$$

where $\text{Pr}_{H_1, r}^{\text{HR-MMD}} = \text{Pr}_{H_1}\left(n\widehat{\text{HR-MMD}}_u^2 > r\right)$ is the test power of HR-MMD, $\Phi$ is the standard normal CDF and $r$ is the rejection threshold related to $\mathbb{P}$ and $\mathbb{Q}$.

Via Theorem 1, we know that $r$, $\text{HR-MMD}(\mathbb{P}, \mathbb{Q})$, and $\sigma_{H_1}$ are constants. Thus, for reasonably large $n$, the test power of HR-MMD is dominated by the first term (inside $\Phi$), and we can optimize $k_\omega$ by maximizing

$$J(\mathbb{P}, \mathbb{Q}; k_\omega) = \text{HR-MMD}^2(\mathbb{P}, \mathbb{Q}; k_\omega)/\sigma_{H_1}(\mathbb{P}, \mathbb{Q}; k_\omega).$$

Note that we omit $C_1$ in $J(\mathbb{P}, \mathbb{Q}; k_\omega)$, since $C_1$ can be upper bounded by a constant $C_0$ (see Appendix C).

**Optimization of HR-MMD.** Although a higher value for the criterion $J(\mathbb{P}, \mathbb{Q}; k_\omega)$ means a higher test power for HR-MMD, we cannot directly maximize $J(\mathbb{P}, \mathbb{Q}; k_\omega)$, since HR-MMD$^2(\mathbb{P}, \mathbb{Q}; k_\omega)$ and $\sigma_{H_1}(\mathbb{P}, \mathbb{Q}; k_\omega)$ depend on the particular $\mathbb{P}$ and $\mathbb{Q}$ that are unknown. However, we can estimate it with

$$\hat{J}_\lambda(S_X, S_Y; k_\omega) := \frac{\widehat{\text{HR-MMD}}_u^2(S_X, S_Y; k_\omega)}{\hat{\sigma}_{H_1, \lambda}(S_X, S_Y; k_\omega)}, \tag{8}$$

where $\hat{\sigma}_{H_1, \lambda}^2$ is a regularized estimator of $\sigma_{H_1}^2$ (Liu et al., 2020b):

$$\frac{4}{n^3} \sum_{i=1}^n \left( \sum_{j=1}^n H_{ij} \right)^2 - \frac{4}{n^4} \left( \sum_{i=1}^n \sum_{j=1}^n H_{ij} \right)^2 + \lambda. \tag{9}$$

Then, we can optimize HR-MMD by maximizing $\hat{J}_\lambda(S_X, S_Y; k_\omega)$ on the $S_X^{tr}$ and $S_Y^{tr}$ (see Algorithm 1).

Note that, although (Sutherland et al., 2017) and (Sutherland, 2019) have given an unbiased estimator for $\sigma_{H_1}^2$, it is much more complicated to implement.

**The HR-MMD test.** Given the assumption that the *target data records* are IID, we can test if $S_Y$ are from $\mathbb{P}$ and $S_Y \cap S_X = \emptyset$. To this end, Algorithm 1 describe the complete flow of the HR-MMD test. In Appendix C, we shown that, under mild assumptions, the proposed HR-MMD test is a *provably consistent* test to detect privacy breaches by membership inference for sets of records.

**Time complexity.** The time complexity of Algorithm 1 can be broken down into two phases: training and testing. During the training phase, the complexity per iteration is $O(nE + n^2 K)$, where $E$ represents the cost of computing an embedding $\phi_p(x)$, $K$ is the cost of computing Eq.(5) given $\phi_p(x), \phi_p(y)$, and $n$ is the minibatch size. For moderate values of $n$ that fit within a GPU-sized minibatch, the $nE$ term typically dominates, making the complexity comparable to that of C2STs (Lopez-Paz & Oquab, 2017). In the testing phase, the time complexity is $O(nE + n^2 K + n^2 n_{\text{perm}})$, compared to the $O(nE + nn_{\text{perm}})$ for permutation-based C2STs.

## 6 EXPERIMENTS

We evaluate the effectiveness of our methods across four image datasets: **CIFAR-10** (Krizhevsky et al., 2009), **CIFAR-100** (Krizhevsky et al., 2009), **SVHN** (Netzer et al., 2011), **CINIC-10** (Darlow et al., 2018), across two different model architectures: **ResNet-18** (He et al., 2016), and **Wide-ResNet-34** (Zagoruyko & Komodakis, 2016). We also show the capacity for extending to a tabular dataset **Purchase-100** (Shokri et al., 2017b) and a text dataset **IMDB** (Maas et al., 2011).

**Baseline methods selection.** Traditional MIAs are not *standard hypothesis testing methods* (such as two-sample tests that guarantee the ideal type I error and then compare test power), making them unacceptable from a statistical perspective. Therefore, when comparing our methods with traditional MIAs, we use an experimental setting (FPR vs TPR) that differs from the one used for comparisons with statistical testing methods, ensuring an appropriate and fair comparison. The detailed experimental setup and results can be found in Appendices G.7 and H.1, respectively. As discussed in Section 4 and Appendix G.5, although the limitations of traditional MIA methods prevent them from being directly applied to privacy breach detection problems, the statistics from MIAs can be incorporated into our non-parametric two-sample test framework (L-MMD, E-MMD, LI-MMD). In the main experiments, we focus on comparing our six HR-MMD test series (HR1-MMD, HR2-MMD, CS-MMD, L-MMD, E-MMD, LI-MMD), defined in Appendix G.5, with six traditional two-sample tests (MMD-G, MMD-O, ME, SCF, C2ST, MMD-D), introduced in Appendix G.3. Additionally, we compare two of our baselines, LR1-MMD and LR2-MMD, also described in Appendix G.5. All baselines are specifically selected for their ability to maintain an ideal type I error rate.

**Experimental results.** As shown in Figure 3, our extensive experimental evaluation demonstrates the effectiveness of the HR-MMD tests. The ablation study further highlights the superior performance of the high-level representations and our deep kernel design. Comprehensive supplementary experimental results, including detailed analyses across different datasets, model architectures, sample sizes, and practical scenarios involving mixed data, are provided in Appendix H.

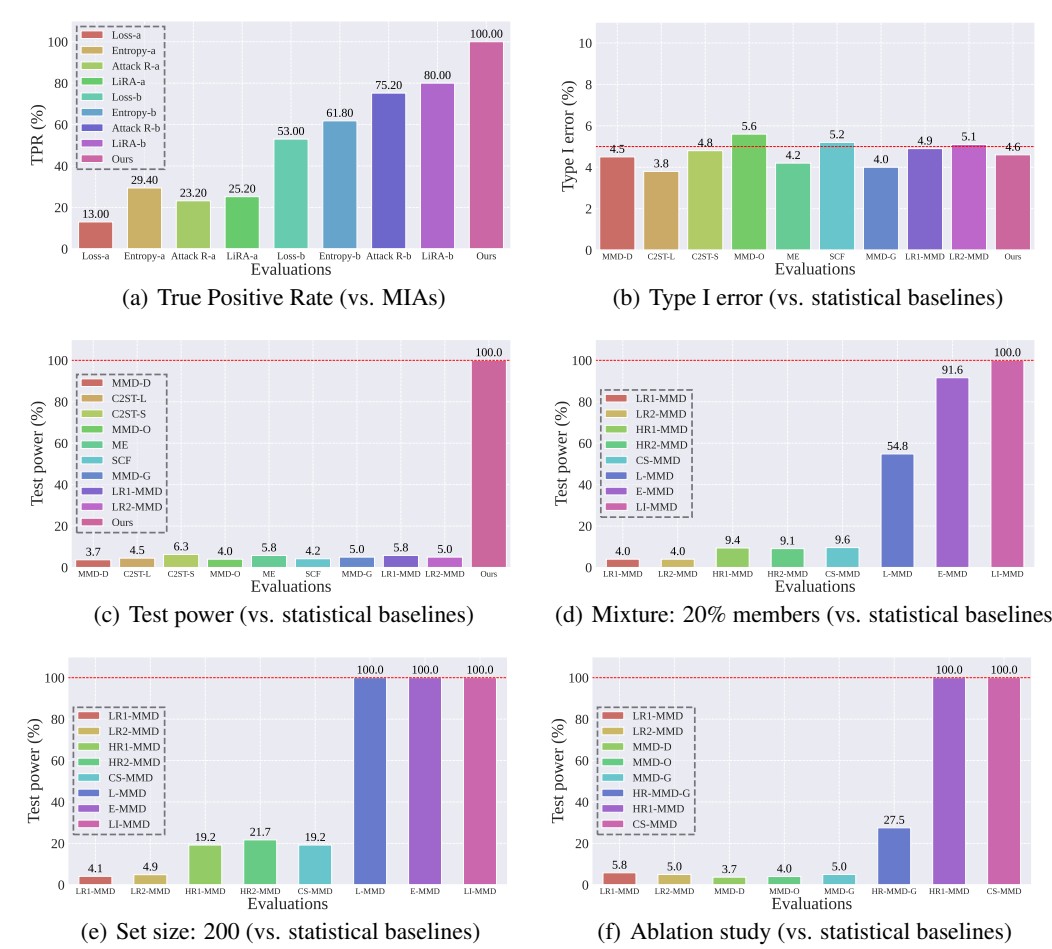

Figure 3: Comparison between our HR-MMD test and other baselines. As shown in subfigure (a), traditional MIAs fail to simultaneously achieve the desired TPR and FPR when non-training members make up 90% of the testing dataset. Thresholds for MIAs were set at different false positive rates (1% for group a, e.g., Loss-a; 10% for group b, e.g., Loss-b). The false positive rate of our proposed HR-MMD test is 0%. The detailed experimental setup is provided in Appendix G.7. Subfigure (b) shows that all statistical baselines achieve an ideal type I error around the significance level of 0.05. In subfigures (c-f), only our HR-MMD test achieves effective test power (greater than 0.05). The experiments are conducted on CIFAR-10 with ResNet-18. Detailed experimental settings and comprehensive results can be found in Appendices G and H.

## 7 CONCLUSION

The risk of unsanctioned data accesses, leading to sensitive data being used in third-party predictive models cannot be ignored. Such accesses may become apparent after the fact, in the models that have been trained on compromised data. This calls for effective membership inference methods, enabling an evaluator to identify privacy breaches[1]. Existing membership inference attacks (MIAs) – on whether a given data record may have been used to train a model or not – suffer from non-ideal evaluation reliability. Therefore, a change both in perspective and in approach in necessary, compared to the MIAs' setting, to enable privacy evaluators to do membership inference for sets of records. In this paper, we propose a novel non-parametric two-sample test, which leverages the differences between high-level representations, from training data and non-training data, to detect whether a set of samples was used for training a given model. Our experiments show that the proposed HR-MMD test exhibits remarkably high sensitivity in distinguishing between the training and non-training sets, with the ideal type I error, making it a powerful membership inference tool for detecting data breaches from machine learning models.

---

[1]We describe three additional application scenarios, pertaining to privacy auditing in *machine unlearning*, user inference and dataset inference, in Appendix B.

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

## A    RELATED WORKS

### A.1    PRIVACY ATTACKS IN MACHINE LEARNING

Machine learning (ML) models, owing to their deep integration in critical domains like healthcare (Miotto et al., 2018), genomics (Xiong et al., 2015), and image recognition (He et al., 2016), hold immense responsibility in safeguarding sensitive information. The success of these models primarily depends on extensive datasets and computational resources (Bengio et al., 2021). However, their capability to memorize training data (Song et al., 2017; Carlini et al., 2019; Zhang et al., 2021) opens avenues for various privacy attacks, highlighting the crucial need for robust defenses against privacy breaches.

Privacy attacks targeting ML models can be broadly categorized into model extraction attacks (Tramèr et al., 2016), attribute inference attacks (Fredrikson et al., 2015), property inference attacks (Ganju et al., 2018), and membership inference attacks (MIAs) (Shokri et al., 2017b). These attacks differ in their objectives and methodologies. While model extraction attacks replicate the target model's functionality, attribute and property inference attacks aim at deducing sensitive information or global properties of the training data, respectively. MIAs, conversely, ascertain whether a particular record was part of the model's training data, thereby posing severe privacy risks.

MIAs specifically target the data used in the training phase of ML models. The confidentiality breach through MIAs is so significant that it's recognized by institutions like the National Institute of Standards and Technology (NIST) (Tabassi et al., 2019) and regulations such as the General Data Protection Regulation (GDPR) (Wikipedia, 2021). The genesis of MIAs traces back to the work of Homer et al. (Homer et al., 2008), evolving over time to affect various ML models including, but not limited to, classification (Shokri et al., 2017b), regression (Gupta et al., 2021), and generation models (Hayes et al., 2019). The threat model typically assumes black-box access, where the attacker only has the model's prediction outputs.

Concurrently, research has proliferated around defensive mechanisms aiming to mitigate the risks posed by MIAs, balancing between preserving data privacy and maintaining model utility. These proposed defenses span across diverse methodologies, indicating the depth and complexity involved in securing ML models against MIAs. As machine learning continues to permeate sensitive applications, understanding and mitigating privacy attacks, especially MIAs, become paramount.

### A.2    MEMBERSHIP INFERENCE ATTACKS

Membership Inference Attacks (MIAs) in machine learning (ML) are pivotal in assessing privacy risks, where the attacker aims to determine if a specific data instance was used in training an ML model. Central to MIAs is the exploitation of the model's learned parameters and predictions. The efficacy of these attacks is closely linked to the adversary's knowledge, which encompasses insights into the training data distribution and details of the target model, including its architecture and parameters. The classification of MIAs into white-box and black-box attacks hinges on this knowledge spectrum, ranging from full access to the model's internals to limited information based on prediction outputs (Shokri et al., 2017b).

In binary classifier-based MIAs, the attack employs a classifier trained to distinguish between training data (members) and non-training data (non-members). This classifier's training involves shadow models to simulate the target model's behavior, a concept introduced by Shokri et al. (2017b). These shadow models, varying in the amount of accessible information, help construct datasets for training the attack classifier. The sophistication of the attack model differs in white-box and black-box settings, with the former having complete access to the target model, including intermediate computations and gradients, and the latter limited to prediction vectors.

Metric-based MIAs, a less complex alternative, infer membership status by calculating metrics on prediction vectors of data records against a predetermined threshold. These MIAs are categorized into several types based on the metric used, such as prediction correctness, loss, confidence, and entropy-based attacks, each with its implications and effectiveness in inferring membership (Yeom et al., 2018b; Salem et al., 2019b). Song & Mittal (2021) improved upon traditional entropy-based MIAs by incorporating ground truth label information into the entropy calculation, making the inference more accurate. Recently, advanced MIAs such as Attack R (Ye et al., 2022) and the Likelihood Ratio

Table 1: Notations

| Notation | Description | Availability for evaluator |
|----------|-------------|----------------------------|
| $\hat{f}$ | The classifier trained on $S_T$ | Yes (black-box) |
| $S_T$ | The training set of $\hat{f}$ | No |
| $S_X$ | A non-training set of $\hat{f}$ | Yes |
| $S_Y$ | The input test data | N/A |

Attack (LiRA) (Carlini et al., 2022) have been proposed to enhance the effectiveness of membership inference. Attack R introduces refined techniques to better capture differences in samples behaviors between the target model and the reference model. LiRA leverages likelihood ratios to improve the statistical power of the attack, achieving state-of-the-art results in certain settings.

In addition to instance-level MIAs, there is a growing interest in *set-based* membership inference attacks, where the adversary aims to determine if a *set* of records was part of the training data. Maini et al. (2021) introduced *Dataset Inference*, providing strong statistical evidence to detect if a given model is derived from its private training data. The key intuition behind their method is that classifiers maximize the distance of training examples from the model's decision boundaries, while test examples are closer to these boundaries since they have no impact on the model weights.

In the context of *large language models* (LLMs), Maini et al. (2024) introduced dataset inference methods to detect datasets used for training. They utilized adaptively weighted statistics but relied on weak t-tests for distinction. Similarly, Kandpal et al. (2023) proposed *User Inference* attacks targeting sets of samples to determine if data from specific users were included in the training set. This approach is akin to applying Attack R (Ye et al., 2022) in the context of LLMs. However, in non-large model scenarios, these methods exhibit significant limitations, as the statistical methods employed may not provide sufficient discriminatory power.

Overall, MIAs represent a significant privacy concern in ML, underscoring the necessity for effective defense mechanisms. The success of these attacks depends on the adversarial knowledge and the chosen methodology, ranging from sophisticated binary classifier-based attacks to simpler metric-based approaches, and extending to set-based attacks in the context of large models. This broad spectrum of attack strategies and defense considerations in MIAs reflects a dynamic area of research, essential for safeguarding privacy in the evolving landscape of machine learning and artificial intelligence (Shokri et al., 2017b; Song et al., 2017; Carlini et al., 2019; Nasr et al., 2019; Murakonda & Shokri, 2020; Zhang et al., 2021; Song & Mittal, 2021; Ye et al., 2022; Carlini et al., 2022; Maini et al., 2021; Kandpal et al., 2023; Maini et al., 2024).

### A.3 TWO-SAMPLE TESTS

Two-sample tests aim to check whether two datasets come from the same distribution. Traditional tests such as $t$-test and Kolmogorov-Smirnov test are the mainstream of statistical applications, but require strong assumptions on the distributions being studied. Researchers in statistics and machine learning have been focusing on relaxing these assumptions, with methods specific to various real-world domains (Sugiyama et al., 2011; Yamada et al., 2011; Kanamori et al., 2012; Gretton et al., 2012a; Jitkrittum et al., 2016; Sutherland et al., 2017; Chen & Friedman, 2017; Ghoshdastidar et al., 2017; Lopez-Paz & Oquab, 2017; Li & Wang, 2018; Kirchler et al., 2020; Liu et al., 2020b). In order to involve distributions with complex structure such as images, deep kernel approaches has been proposed (Sutherland et al., 2017; Wenliang et al., 2019; Jean et al., 2018), the foremost study has shown that kernels parameterized by deep neural nets, can be trained to maximize test power in high-dimensional distribution such as images (Liu et al., 2020b). They propose statistical tests of the null hypothesis that the two distributions are equal against the alternative hypothesis that the two distributions are different. Such tests have applications in a variety of machine learning problems such as domain adaptation, covariate shift, label-noise learning, generative modeling, fairness and causal discovery (Binkowski et al., 2018; Fang et al., 2020a; Gong et al., 2016; Fang et al., 2020b;

Liu et al., 2019; Zhang et al., 2020b;a; Liu et al., 2020a; Zhong et al., 2021; Yu et al., 2020; Stojanov et al., 2019; Lopez-Paz & Oquab, 2017; Oneto et al., 2020).

## A.4    MACHINE UNLEARNING

The *right to be forgotten* is a legal principle that allows individuals to request the deletion of their personal data from entities that store it. This right has gained significant attention in recent years due to concerns over privacy and data protection. To implement the *right to be forgotten* in the context of machine learning, it is necessary to ensure that any influence that the data sample may have on the *original model* is removed (Cao & Yang, 2015; Villaronga et al., 2018). This process, known as machine unlearning, has gained momentum both in academia and industry (Neel et al., 2021; Cao & Yang, 2015; Xue et al., 2016; Cao et al., 2018; Du et al., 2019; Ginart et al., 2019; Sommer et al., 2020; Golatkar et al., 2020; Guo et al., 2020; Liu et al., 2020c; Baumhauer et al., 2020; Izzo et al., 2021; Neel et al., 2021; Bourtoule et al., 2021).

In practice, implementing machine unlearning can be challenging. The most legitimate approach is to remove the data sample requested to be deleted (referred to as *target data*) and retrain the machine learning model from scratch. The model generated by this approach is referred to as the *retrained model*. However, this approach can incur high computational overhead. To mitigate this, several approximate approaches to machine unlearning have been proposed. The models generated by these approaches are referred to as the *unlearned models*. These techniques aim to minimize the computational overhead while ensuring that the outputs of the *unlearned model* and the *retrained model* are statistically indistinguishable (Izzo et al., 2021; Cao & Yang, 2015; Baumhauer et al., 2020; Bourtoule et al., 2021).

In addition to the approaches of machine unlearning, it is increasingly necessary to conduct research on their evaluation. Legislation such as the *general data protection regulation* (GDPR) in the European Union, the *california consumer privacy act* (CCPA) in California, and the *personal information protection and electronic documents act* (PIPEDA) in Canada have legally solidified the *right to be forgotten*. These laws provide individuals with the legal framework to exercise their right to request the deletion of their personal data, which makes the effective evaluation of machine unlearning approaches necessary. For example, as the regulator, we need to inspect the privacy protection effectiveness of machine unlearning of artificial intelligence service providers. Alternatively, as an artificial intelligence service provider who has employed a contractor to complete the machine unlearning service, we need to inspect the contractor's performance. Although statistical indistinguishability is a good measure of machine unlearning approaches in previous research (Izzo et al., 2021; Cao & Yang, 2015; Baumhauer et al., 2020; Bourtoule et al., 2021), it is not applicable in real-world scenarios where the *retrained model* is unavailable for evaluators.

## B    PRACTICAL SCENARIOS FOR MEMBERSHIP INFERENCE BY PRIVACY EVALUATORS

**Scenario 1 – Privacy breach detection.** A large-scale clinical trial for a common disease is conducted across multiple hospitals. Each hospital may recruit patients from a broad population base, ensuring a diverse and representative sample. This recruitment strategy is uniform across all hospitals, leading to an IID distribution of patient data at each hospital. Although clinical records are highly sensitive, different hospitals may be more or less able to protect them. Some data may be exposed to data theft, and may then make its way into training datasets $S_T$ for third-party predictive models (for that disease). By identifying whether a particular hospital's records ($S_Y$) have been used to train such models, a privacy evaluator aims to determine if clinical records from a specific hospital have been exposed to privacy attackers. Therefore, the regulator has black-box access to the model $\hat{f}$ associated with that disease and has access to a validation dataset $S_X$, and the goal is to assess whether the data $S_Y$ from the specific hospital contains elements of the training set $S_T$.

**Scenario 2 – Privacy auditing in machine unlearning.** In this scenario, the evaluator needs to assess the effectiveness of privacy measures achieved through *machine unlearning* by an ML service provider. The service provider claims to have performed machine unlearning on data $S_Y$ for the model $\hat{f}$ and asserts that data $S_Y$ has been successfully "unlearned" from the model. The evaluator, with black-box access to the updated model $\hat{f}$ (after the unlearning process), also has access to a

validation dataset $S_X$. The objective is to determine whether the data $S_Y$, which was supposedly unlearned, still poses any membership privacy risks by auditing if the model retains traces of $S_Y$ in its training set $S_T$.

**Scenario 3 – User inference in personalized recommendation systems.** A streaming service collects user interaction data to enhance its recommendation algorithms. While the service assures users that their personal data is anonymized and securely handled, concerns arise about whether specific users' data ($S_Y$) have been directly utilized in training the recommendation model $\hat{f}$. A privacy evaluator, with black-box access to $\hat{f}$ and a validation dataset $S_X$, aims to determine if the model's training set $S_T$ includes data from a particular user. The goal is to assess potential privacy risks and verify the company's compliance with user data protection policies by detecting any inadvertent inclusion of individual user data in the training process.

**Scenario 4 – Dataset inference for compliance auditing.** An organization develops a machine learning model $\hat{f}$ using a vast amount of data sourced from various datasets. Due to legal restrictions and licensing agreements, certain datasets ($S_Y$) are prohibited from being used in training. The organization claims adherence to these regulations, but an external auditor seeks to verify this compliance. With black-box access to the trained model $\hat{f}$ and a validation dataset $S_X$, the auditor employs dataset inference techniques to determine whether the restricted data $S_Y$ was inadvertently included in the training set $S_T$. This assessment helps ensure that the organization is not violating data usage policies and maintains the integrity of legal and ethical standards in its model development.

# C ASYMPTOTICS OF THE HR-MMD

In this section, we will prove the asymptotics of the HR-MMD by assuming that the upcoming data $\{Y_i\}_{i \in \mathbb{Z}^+}$ are an absolutely regular process with mixing coefficients $\{\beta_k\}_{k>0}$ defined in the following.

**Definition 1** (Absolutely regular process). *(i) Let $(\Omega, \mathcal{A}, \mathbb{Q})$ be a probability space, and let $\mathcal{A}_1, \mathcal{A}_2$ be sub-$\sigma$-field of $\mathcal{A}$. We define*

$$\beta(\mathcal{A}_1, \mathcal{A}_2) = \sup_{A_1,\dots,A_n,B_1,\dots,B_m} \sum_{i=1}^{n} \sum_{j=1}^{m} |\mathbb{Q}(A_i \cap B_j) - \mathbb{Q}(A_i)\mathbb{Q}(B_j)|, \qquad (10)$$

*where the supremum is taken over all partitions $A_1, \dots, A_n$ and $B_1, \dots, B_m$ of $\Omega$ into elements of $\mathcal{A}_1$ and $\mathcal{A}_2$, respectively.*
*(ii) Given a stochastic process $\{Y_i\}_{i \in \mathbb{Z}^+}$ and integers $1 \leq a \leq b$, we denote by $\mathcal{A}_a^b$ the $\sigma$-field generated by the random variables $Y_{a+1}, \dots, Y_b$. We define the mixing coefficients of absolute regularity by*

$$\beta_k = \sup_{n \in \mathbb{Z}^+} \beta(\mathcal{A}_1^n, \mathcal{A}_{n+k}^\infty). \qquad (11)$$

*The process $\{Y_i\}_{i \in \mathbb{Z}^+}$ is called absolutely regular if $\lim_{k \to \infty} \beta_k = 0$.*

Then, we can obtain the main theorem in the following.

**Theorem 2** (Asymptotics under $H_1$). *Under the alternative, $H_1 : S_Y$ are from a stochastic progress $\{Y_i\}_{i=1}^{+\infty}$, if $\{Y_i\}_{i=1}^{+\infty}$ is an absolutely regular process with mixing coefficients $\{\beta_k\}_{k>0}$ satisfying $\sum_{k=1}^{+\infty} \beta_k^{\delta/(2+\delta)} < +\infty$ for some $\delta > 0$, then $\widehat{\text{HR-MMD}}_u^2$ is $\mathcal{O}_P(1/n)$, and in particular*

$$\sqrt{n}(\widehat{\text{HR-MMD}}_u^2 - \text{HR-MMD}^2) \xrightarrow{d} \mathcal{N}(0, C_1^2 \sigma_{H_1}^2),$$

*where $Y_i = \mathcal{G}_{\ell,\phi}(\mathcal{B}_\epsilon[X_i'']) \sim \mathbb{Q}$, $X_i'' \sim \mathbb{P}$, $\sigma_{H_1}^2 = 4(\mathbb{E}_Z[(\mathbb{E}_{Z'}h(Z, Z'))^2] - [(\mathbb{E}_{Z,Z'}h(Z, Z'))^2])$, $h(Z, Z') = k_\omega(X, X') + k_\omega(Y, Y') - k_\omega(X, Y') - k_\omega(X', Y)$, $Z := (X, Y)$, $X \sim \mathbb{P}$ and $X''$ are independent and $C_1 < +\infty$ is a constant for a given $\omega$.*

*Proof.* Without loss of generality, let $Z$ be a random variable on a probability space $(\Omega^Z, \mathcal{A}^Z, \mathbb{Q}^Z)$. We will first prove that $\{Z\}_{i=1}^{+\infty}$ is an absolutely regular process. According to Eq.(10), we have

$$\beta^Z(\mathcal{A}_1^Z, \mathcal{A}_2^Z) = \sup_{A_1^Z,\dots,A_n^Z,B_1^Z,\dots,B_m^Z} \sum_{i=1}^{n} \sum_{j=1}^{m} |\mathbb{Q}^Z(A_i^Z \cap B_j^Z) - \mathbb{Q}^Z(A_i^Z)\mathbb{Q}^Z(B_j^Z)|, \qquad (12)$$

where $\mathcal{A}_1^Z, \mathcal{A}_2^Z$ are sub-$\sigma$-field of $\mathcal{A}^Z$ generated by $\{Z\}_{i=1}^{+\infty}$ and the supremum is taken over all partitions $A_1^Z, \ldots, A_n^Z$ and $B_1^Z, \ldots, B_m^Z$ of $\Omega$ into elements of $\mathcal{A}_1^Z$ and $\mathcal{A}_2^Z$, respectively. Since $X$ and $Y$ are independent and $Z = (X, Y)$, $\mathbb{Q}^Z(Z \in A^Z) = \mathbb{Q}^Z(X \in A^X, Y \in A) = \mathbb{P}(A^X)\mathbb{Q}(A)$. Thus, we have $\mathbb{Q}^Z(A_i^Z \cap B_j^Z) = \mathbb{P}(A_i^X \cap B_i^X)\mathbb{Q}(A_i \cap B_i)$, $\mathbb{Q}^Z(A_i^Z) = \mathbb{P}(A_i^X)\mathbb{Q}(A_i)$, and $\mathbb{Q}^Z(B_i^Z) = \mathbb{P}(B_i^X)\mathbb{Q}(B_i)$. Since $X$ and $X'$ are independent, we have $\mathbb{P}(A_i^X \cap B_i^X) = \mathbb{P}(A_i^X)\mathbb{P}(B_i^X)$, meaning that

$$\beta^Z(\mathcal{A}_1^Z, \mathcal{A}_2^Z) = \sup_{A_1^Z, \ldots, A_n^Z, B_1^Z, \ldots, B_m^Z} \sum_{i=1}^{n} \sum_{j=1}^{m} \mathbb{P}(A_i^X \cap B_i^X)|\mathbb{Q}(A_i \cap B_j) - \mathbb{Q}(A_i)\mathbb{Q}(B_j)|. \quad (13)$$

Due to the supremum, we can safely make $\mathbb{P}(A_i^X \cap B_i^X)$ be 1. Thus, we have $\beta^Z(\mathcal{A}_1^Z, \mathcal{A}_2^Z) = \beta(\mathcal{A}_1, \mathcal{A}_2)$. Namely, $\{Z\}_{i=1}^{+\infty}$ is an absolutely regular process with mixing coefficients $\{\beta_k\}_{k>0}$ satisfying $\sum_{k=1}^{+\infty} \beta_k^{\delta/(2+\delta)} < +\infty$. Based on Theorem 1 in (Denker & Keller, 1983), since $h(\cdot, \cdot) \le 2$, we know that

$$\sqrt{n}(\widehat{\text{HR-MMD}}_u^2 - \text{HR-MMD}^2) \xrightarrow{\mathbf{d}} \mathcal{N}(0, 4\sigma^2), \quad (14)$$

where

$$\sigma^2 = \underbrace{\mathbb{E}[h_1(Z_1)]^2}_{\sigma_{H_1}^2} + 2\sum_{j=1}^{+\infty} \text{cov}(h_1(Z_1), h_1(Z_j)), \quad (15)$$

$h_1(Z_j) = \mathbb{E}_{Z_i} h(Z_i, Z_j) - \theta$, and $\theta = \mathbb{E}_{Z_i, Z_j} h(Z_i, Z_j)$. Note that, due to $\mathbb{P} \ne \mathbb{Q}$, we know $\sigma > 0$; due to the absolute regularity, $\sigma < +\infty$. Since the possible dependence between $Z_1$ and $Z_j$ are caused by $Y_1$ and $Y_j$, we will calculate the second term in the right side of Eq.(15) in the following. First, we introduce two notations for the convenience.

$$\mathbb{E}_X^{(i)} = \mathbb{E}_X[k_\omega(X_i, X) - k_\omega(Y_i, X)], \quad (16)$$

$$\mathbb{E}_Y^{(i)} = \mathbb{E}_Y[k_\omega(X_i, Y) - k_\omega(Y_i, Y)]. \quad (17)$$

Thus, we know

$$h_1(Z_1) = \underbrace{\mathbb{E}_X^{(1)} + \mathbb{E}_Y^{(1)}}_{\tilde{h}_1(Z_1)} - \theta, \ h_1(Z_j) = \underbrace{\mathbb{E}_X^{(j)} + \mathbb{E}_Y^{(j)}}_{\tilde{h}_1(Z_j)} - \theta, \ \theta = \mathbb{E}_{Z_1}[\mathbb{E}_X^{(1)} + \mathbb{E}_Y^{(1)}], \quad (18)$$

and

$$\theta^2 = \mathbb{E}_{Z_1}[\mathbb{E}_X^{(1)} + \mathbb{E}_Y^{(1)}]\mathbb{E}_{Z_j}[\mathbb{E}_X^{(j)} + \mathbb{E}_Y^{(j)}] = \left(\mathbb{E}_{Z_1}[\mathbb{E}_X^{(1)}] + \mathbb{E}_{Z_1}[\mathbb{E}_Y^{(1)}]\right)\left(\mathbb{E}_{Z_j}[\mathbb{E}_X^{(j)}] + \mathbb{E}_{Z_j}[\mathbb{E}_Y^{(j)}]\right). \quad (19)$$

Then, we can compute the $\text{cov}(h_1(Z_1), h_1(Z_j))$.

$$\begin{aligned}
\text{cov}(h_1(Z_1), h_1(Z_j)) &= \mathbb{E}_{Z_1, Z_j}[(\tilde{h}_1(Z_1) - \theta)(\tilde{h}_1(Z_j) - \theta)] \\
&= \mathbb{E}_{Z_1, Z_j}[\tilde{h}_1(Z_1)\tilde{h}_1(Z_j) - \theta\tilde{h}_1(Z_j) - \theta\tilde{h}_1(Z_1) + \theta^2] \\
&= \mathbb{E}_{Z_1, Z_j}[\tilde{h}_1(Z_1)\tilde{h}_1(Z_j)] - \theta^2 \\
&= \mathbb{E}_{Z_1, Z_j}[(\mathbb{E}_X^{(1)} + \mathbb{E}_Y^{(1)})(\mathbb{E}_X^{(j)} + \mathbb{E}_Y^{(j)})] - \theta^2 \\
&= \mathbb{E}_{Z_1, Z_j}[\mathbb{E}_X^{(1)}\mathbb{E}_X^{(j)} + \mathbb{E}_X^{(1)}\mathbb{E}_Y^{(j)} + \mathbb{E}_Y^{(1)}\mathbb{E}_X^{(j)} + \mathbb{E}_Y^{(1)}\mathbb{E}_Y^{(j)}] - \theta^2 \\
&= \mathbb{E}_{Z_1}[\mathbb{E}_X^{(1)}]\mathbb{E}_{Z_j}[\mathbb{E}_X^{(j)}] + \mathbb{E}_{Z_1}[\mathbb{E}_X^{(1)}]\mathbb{E}_{Z_j}[\mathbb{E}_Y^{(j)}] + \\
&\quad \mathbb{E}_{Z_1}[\mathbb{E}_Y^{(1)}]\mathbb{E}_{Z_j}[\mathbb{E}_X^{(j)}] + \mathbb{E}_{Z_1, Z_j}[\mathbb{E}_Y^{(1)}\mathbb{E}_Y^{(j)}] - \theta^2.
\end{aligned} \quad (20)$$

Substituting Eq.(19) into Eq.(20), we have

$$\text{cov}(h_1(Z_1), h_1(Z_j)) = \mathbb{E}_{Z_1, Z_j}[\mathbb{E}_Y^{(1)}\mathbb{E}_Y^{(j)}] - \mathbb{E}_{Z_1}[\mathbb{E}_Y^{(1)}]\mathbb{E}_{Z_j}[\mathbb{E}_Y^{(j)}]. \quad (21)$$

Then, substituting Eq.(17) into Eq.(21), we have

$$
\begin{aligned}
\mathrm{cov}(h_1(Z_1), h_1(Z_j)) =& \mathbb{E}_{Y_1, Y_j}\big[\mathbb{E}_Y\mathbb{E}_Y[k_\omega(Y_1, Y)k_\omega(Y_j, Y)]\big] - \mathbb{E}_{Y_1}\big[\mathbb{E}_Y[k_\omega(Y_1, Y)]\big]\mathbb{E}_{Y_j}\big[\mathbb{E}_Y[k_\omega(Y_j, Y)]\big] \\
=& \mathbb{E}_Y\mathbb{E}_Y\big[\mathbb{E}_{Y_1, Y_j}[k_\omega(Y_1, Y)k_\omega(Y_j, Y)] - \mathbb{E}_{Y_1}[k_\omega(Y_1, Y)]\mathbb{E}_{Y_j}[k_\omega(Y_j, Y)]\big].
\end{aligned}
\tag{22}
$$

Since $k_\omega(\cdot, \cdot) \leq 1$, according to Lemma 1 in (Yoshihara, 1976), we have $\mathrm{cov}(h_1(Z_1), h_1(Z_j)) < 4\beta_j^{\delta/(2+\delta)}$. Because $\sum_{k=1}^{+\infty} \beta_k^{\delta/(2+\delta)} < +\infty$, we know, $\forall \epsilon' \in (0, 1)$, there exists an $N$ such that $\sum_{k=N+1}^{+\infty} \beta_k^{\delta/(2+\delta)} < \epsilon'$. Hence

$$
\begin{aligned}
\sum_{j=1}^{+\infty} \mathrm{cov}(h_1(Z_1), h_1(Z_j)) = \sum_{j=1}^{N} \mathbb{E}_Y\mathbb{E}_Y\big[&\mathbb{E}_{Y_1, Y_j}[k_\omega(Y_1, Y)k_\omega(Y_j, Y)] \\
&- \mathbb{E}_{Y_1}[k_\omega(Y_1, Y)]\mathbb{E}_{Y_j}[k_\omega(Y_j, Y)]\big] + c'.
\end{aligned}
\tag{23}
$$

where $c'$ is a small constant. Without loss of generality, we assume the small constant $c'$ is smaller than $\mathbb{E}[h_1(Z_1)]^2$. Thus, there exists a constant $C_1^2 - 1$ such that $2\sum_{j=1}^{+\infty} \mathrm{cov}(h_1(Z_1), h_1(Z_j)) = (C_1^2 - 1)\mathbb{E}[h_1(Z_1)]^2$. Namely, $\sigma^2 = C_1^2\sigma_{H_1}^2$, which completes the proof. $\square$

## D    DISCUSSION ON ADDITIONAL SCENARIOS

As outlined in the main paper, our primary objective is to evaluate whether $S_Y$ contains elements from the training set $S_T$, under Assumption IV presented in Section 3 that $S_Y$ consists solely of in-distribution samples. In this section, we extend our analysis to encompass scenarios where the dataset $S_Y$ may include *out-of-distribution* (OOD) samples.

### D.1    ASSUMPTION OF IN-DISTRIBUTION SAMPLES

Initially, our analysis operates under the assumption that $S_Y$ does not contain any OOD samples. This simplification enables the direct application of our proposed HR-MMD test to ascertain whether $S_Y$ contains elements from $S_T$, thereby detecting potential privacy breaches through statistical membership inference. Under this assumption, traditional two-sample tests (TST) are unnecessary, as the focus remains exclusively on membership verification within a consistent data distribution. There is also a common assumption in the field of membership inference (Hayes et al., 2017; Pyrgelis et al., 2017; Nasr et al., 2018; Rahman et al., 2018; Salem et al., 2018; Yeom et al., 2018b; Truex et al., 2018; Jia et al., 2019; Sablayrolles et al., 2019; Song & Marn, 2020; Leino & Fredrikson, 2020b; Rahimian et al., 2020; Song & Mittal, 2021; Choquette-Choo et al., 2021; Li et al., 2021; Liu et al., 2022; Ye et al., 2022; Carlini et al., 2022).

### D.2    INCLUSION OF OUT-OF-DISTRIBUTION SAMPLES

Previous MIA methods have exclusively conducted experiments on samples from the same distribution to ascertain whether a sample is a training member. However, in practical applications, $S_Y$ may inadvertently include OOD samples. To address this more complex scenario, we propose a two-step approach:

- **Detection of OOD Samples:** When there is a possibility that $S_Y$ contains OOD samples, the initial step involves employing traditional two-sample tests (TST) to evaluate whether $S_Y$ deviates from the distribution of $S_X$. This preliminary detection is crucial to ensure the integrity of subsequent membership inference analyses.

- **Membership Inference on In-Distribution Samples:** If the TST confirms that $S_Y$ does not include OOD samples, we proceed to apply our HR-MMD test to determine whether $S_Y$ contains any elements from $S_T$. This sequential approach ensures that membership inference is conducted solely on in-distribution data, thereby mitigating the risk of adversarial outcomes arising from the presence of OOD samples.

Table 2: Misclassification Rate of MIAs on Perturbed OOD Samples.

| Testing samples | Loss-b | Entropy-b | Attack R-b | LiRA-b |
|---|---|---|---|---|
| CIFAR-10 | | | | |
| Perturbed non-training set | 100% | 100% | 100% | 100% |
| CIFAR-100 | | | | |
| Perturbed non-training set | 100% | 100% | 100% | 100% |
| SVHN | | | | |
| Perturbed non-training set | 100% | 100% | 100% | 100% |
| CINIC-10 | | | | |
| Perturbed non-training set | 100% | 100% | 100% | 100% |

Our two-step approach is necessary in real-world scenarios where OOD samples may be inadvertently included. This necessity arises primarily because membership inference cannot reliably perform on out-of-distribution samples.

It is important to note that not all out-of-distribution samples are more likely to be classified as non-members. We leveraged a classic method from the adversarial attack domain, *Projected Gradient Descent* (PGD) (Madry et al., 2017), which aims to add perturbations to natural data, thereby increasing the loss of newly generated samples on the target classifier. By reversing the objective function of PGD, we added perturbations to non-training data to decrease their loss on the target classifier. Following the methodology of Madry et al. (2017), we used the $\ell_\infty$ norm to constrain the perturbations, ensuring they do not exceed $8/255$ for each pixel.

We performed four Membership Inference Attacks (MIAs), including Loss (Yeom et al., 2018b), Entropy (Salem et al., 2019a), Attack R (Ye et al., 2022), and Likelihood Ratio Attack (LiRA) (Carlini et al., 2022), on the generated out-of-distribution non-members, with the MIA threshold set to achieve a 10% False Positive Rate (FPR) on natural in-distribution data. As illustrated in Table 2, the deliberately generated out-of-distribution non-members successfully deceived the MIAs into classifying them as members, achieving a success rate of up to 100%.

These results demonstrate the vulnerability of existing MIAs when faced with out-of-distribution non-members, highlighting the necessity of our proposed two-step approach to enhance the robustness of membership inference techniques. By doing so, we can ensure that our HR-MMD test operates under the correct assumptions, thereby maintaining the integrity and effectiveness of privacy evaluations in diverse and potentially adversarial environments.

# E DISCUSSION ON THE USE OF HIGH-LEVEL REPRESENTATIONS

Unlike traditional MIAs that rely on specific metrics such as loss values, our approach leverages high-level representations to enhance the discriminative power between training and non-training sets. Additionally, previous statistics like loss, entropy, and likelihood ratios can be seamlessly integrated into our framework. This flexibility is particularly advantageous in real-world scenarios where certain information may be unavailable or deliberately obfuscated.

For instance, in settings where true labels are inaccessible, making loss-based metrics infeasible, our framework can utilize alternative high-level representations such as confidence scores. Furthermore, adversarial strategies like MemGuard (Jia et al., 2019) introduce perturbations to confidence scores to thwart membership inference, thereby reducing the effectiveness of loss-based and confidence score-based methods. In such cases, our HR-MMD test can employ other high-level representations, such as those from the penultimate layer of a neural network, to maintain robust membership inference capabilities. By accommodating a diverse array of high-level representations, HR-MMD extends the evaluative toolkit for privacy breach detection, ensuring adaptability and resilience across varying operational conditions.

Our approach distinctly differs from previous metric-based MIAs (Yeom et al., 2018a; Salem et al., 2018; Song & Mittal, 2021) in several key aspects. It does not rely on predefined functions and preset thresholds. Instead, we propose a comprehensive and reliable hypothesis testing framework coupled with a high-level representation-based deep kernel. Crucially, our method does not merely select an existing kernel; it innovatively learns one by optimizing its parameters during training. To the best of our knowledge, we are the first to introduce such a non-parametric two-sample test into the field of membership privacy.

Our ablation study, presented in Figure 7, demonstrates that using a non-deep kernel-based method with high-level representations (HR+MMD-G) does not achieve the same effectiveness as our proposed HR-MMD approach. This underscores the superiority of our deep kernel learning methodology in enhancing membership inference performance.

While many studies have highlighted the utility of the penultimate layer of Deep Neural Networks (DNNs) for tasks such as out-of-Distribution (OOD) detection and adversarial detection (Lee et al., 2018), our high-level representation-based MMD test offers significant implications beyond membership inference. Specifically, the objective of OOD detection is to verify whether input testing data is out of distribution, and the goal of adversarial detection is to identify whether input testing data comprises adversarial samples. Our work reveals that even when input testing samples are neither out of distribution nor adversarial, differences in the high-level representations of members and non-members relative to the same model still exist. This insight has been largely overlooked by previous detection research.

As membership privacy remains one of the most established methods for quantifying privacy risks associated with machine learning models, our contribution offers valuable insights into an alternative scenario for privacy breach detection. By demonstrating that high-level representations can effectively distinguish between members and non-members, our work provides a robust foundation for enhancing privacy protection mechanisms in various machine learning applications.

**Algorithm 2** Using the HR-MMD test in our experiments.

**Input:** $S_{non}$, $S_Y$, $\phi$, various hyperparameters used below;

$\omega \leftarrow \omega_0$; $\lambda \leftarrow 10^{-8}$; $\beta \leftarrow 0.5$; $\alpha \leftarrow 0.05$;      *▷ Initialize parameters*

Split the data as $S_{non} = S_X^{tr} \cup S_{non}^r$ and $S_Y = S_Y^{tr} \cup S_Y^{te}$; *▷ Divide data into training and testing sets*

*# Phase 1: training the kernel parameters $\omega$ on $S_X^{tr}$ and $S_Y^{tr}$*

**for** $T = 1, 2, \ldots, T_{max}$ **do**

    $S'_X \leftarrow$ minibatch from $S_X^{tr}$;      *▷ Sample a minibatch from $S_X^{tr}$*

    $S'_Y \leftarrow$ minibatch from $S_Y^{tr}$;      *▷ Sample a minibatch from $S_Y^{tr}$*

    $k_\omega \leftarrow$ kernel function with parameters $\omega$ using Eq.(5); *▷ Define kernel function based on current $\omega$*

    $M(\omega) \leftarrow \widehat{\text{HR-MMD}}_u^2(S'_X, S'_Y; k_\omega)$ using Eq.(6); *▷ Compute the HR-MMD statistic for minibatch*

    $V_\lambda(\omega) \leftarrow \hat{\sigma}_{H_1,\lambda}^2(S'_X, S'_Y; k_\omega)$ using Eq.(9);      *▷ Estimate variance under $H_1$ with regularization*

    $\hat{J}_\lambda(\omega) \leftarrow M(\omega)/\sqrt{V_\lambda(\omega)}$ using Eq.(8);      *▷ Compute objective function for optimization*

    $\omega \leftarrow \omega + \eta\nabla_{\text{Adam}}\hat{J}_\lambda(\omega)$;      *▷ Update kernel parameters via gradient ascent to maximize $\hat{J}_\lambda(\omega)$*

**end for**

*# Phase 2: testing with $k_\omega$ on $S_{non}^r$ and $S_Y^{te}$ multiple times*

**for** $i = 1, 2, \ldots, n_{eval}$ **do**

    $S_X^{te} \leftarrow$ random selection from $S_{non}^r$;      *▷ Randomly select test samples from $S_{non}^r$*

    $est \leftarrow \widehat{\text{HR-MMD}}_u^2(S_X^{te}, S_Y^{te}; k_\omega)$      *▷ Compute the HR-MMD statistic on test data*

    **for** $j = 1, 2, \ldots, n_{perm}$ **do**

        Shuffle $S_X^{te} \cup S_Y^{te}$ into $X$ and $Y$      *▷ Shuffle data to create permutation samples*

        $perm_j \leftarrow \widehat{\text{HR-MMD}}_u^2(X, Y; k_\omega)$      *▷ Compute HR-MMD statistic for each permutation*

    **end for**

    $p^i \leftarrow \frac{1}{n_{perm}}\sum_{j=1}^{n_{perm}} \mathbb{1}(perm_j \geq est)$      *▷ Compute p-value for each evaluation*

**end for**

**Output:** rejection rate: $\frac{1}{n_{eval}}\sum_{i=1}^{n_{eval}} \mathbb{1}(p^i \leq \alpha)$      *▷ Return average rejection rate over evaluations*

## F THE VISUALIZATION OF HIGH-LEVEL REPRESENTATION

In this section, we vizualize the high-level representation of the penultimate layer of a well-tained DNN using t-SNE (Maaten & Hinton, 2008) in Figure 4 and Figure 5. These figures show that the high-level representation of members is different from that of non-members.

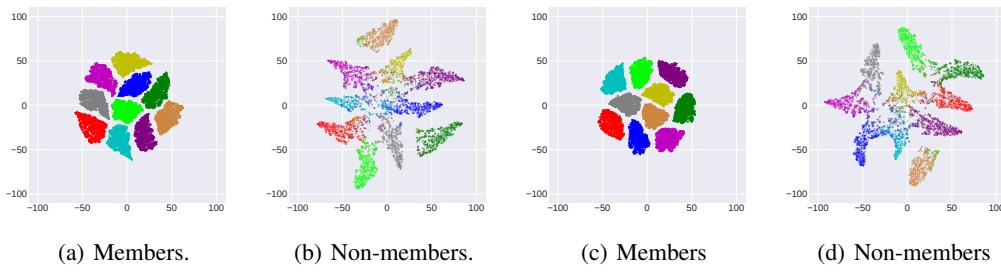

    (a) Members.          (b) Non-members.          (c) Members          (d) Non-members

Figure 4: Visualization of high-level representations using t-SNE (Maaten & Hinton, 2008). Subfigures (a) and (b) depict the confidence scores, while subfigures (c) and (d) illustrate the high-level representations of the penultimate layers in ResNet-18. Different colors represent different semantics (classes in *CIFAR-10*). It is apparent that the high-level representations of members differ from those of non-members. This distinction can help in differentiating between members and non-members.

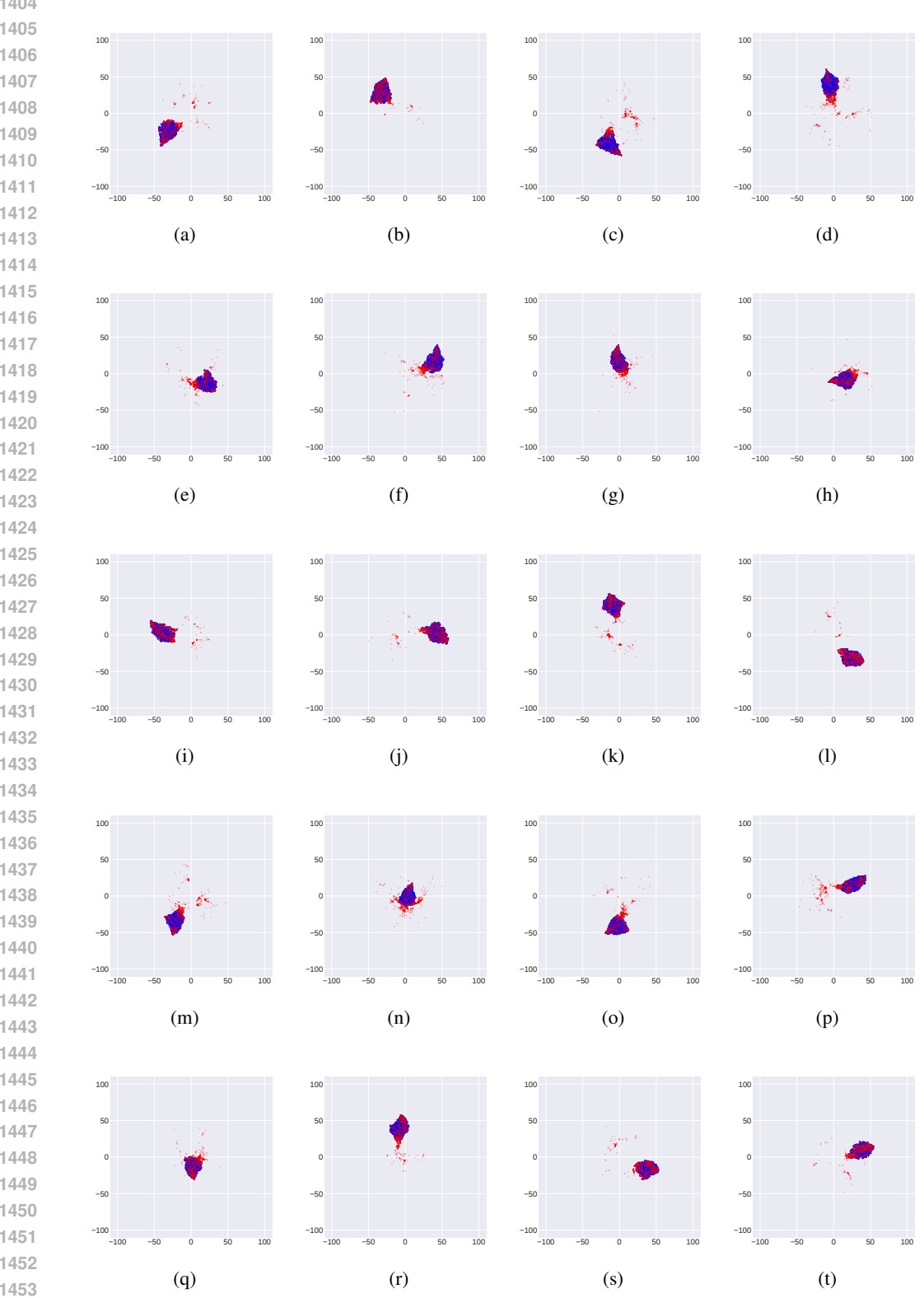

Figure 5: Visualization of high-level representation using t-SNE (Maaten & Hinton, 2008). Subfigures (a-j) depict the confidence scores, while subfigures (k-t) illustrate the high-level representations of the penultimate layers in ResNet-18. Different subfigures represent different classes in the *CIFAR-10*. In each subfigure, red dots represent the high-level representation of non-members and blue dots represent the high-level representation of members. It is apparent that, within each class, the high-level representation of members is rather different from the one of non-members.

# G    EXPERIMENTAL DETAILS

## G.1    DATASETS

**CIFAR-10 (Krizhevsky et al., 2009)**    CIFAR-10 is a widely-used dataset consisting of 60,000 color images, each with a resolution of $32 \times 32 \times 3$. The images are evenly distributed across 10 distinct classes, which represent various objects such as airplanes, automobiles, birds, and more. In our preprocessing, we follow the standard procedure recommended in the PyTorch ResNet documentation[2], which involves normalizing the pixel values so that the dataset has a mean of zero and a standard deviation of one. This normalization aids in stabilizing the training process of deep neural networks and ensures faster convergence.

**CIFAR-100 (Krizhevsky et al., 2009)**    CIFAR-100 shares the same structure and resolution as CIFAR-10, with 60,000 color images of $32 \times 32 \times 3$. However, it poses a more complex classification challenge, as it is divided into 100 classes, each containing 600 images. Similar to CIFAR-10, we preprocess CIFAR-100 using mean subtraction and standardization, normalizing the pixel values to achieve a mean of zero and a standard deviation of one. This ensures consistency across the datasets and helps improve model training performance, particularly for complex models like ResNet.

**SVHN (Netzer et al., 2011)**    The Street View House Numbers (SVHN) dataset consists of over 600,000 digit images captured from real-world street views. Each image has a resolution of $32 \times 32 \times 3$ and contains a single digit (0-9). SVHN is particularly challenging due to variations in lighting, backgrounds, and digit orientations. For preprocessing, we apply normalization similar to CIFAR-10 and CIFAR-100, scaling the pixel values to have a mean of zero and a standard deviation of one.

**CINIC-10 (Darlow et al., 2018)**    CINIC-10 is an extension of the CIFAR dataset, designed to bridge the gap between CIFAR-10 and more complex datasets like ImageNet. It contains 270,000 images with a resolution of $32 \times 32 \times 3$, with classes and data distribution similar to CIFAR-10. CINIC-10 combines CIFAR-10 images with a subset of ImageNet images, providing a larger and more diverse dataset. We apply the same preprocessing steps as in CIFAR-10 and CIFAR-100, ensuring pixel normalization to improve training stability and ensure compatibility with deep learning models.

**Purchase-100 (Shokri et al., 2017b)**    Purchase-100 is a comprehensive dataset comprising 100,000 samples of purchase records from an online retail platform. Each sample is represented by a 600-dimensional binary vector, where each dimension indicates the presence or absence of a specific product in a transaction. The dataset is categorized into 100 distinct classes, corresponding to different product categories, with 80,000 samples allocated for training and 20,000 samples designated for testing. This dataset is instrumental for tasks such as multi-label classification, purchase prediction, and customer behavior analysis.

**IMDB (Maas et al., 2011)**    The IMDB dataset consists of 50,000 movie reviews labeled as positive or negative sentiment. Each review is a sequence of words with varying lengths, making it a suitable benchmark for natural language processing tasks such as sentiment analysis. We preprocess the text data by tokenizing the reviews, converting words to lower case, removing stop words, and applying techniques like padding or truncation to ensure uniform input length. Additionally, we employ word embeddings to transform textual data into dense vector representations, facilitating the learning process of deep neural networks.

## G.2    MODEL STRUCTURE

**ResNet-18 (He et al., 2016)**    ResNet-18 is a deep residual neural network consisting of 18 layers, designed to address the vanishing gradient problem in deep networks through the introduction of residual connections. These connections allow gradients to flow more easily during backpropagation, enabling the training of deeper architectures without degradation in performance. ResNet-18 comprises multiple residual blocks, each containing convolutional layers, batch normalization, and ReLU activation functions. The architecture concludes with a fully connected layer for classification. In our

---

[2]https://pytorch.org/hub/pytorch_vision_resnet/

experiments, we utilize ResNet-18 due to its balance between computational efficiency and model performance, making it suitable for a variety of image classification tasks.

**Wide-ResNet-34** (Zagoruyko & Komodakis, 2016)   Wide-ResNet-34 is an extension of the original ResNet architecture, characterized by increased width (i.e., more convolutional filters) instead of deeper layers. This modification allows the network to capture more complex features without the computational overhead associated with very deep networks. Wide-ResNet-34 consists of 34 layers with widened residual blocks, incorporating more filters in each convolutional layer to enhance the model's capacity. Additionally, it employs techniques such as dropout and batch normalization to improve generalization and training stability. In our study, Wide-ResNet-34 is chosen for its superior performance on more challenging datasets, where increased model capacity can lead to better feature representation and classification accuracy.

### G.3   BASELINES

We compare our methods against six state-of-the-art (SOTA) two-sample tests and two of our defined baselines, which were specifically chosen for their capability to maintain an ideal type I error rate.

These include: 1) the MMD-G test (Grosse et al., 2017), 2) the MMD-O test (Sutherland et al., 2017), 3) the Mean Embedding (ME) test (Jitkrittum et al., 2016), 4) the Smooth Characteristic Functions (SCF) test (Chwialkowski et al., 2015), 5) the Classifier Two-Sample Test (C2ST) (Lopez-Paz & Oquab, 2017), and 6) the MMD-D test (Liu et al., 2020b). Additionally, we include two of our defined baselines, LR1-MMD and HR2-MMD, which use low-level representations as described in Appendix G.5.

### G.4   EXPERIMENTAL SETUP

We verify our methods with the model structures ResNet-18 (He et al., 2016) and Wide-ResNet-34 (WRN-34) (Zagoruyko & Komodakis, 2016), on the four benchmark image datasets (**CIFAR-10** (Krizhevsky et al., 2009), **CIFAR-100** (Krizhevsky et al., 2009), **SVHN** (Netzer et al., 2011), **CINIC-10** (Darlow et al., 2018)). We adopt a training setup for the classifier $\hat{f}$ that is consistent with prior studies. All networks are trained for 100 epochs using SGD with 0.9 momentum. The initial learning rate is 0.1 (0.01 for SVHN), and is divided by 10 at epoch 60 and 90, respectively. The weight decay is 0.0002 (0.0035 for SVHN). We also verify our methods on a tabular dataset **Purchase-100** (Shokri et al., 2017b). Following the training details from (Ye et al., 2022), we use a 4-layer MLP model with layer units [512, 256, 128, 64], employing the SGD optimizer algorithm. Additionally, we verify our methods on a text dataset **IMDB** (Maas et al., 2011) on a simple 2-layer LSTM model with hidden sizes [128, 128].

We implemented all methods on Python 3.7 (Pytorch 1.7.1) with an NVIDIA GeForce RTX 3090 GPU with AMD Ryzen Threadripper 3960X 24 Core Processor. The CIFAR-10, the SVHN, the CIFAR-100 and CINIC-10 dataset can be downloaded via Pytorch.

### G.5   CHOICES OF HR-MMD TEST SERIES

Our approach leverages high-level representations to enhance the discriminative power between training and non-training sets. Additionally, statistics such as loss value, entropy, and likelihood ratio can be seamlessly integrated into our framework. This flexibility is particularly advantageous in real-world scenarios where certain information may be unavailable or deliberately obfuscated.

For instance, in settings where true labels are inaccessible, rendering loss-based metrics infeasible, our framework can utilize alternative high-level representations such as confidence scores. Furthermore, adversarial strategies like MemGuard (Jia et al., 2019) introduce perturbations to confidence scores to thwart membership inference, thereby reducing the effectiveness of loss-based and confidence score-based methods. In such cases, our HR-MMD test can employ other high-level representations, such as those from the penultimate layer of a neural network, to maintain robust membership inference capabilities. By accommodating a diverse array of high-level representations, HR-MMD extends the evaluative toolkit for privacy breach detection, ensuring adaptability and resilience across varying operational conditions.

Our HR-MMD test framework encompasses a wide range of series. In the experimental section, we employed the following six HR-MMD variants:

- **HR1-MMD**: In the deep kernel defined in Eq.(5), $\phi_p$ is used to extract high-level representations from the last layer of the target model.

- **HR2-MMD**: In the deep kernel defined in Eq.(5), $\phi_p$ is used to extract high-level representations from the second-to-last layer of the target model.

- **CS-MMD**: In the deep kernel defined in Eq.(5), $\phi_p$ is used to the confidence scores (derived from high-level representations).

- **L-MMD**: In the deep kernel defined in Eq.(5), $\phi_p$ is used to calculate the loss values (derived from high-level representations).

- **E-MMD**: In the deep kernel defined in Eq.(5), $\phi_p$ is used to calculate the entropies (derived from high-level representations).

- **LI-MMD**: In the deep kernel defined in Eq.(5), $\phi_p$ is used to calculate the likelihood ratio (derived from high-level representations).

Additionally, we incorporate our deep kernel approach using low-level representations into the baseline methods:

- **LR1-MMD**: In the deep kernel defined in Eq.(5), $\phi_p$ is used to extract low-level representations from the first layer of the target model.

- **LR2-MMD**: In the deep kernel defined in Eq.(5), $\phi_p$ is used to extract low-level representations from the second layer of the target model.

### G.6 IMPLEMENTATION DETAILS OF THE COMPARISON WITH TRADITIONAL TWO-SAMPLE TESTS

In our experimental framework, the training set size of $\hat{f}$, denoted by $S_T$, is set to $25,000$ for CIFAR-10, CIFAR-100 and SVHN, and $90,000$ for CINIC-10. Additionally, we assume the evaluator has a non-training set $S_{non}$, each containing $5,000$ images from the respective datasets. $S_X^{tr}$ and $S_X^{te}$ are selected from $S_{non}$. We fix the size of the input test dataset $S_Y$ to $1,000$ in our main experiments. When comparing our method with the baseline methods in terms of test power, the samples $S_Y$ are randomly selected from the training subsets ($S_T$) of the above datasets. In this context, $S_T \cap S_X = \emptyset$, $S_T \cap S_Y = S_Y$, and $S_X \cap S_Y = \emptyset$. When comparing our method with the baselines in terms of type I error, the samples $S_Y$ are randomly selected from the testing subsets of the four datasets. In this context, the sets are disjoint such that $S_T \cap S_X = \emptyset$, $S_T \cap S_Y = \emptyset$, and $S_X \cap S_Y = \emptyset$.

In Algorithm 1, for the input test data set $S_Y$ (size $1,000$) and the non-training set $S_X$ (size $1,000$), we select subsets containing $500$ images each for $S_X^{tr}$ and $S_Y^{tr}$ and train our deep kernel on these subsets. The evaluation is conducted once on $S_X^{te}$ and $S_Y^{te}$ with the set size of $500$, each disjoint from the training sets $S_X^{tr}$ and $S_Y^{tr}$, of the remaining data. We compare the p-value obtained from Algorithm 1 with the significance level $\alpha$ to determine whether to reject $H_0$.

Recall that, in our experiments, to obtain a more reliable assessment, the evaluation (Phase 2 in Algorithm 1) is repeated 100 times on a fixed $S_Y^{te}$ (size 500) and 100 different random subsets $S_X^{te}$ (size 500). Here we assume the evaluator has a non training set $S_{non}$ comprising $5,000$ samples. Out of these, $500$ samples are selected for $S_X^{tr}$, and the remaining $4,500$ samples are used to generate 100 different random subsets $S_X^{te}$. Algorithm 2 describes the complete flow of this process. When the inputted samples $S_Y$ are non-members, the rejection rate outputted is the type I error. When the inputted samples $S_Y$ are members, the rejection rate outputted is the test power. We repeat this process 10 times, and report the mean test power of each test. The learning rate of our HR-MMD test and all baselines is 0.02.

It is noteworthy that the size of $S_Y$ in our main experiments is significantly smaller compared to the ones used in previous two-sample tests studies. For instance, the experiments in (Liu et al., 2020b) use an $S_Y$ of size $2,000$. Our results, as presented in Figure 12 and Figure 13, demonstrate the effectiveness of our HR-MMD test even with smaller sample datasets.

### G.7 Implementation Details of the Comparison with Traditional MIAs

In our experiments comparing the false positive rates and true positive rates between prior Membership Inference Attacks (MIAs) and our proposed HR-MMD test, we conducted comparisons with four typical MIA methods: Loss Attack (Yeom et al., 2018b), Entropy Attack (Salem et al., 2019a), Attack R (Ye et al., 2022), and the Likelihood Ratio Attack (LiRA) (Carlini et al., 2022).

We focused on predicting membership risks for datasets where only a small proportion (10%) of the records are training members. For the LiRA attack, we selected its offline attack version to save computational cost. For model training, we followed the process in (Carlini et al., 2022), training a ResNet-18 model to 60% accuracy on 25,000 CIFAR-10 examples, denoted as $S_T$, which were randomly selected from the original CIFAR-10 training set. We assumed that the adversary (evaluator) has access to a non-training set $S_{non}$ containing the remaining 25,000 examples from the original CIFAR-10 training set, which were used to train 64 shadow models.

For the baseline MIAs, which output decisions for individual data points, we aggregated their outputs to the dataset level by computing average statistics such as loss, entropy, or likelihood ratio over the input test dataset $S_Y$, and setting thresholds to make set-level decisions, similar to the approach used in set-based attacks like (Kandpal et al., 2023; Maini et al., 2024). In the evaluation, when assessing the false positive rate, we selected $S_Y$ of size 1,000 from the original CIFAR-10 testing set (10,000 samples), containing no training members. When assessing the true positive rate, we selected $S_Y$ of size 1,000, where 90% of the samples were from the CIFAR-10 testing set and 10% were from the training set $S_T$, thereby containing training members. We repeated this full process 1,000 times for datasets with members and 1,000 times for datasets without members, and reported the average statistics. When the average statistics such as loss for one dataset $S_Y$ was no greater than the threshold, we considered that $S_Y$ contained training members. When the average statistics was greater than the threshold, we considered that $S_Y$ contained no training members.

For our HR-MMD test methods, we also assumed that the evaluator has a non-training set $S_{non}$ comprising 5,000 samples. Out of these, 500 samples were selected for $S_X^{tr}$, and the remaining 4,500 samples were used to randomly generate 100 different subsets $S_X^{te}$. We divided the input test data $S_Y$ (size 1,000) into $S_Y^{tr}$ and $S_Y^{te}$, each of size 500. We trained our deep kernel once on $S_X^{tr}$ and $S_Y^{tr}$, and then evaluated on $S_Y^{te}$ and 100 different $S_X^{te}$ sets. We repeated this process 1,000 times for datasets with members and 1,000 times for datasets without members, and reported the rejection rate of each dataset. When the rejection rate for one dataset $S_Y$ was no greater than 0.065, we considered that $S_Y$ contained no training members. When the rejection rate was greater than 0.065, we considered that $S_Y$ contained training members.

In our experiments, the true positive rate (TPR) is defined as the proportion of datasets correctly identified as containing training members, and the false positive rate (FPR) is the proportion of datasets incorrectly identified as containing training members when they do not. Experimental results show that our HR-MMD test significantly outperforms prior set-based MIAs.

## H Supplementary Experimental Results

In this section, we present very comprehensive supplementary experimental results to detail the performance of our method HR-MMD alongside various baselines. The results comprehensively demonstrate the advantages of our method over the baselines. It should be noted that LR1-MMD and LR2-MMD are our deep kernels that utilize the low-level representations from the first and second layers of the target model, respectively, while HR1-MMD and HR2-MMD are our deep kernels that employ the high-level representations from the last and second-to-last layers of the target model.

### H.1 Comparison with Traditional MIAs.

Figure 6 (a) assesses cases where the testing subjects are non-training members, targeting an ideal false positive rate of 0%. Figure 6 (b) assesses cases where the testing subjects are a mix of 10% training members and 90% non-training members, aiming for an ideal true positive rate of 100%. Thresholds for prior methods were set for different false positive rates (1% for group (a) (e.g., Loss-a); 10% for group (b) (e.g., Loss-b)). A critical limitation of these attacks is that striving for a low false positive rate often results in a low true positive rate. This is contrasted with our proposed HR-MMD

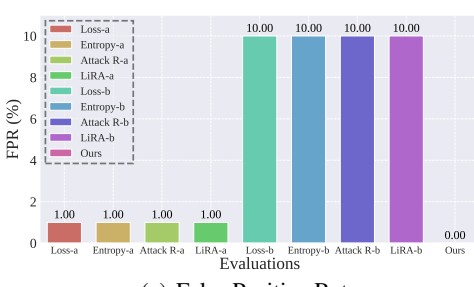 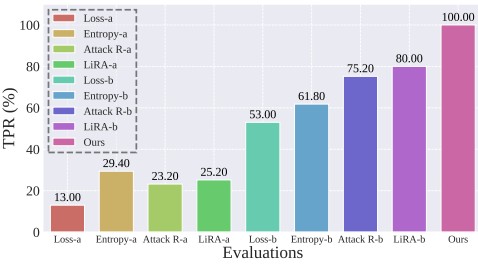

(a) False Positive Rate.      (b) True Positive Rate.

Figure 6: Comparison of false positive rates (subfigure a) and true positive rates (subfigure b) between MIAs methods and our proposed HR-MMD test. Subfigure (a) assesses where the testing subjects are non-training members, targeting an ideal false positive rate of 0%. Subfigure (b) assesses where the testing subjects are a mix of 10% training members and 90% non-training members, aiming for an ideal true positive rate of 100%. Thresholds for prior methods were set for different false positive rates (1% for group a (e.g., Loss-a); 10% for group b (e.g., Loss-b)). A critical limitation of these attacks is that striving for a low false positive rate often results in a low true positive rate. This is contrasted with our proposed HR-MMD test, which perform more effectively as illustrated in the figure.

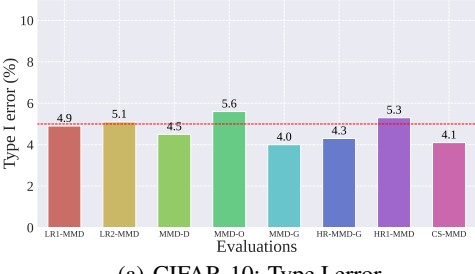 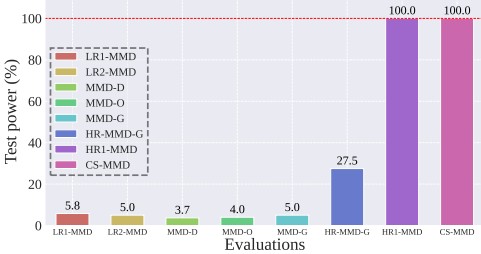

(a) CIFAR-10: Type I error      (b) CIFAR-10: Test power

Figure 7: Ablation study: the test power (%) of the HR-MMD test and MMD-based methods on ResNet-18. As seen in subfigure (a), all methods achieve an ideal type I error around the significance level of 0.05. Results in subfigure (b) further support our thesis that high-level representation are more effective in differentiating between members and non-members, compared to raw features and low-level representation (CS-MMD (or HR1-MMD) vs MMD-O; CS-MMD (or HR1-MMD) vs MMD-G; CS-MMD (or HR1-MMD) vs LR-MMD), as well as outperform learned features in other deep kernels (CS-MMD (or HR1-MMD) vs MMD-D). Additionally, our findings indicate that our deep kernel approach outperforms the non-deep kernel methods (CS-MMD (or HR1-MMD) vs HR+MMD-G). The experiments were conducted on CIFAR-10 using ResNet-18.

test, which performs more effectively as illustrated in the figure. Note that traditional MIAs are not standard hypothesis testing methods, so when comparing our methods with traditional MIAs, we use the experimental setup in Appendix G.7 to conduct an appropriate and fair comparison.

## H.2 TEST POWER ON DIFFERENT DATASETS AND MODEL STRUCTURES.

We evaluate the effectiveness of our methods across four image datasets: **CIFAR-10** (Krizhevsky et al., 2009), **CIFAR-100** (Krizhevsky et al., 2009), **SVHN** (Netzer et al., 2011), and **CINIC-10** (Darlow et al., 2018), across two different model architectures: **ResNet-18** (He et al., 2016) and **Wide-ResNet-34** (Zagoruyko & Komodakis, 2016). We also demonstrate the capacity for extending to a tabular dataset **Purchase-100** (Shokri et al., 2017b) and a text dataset **IMDB** (Maas et al., 2011). The experimental results are detailed in Figure 10, Figure 11, Figure 16 and Figure 17. The results demonstrate the effectiveness of our HR-MMD tests over the baselines across extensive datasets and model structures.

## H.3 TEST POWER ON DIFFERENT SET SIZES.

From Eq.(7), it is clear that the test power of the HR-MMD test is contingent upon the sample size, with larger $m$ yielding enhanced test power. This is consistent with other kernel-based non-parametric two-sample tests, such as C2ST and MMD-D (Lopez-Paz & Oquab, 2017; Liu et al., 2020b). In the

Table 3: The test power and type I error rates (%) of the HR-MMD test and baselines on the LLMs scenarios.

| Set size | Test Power (%) | | | Type I Error Rate (%) | | |
|---|---|---|---|---|---|---|
| | HR-T | HR-MMD | HR-MMD-G | HR-T | HR-MMD | HR-MMD-G |
| 100 | 2.2 | **18.7** | 8.6 | 1.6 | 2.3 | 2.2 |
| 200 | 1.6 | **21.7** | 12.7 | 0.5 | 3.3 | 3.1 |

realm of high-dimensional data analysis with deep kernels, the minimal sample size necessary to ensure test validity is dataset-dependent and challenging to estimate. Consequently, we conducted a comparative analysis of our HR-MMD tests against the baseline methods for various sample sizes. The outcomes in Figure 12 and Figure 13 further corroborate the effectiveness of our HR-MMD tests, which surpass the baselines for diverse sample sizes.

### H.4  TEST POWER FOR A MIXTURE OF TRAINING AND NON-TRAINING RECORDS.

In a more practical scenario, it may be the case that only part of the tested data has been breached and used for training. We analyze the test power of the HR-MMD test and the baseline methods in this case, with the proportion of members ranging from $10\%$ to $100\%$. The experimental results on ResNet-18 are presented in Figure 14 and Figure 15. These results show that the performance of our HR-MMD test is once again superior to that of the baseline methods.

### H.5  ABLATION STUDY.

Finally, to demonstrate the superior performance of the high-level representations, we conducted a comparative analysis of our HR-MMD tests against other MMD-based methods. The experimental results are detailed in Figure 7. They further support our thesis that high-level representations are more effective in differentiating between members and non-members compared to raw features and low-level representations (CS-MMD (or HR1-MMD) vs MMD-O; CS-MMD (or HR1-MMD) vs MMD-G; CS-MMD (or HR1-MMD) vs LR-MMD), as well as outperforming learned features in other deep kernels (CS-MMD (or HR1-MMD) vs MMD-D). Additionally, our findings indicate that our deep kernel approach outperforms the non-deep kernel methods (CS-MMD (or HR1-MMD) vs HR+MMD-G).

## I  DISCUSSION OF APPLICATIONS IN LARGE LANGUAGE MODEL SCENARIOS

In this section, we demonstrate the applicability of our HR-MMD test within the context of large language models (LLMs), focusing specifically on dataset inference scenarios as discussed in cutting-edge LLMs studies (Kandpal et al., 2023; Maini et al., 2024). These LLM dataset inference tasks represent a subset of the privacy breach detection problems that our method aims to address. We illustrate how our approach can be effectively integrated with the current state-of-the-art (SOTA) method proposed by Maini et al. (2024), yielding improved results and offering valuable insights into privacy breach detection in the realm of LLMs.

The dataset inference procedure outlined by Maini et al. (2024) comprises two primary components. The first involves aggregating features using Membership Inference Attacks (MIAs). Specifically, the suspect set and validation set are passed through the LLM to obtain features derived from various MIAs, resulting in a single feature vector for each sample. This aggregated feature vector is then used to train a linear regression model, which learns the importance weights for different MIA features to predict membership status. The second component performs dataset inference using statistical t-tests to determine whether the suspect set was utilized during the model's training.

Our method integrates seamlessly with the first component of Maini et al. (2024)'s procedure. We adhere strictly to their experimental setup, utilizing the weighted results from the linear regression model as our high-level representations. Specifically, we replace the feature extractor $\phi_p$ of our deep kernel in Eq.(5) with these weighted MIA features. Unlike traditional t-tests, our deep kernel, equipped with a learning process, effectively captures distributional differences, providing a more

Table 4: The test power (%) of the HR-MMD test and baselines using the *zlib_ratio* feature on different set sizes.

| Method | 100 | 200 | 300 | 400 | 500 | 600 | 700 | 800 | 900 |
|---|---|---|---|---|---|---|---|---|---|
| HR-T | 1.2 | 0.7 | 2.7 | 8.5 | 23.4 | 18.7 | 44.7 | 62.7 | 75.5 |
| HR-MMD-G | 3.5 | 4.4 | 6.3 | 8.9 | 10.5 | 12.4 | 22.9 | 32.5 | 41.1 |
| HR-MMD | **5.6** | **9.2** | **12.1** | **24.2** | **33.3** | **30.5** | **58.9** | **73.8** | **93.1** |

powerful statistical test. We then employ our Algorithm 1 to perform the HR-MMD test and compute the p-value.

Our experimental setup closely follows that of Maini et al. (2024). We conduct dataset inference experiments on subsets of the Pile dataset (Gao et al., 2020), utilizing models from the Pythia family (Biderman et al., 2023) at the 410M parameter scale. Although computational resource limitations preclude experiments with larger models, this scale suffices to demonstrate the feasibility and efficacy of our strategy. The Pile dataset comprises 50,000 records used in training the Pythia models and 2,434 records not included in the training set. Following the methodology of Maini et al. (2024), we use 1,000 of the 50,000 training records and 1,000 of the 2,434 non-training records to train the linear regression model, thereby learning the importance weights for different MIA features. We then apply the weighted results from the linear regression model to the remaining data to evaluate dataset inference performance.

Aligning with the well-established two-sample test framework (Grosse et al., 2017; Sutherland et al., 2017; Jitkrittum et al., 2016; Chwialkowski et al., 2015; Lopez-Paz & Oquab, 2017; Liu et al., 2020b), and similar to the implementation details described in Appendix G.6, we conduct multiple tests: 100 evaluations on a fixed $S_Y^{\text{te}}$ (with sizes ranging from 100 to 200) and 100 different random subsets $S_X^{\text{te}}$ (of corresponding sizes). It is noteworthy that the set sizes are relatively small due to the limited size of the known non-training set in the Pile dataset. Larger set sizes could diminish the diversity of sample pairs in multiple tests, potentially affecting test performance. Additionally, since statistical test performance generally improves with larger set sizes, conducting comparisons under this challenging small set size setting allows for a more effective distinction of test power among different methods. When the input samples $S_Y$ are non-members, the rejection rate corresponds to the type I error, whereas for members, it reflects the test power. We repeat this process 10 times and report the mean rejection rate of each test.

In this comparative study, we refer to the method employing t-tests on the weighted results as **HR-T**. We compare our **HR-MMD** test, which utilizes a deep kernel, against HR-T using the same high-level representations (weighted results from Maini et al. (2024)). Additionally, we examine the performance of using a fixed non-deep kernel (e.g., a standard Gaussian kernel) in the same setting, which we denote as the **HR-MMD-G** test.

Furthermore, we explore different high-level representations by directly using the original feature vectors (e.g., *zlib_ratio*) without employing the weighted results from Maini et al. (2024). In this setting, we do not require training a linear regression model as in Maini et al. (2024), enabling us to utilize more data for evaluating dataset inference performance. Consequently, we conduct tests with larger set sizes, performing 100 evaluations on fixed $S_Y^{\text{te}}$ (sizes ranging from 100 to 500) and 100 different random subsets $S_X^{\text{te}}$ of corresponding sizes.

Our results, presented in Table 3 and Table 4, indicate that our HR-MMD test enhances the detection of whether the suspect set was employed during model training compared to using t-tests in (Maini et al., 2024). The HR-MMD test more effectively distinguishes the distributional differences between the suspect and validation sets, leading to higher rejection rates when appropriate. This demonstrates that our method is applicable to LLM dataset inference scenarios and can augment current SOTA methods, providing substantial insights into privacy breach detection within the context of large language models.

## J    DISCUSSION OF EVALUATIONS UNDER MEMBERSHIP INFERENCE DEFENSES

In this section, we delve into the robustness of our HR-MMD test when confronted with advanced membership inference defenses. Our aim is to demonstrate that our method maintains high efficacy even when models employ sophisticated techniques to obfuscate membership information. The experimental setup aligns with the details provided in Appendix G.6, ensuring consistency and comparability across our evaluations. To rigorously assess the resilience of our HR-MMD test, we implemented three prevalent membership inference defense strategies in training ResNet-18 models on the CIFAR-10 dataset:

**Knowledge Distillation.** We utilized a teacher-student framework where the student model learns from the softened outputs (soft labels) of a pre-trained teacher model. This approach is intended to enhance generalization and obscure membership information by transferring knowledge in a manner that reduces overfitting to the training data.

**L1 Regularization.** We incorporated an L1 regularization term into the loss function during training. By penalizing the absolute values of the model weights, L1 regularization encourages sparsity in the parameters, which can mitigate overfitting and, consequently, reduce the leakage of membership information.

**Differential Privacy (DP-SGD).** We implemented the *Differentially Private Stochastic Gradient Descent* (DP-SGD) algorithm (Abadi et al., 2016), setting the privacy budget to $\varepsilon = 1$. DP-SGD introduces carefully calibrated noise to the gradients during training, providing formal privacy guarantees that limit the influence of any individual training sample on the model parameters.

The datasets used in these experiments were partitioned as per Section G.6. Specifically, the training set $S_T$ comprised 25,000 images from CIFAR-10, while the non-training set $S_{\text{non}}$ consisted of 5,000 images disjoint from $S_T$. The input test dataset $S_Y$ was a size of 1,000 for the main experiments, with samples drawn either from the training set (for test power assessments) or the testing set (for type I error evaluations). Our experimental procedure adhered to Algorithm 1, with the evaluation phase repeated 100 times to obtain reliable assessments. The rejection rates were calculated based on the proportion of times the null hypothesis $H_0$ was rejected across these repetitions.

As shown in Figure 8, The HR-MMD test consistently demonstrated superior performance compared to traditional two-sample tests across all defense mechanisms. Under knowledge distillation and L1 regularization, our method maintained high test power, effectively detecting the presence of training members in the input dataset $S_Y$. Notably, even under the stringent privacy guarantees of DP-SGD with $\varepsilon = 1$, our HR-MMD test achieved significant detection rates, indicating robustness against differential privacy defenses. These findings validate the HR-MMD test as a robust and reliable method for membership inference in adversarial settings where models employ advanced defense strategies. The ability to effectively detect membership information underlines the practical relevance of our method in privacy risk assessments and reinforces its standing as a state-of-the-art benchmark in this domain.

## K    DISCUSSION OF EXTENSIVE MIXUP MEMBERSHIP SCENARIOS

We further extended our experimental framework to investigate mixed membership scenarios, reflecting real-world situations where the input test data $S_Y$ contains varying proportions of training members and non-members. This exploration aims to assess the sensitivity and reliability of our HR-MMD test in detecting membership information under more complex and challenging conditions.

The experimental setup remains consistent with Section G.7. We constructed input datasets $S_Y$ of size 1,000, comprising different mixup ratios $r \in \{40\%, 50\%, 60\%, 70\%, 80\%, 90\%\}$ of training members from $S_T$ and non-members from $S_{\text{non}}$. The partitioning into training and testing subsets for both $S_X$ and $S_Y$, as well as the selection of $S_X^{\text{tr}}$ and $S_X^{\text{te}}$, followed the procedures outlined previously. The true positive rate (TPR) is defined as the proportion of datasets correctly identified as containing training members, and the false positive rate (FPR) is the proportion of datasets incorrectly identified as containing training members when they do not. We evaluated our HR-MMD test and traditional set-based MIAs under the membership inference defense strategy L1 regularization discussed in Section J.

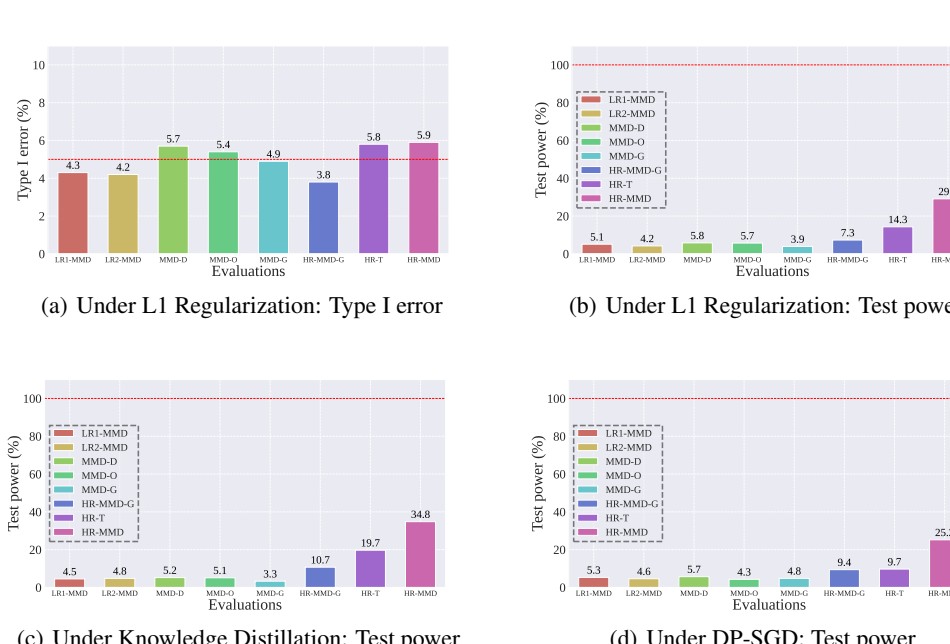

(a) Under L1 Regularization: Type I error

(b) Under L1 Regularization: Test power

(c) Under Knowledge Distillation: Test power

(d) Under DP-SGD: Test power

Figure 8: Evaluations under membership inference defenses. As shown, the HR-MMD test consistently demonstrated superior performance compared to traditional two-sample tests across all defense mechanisms. Under knowledge distillation and L1 regularization, our method maintained high test power, effectively detecting the presence of training members in the input dataset $S_Y$. Notably, even under the stringent privacy guarantees of DP-SGD with $\varepsilon = 1$, our HR-MMD test achieved significant detection rates, indicating robustness against differential privacy defenses. The experiments were conducted on CIFAR-10 using ResNet-18.

Experimental results in Figure 9 show that our HR-MMD test significantly outperforms prior set-based MIAs. The HR-MMD test consistently exhibited strong performance across all mixup ratios and the defense mechanism. With a varying proportion of training members, our method maintained best TPR at the same $FPR = 10\%$, effectively detecting the presence of training data within $S_Y$. This sensitivity demonstrates the capability of the HR-MMD test to discern subtle shifts in data distributions resulting from the inclusion of training members. Under the implemented defenses, our method's performance remained robust. The application of defense mechanisms did not significantly impair the HR-MMD test's ability to detect membership information, highlighting its resilience in adversarial settings.

In contrast, traditional set-based MIAs struggled notably in these scenarios. Their true positive rates decreased markedly as the mixup ratio decreased or when defenses were applied. This limitation underscores the challenges traditional MIAs face in handling mixed membership data and the necessity for more sophisticated approaches like the HR-MMD test. This disparity highlights the advantage of our approach, which leverages high-level representations and a learned deep kernel to capture subtle distributional differences that persist despite the application of defenses.

The comprehensive evaluations conducted in these mixed membership scenarios reinforce the reliability and practical applicability of our HR-MMD test. Its consistent performance across varying proportions of training members and in the presence of advanced defense mechanisms underscores its utility for privacy-sensitive applications. The HR-MMD test's robustness affirms its suitability for real-world deployment, where data may not be cleanly partitioned and defense strategies are commonly employed.

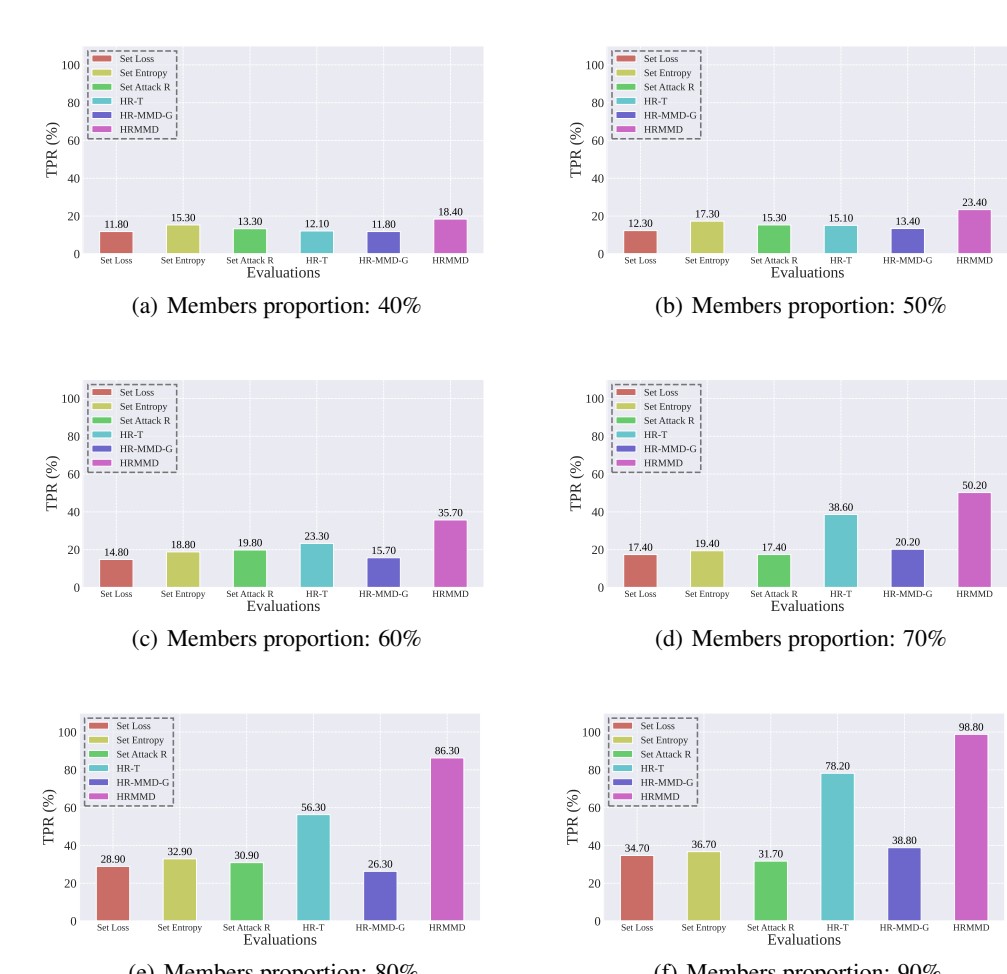

Figure 9: Evaluations on extensive mixup membership scenarios The true positive rate (TPR) is defined as the proportion of datasets correctly identified as containing training members, and the false positive rate (FPR) is the proportion of datasets incorrectly identified as containing training members when they do not. We evaluated our HR-MMD test and traditional set-based MIAs under the membership inference defense strategy L1 regularization discussed in Section J. As shown, the HR-MMD test consistently exhibited strong performance across all mixup ratios and the defense mechanism. With a varying proportion of training members, our method maintained best TPR at the same $FPR = 10\%$, effectively detecting the presence of training data within $S_Y$. The experiments were conducted on CIFAR-10 using ResNet-18.

## L    DISCUSSION ON THE APPLICATION FOR FRACTION ESTIMATION TASK

In this section, we explore the application of our HR-MMD test to the task of estimating the fraction of training members within a given input test dataset $S_Y$. This task extends beyond traditional binary membership inference, requiring precise statistical modeling to determine the proportion of members present in the dataset.

When the model has been trained using membership inference defenses, the HR-MMD test exhibits rejection rates that vary with different membership fractions $f$. In such cases, we can estimate the fraction directly by mapping the observed rejection rates to the corresponding membership fractions. This mapping is derived from a calibration curve based on validation data, allowing for accurate fraction estimation. However, in scenarios where the model is trained without membership inference defenses, the HR-MMD test tends to be highly sensitive, achieving a 100% rejection rate even at low membership fractions. In these cases, the rejection rate does not provide granular information correlated with $f$, necessitating an alternative approach for fraction estimation.

Our alternative methodology involves the following steps: **I.** Random Subset Sampling: We randomly sample multiple subsets from the input dataset $S_Y$. **II.** HR-MMD Testing on Subsets: For each subset, we perform the HR-MMD test to detect the presence of training members. **III.** Collection of Non-Rejected Subsets: Among the subsets where the test *does not* detect any members (i.e., the null hypothesis is not rejected), we collect all the elements and take their union. This union represents the portion of $S_Y$ estimated to consist of non-members.

We then estimate the fraction of non-members as:

$$\hat{f}_{\text{non-members}} = \frac{\text{Number of unique elements in the union of non-rejected subsets}}{N},$$

where $N$ is the total number of samples in $S_Y$. Accordingly, the estimated fraction of members is:

$$\hat{f} = 1 - \hat{f}_{\text{non-members}}.$$

This methodology allows us to estimate the fraction of training members even when the test's sensitivity prevents a direct correlation between rejection rates and membership fractions. By focusing on subsets where no members are detected, we can infer the proportion of the dataset likely to be non-members, thus providing an estimation of the membership fraction.

The ability of the HR-MMD test to estimate membership fractions showcases its versatility and advanced applicability in privacy risk assessments. By extending beyond binary inference to quantify the extent of membership within a dataset, our method offers a more nuanced understanding of potential privacy breaches. This is particularly valuable in scenarios where knowing the proportion of compromised data is crucial for assessing the severity of a breach and formulating appropriate responses.

## M    DISCUSSION ON HANDLING OUT-OF-DISTRIBUTION DATA

In practical applications, datasets often contain out-of-distribution (OOD) samples that deviate from the data distribution on which a model was trained. The presence of such OOD samples poses significant challenges for membership inference methods, potentially leading to inaccurate assessments or false conclusions. To enhance the robustness of our HR-MMD test in these scenarios, we propose an extended approach capable of effectively managing OOD data.

Our method involves a two-step process designed to first detect and then mitigate the influence of OOD samples within the input test data $S_Y$. Initially, we employ a traditional two-sample test between $S_Y$ and the non-training data $S_{\text{non}}$ to identify any significant deviations from the expected data distribution. This preliminary detection is crucial for ensuring that subsequent membership inference analyses are not compromised by the presence of OOD samples.

Upon detection of OOD samples, we proceed to isolate in-distribution subsets from $S_Y$. This can be achieved by selecting subsets of $S_Y$ that are statistically similar to $S_{\text{non}}$ according to the two-sample test, thereby filtering out the OOD samples. By focusing on these in-distribution subsets, we can apply the HR-MMD test more effectively to assess the presence of training members without the confounding effects of OOD data.

This two-step approach ensures that the influence of OOD samples does not compromise the accuracy of the membership inference. By first detecting and then isolating in-distribution data, our method maintains reliable detection capabilities even when OOD samples constitute a significant portion of $S_Y$. This enhances the applicability of the HR-MMD test in real-world scenarios, where datasets may not be clean or fully representative of the training distribution.

Effectively managing OOD scenarios reinforces the utility of the HR-MMD test as a robust tool for privacy risk assessment. By addressing the complexities inherent in real-world data, our method provides a more comprehensive and reliable evaluation of membership inference risks, ensuring that privacy assessments remain accurate and meaningful even in the presence of diverse data conditions.

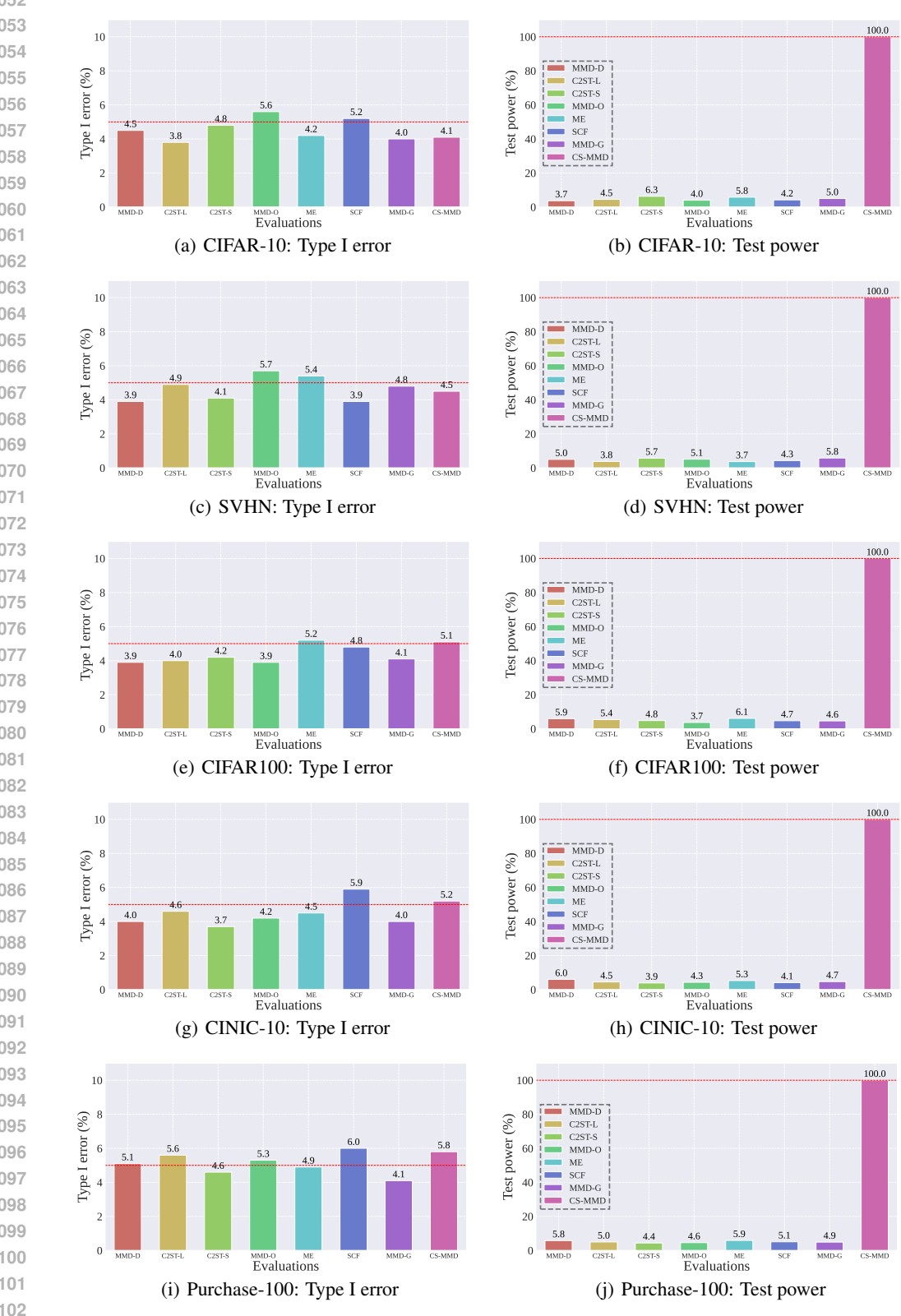

Figure 10: Test power vs. type I error comparison between our HR-MMD test and other baselines. As seen in subfigure (a,c,e,g,i), all methods achieve an ideal type I error around the significance level of 0.05. In subfigure (b,d,f,h,j), only our HR-MMD test (referred to as CS-MMD here) can achieve an ideal test power of 1. The experiments of subfigure (a-h) are conducted on ResNet-18. The experiments of subfigure (i) and (j) are conducted on a 4-layer MLP model.

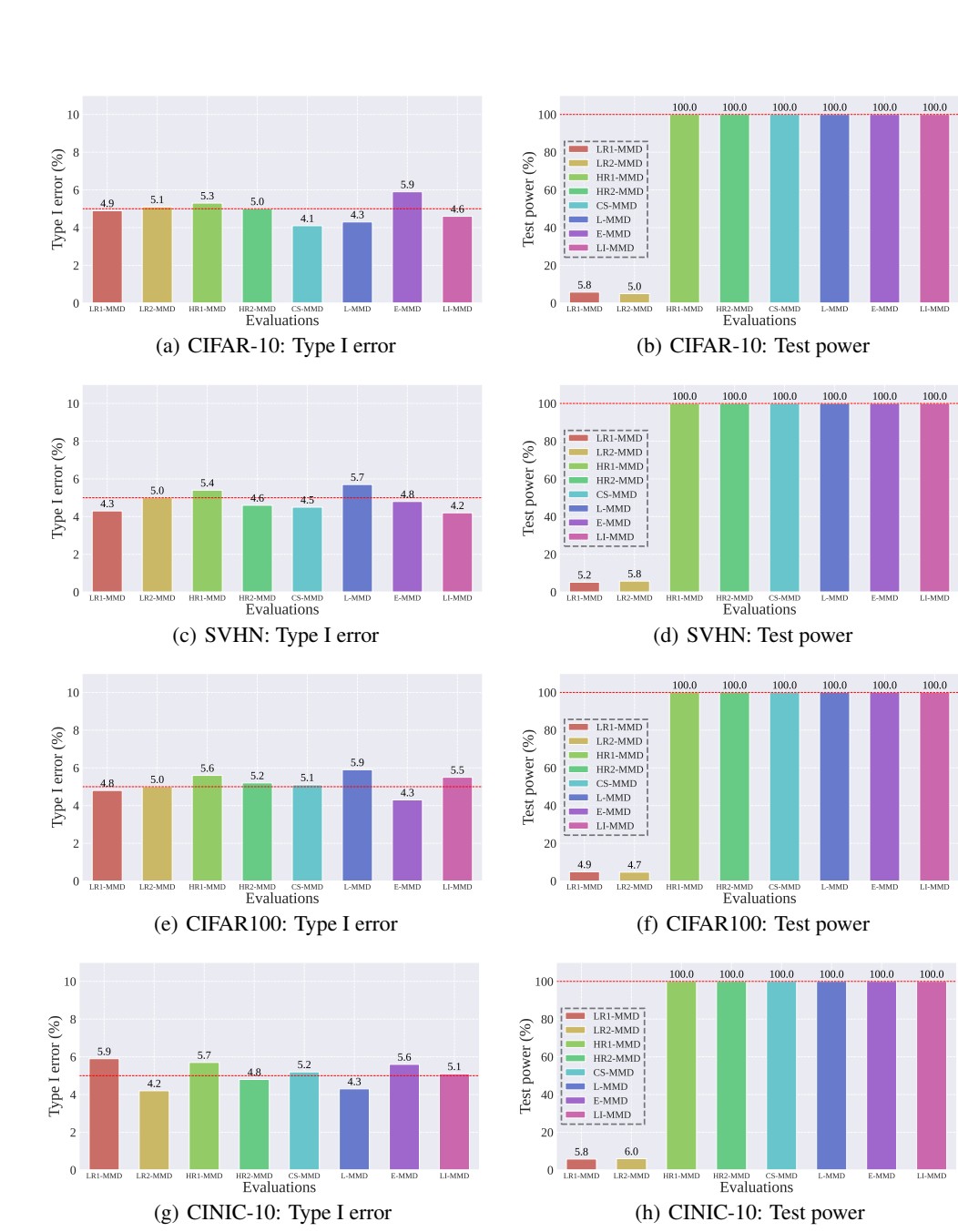

Figure 11: Test power vs. type I error comparison between our HR-MMD test and other baselines. As seen in subfigure (a,c,e,g), all methods achieve an ideal type I error around the significance level of 0.05. In subfigure (b,d,f,h), only our HR-MMD tests (referred to as HR1-MMD, HR2-MMD, CS-MMD, L-MMD, E-MMD and LI-MMD here) can achieve an ideal test power of 1. The experiments are conducted on ResNet-18.

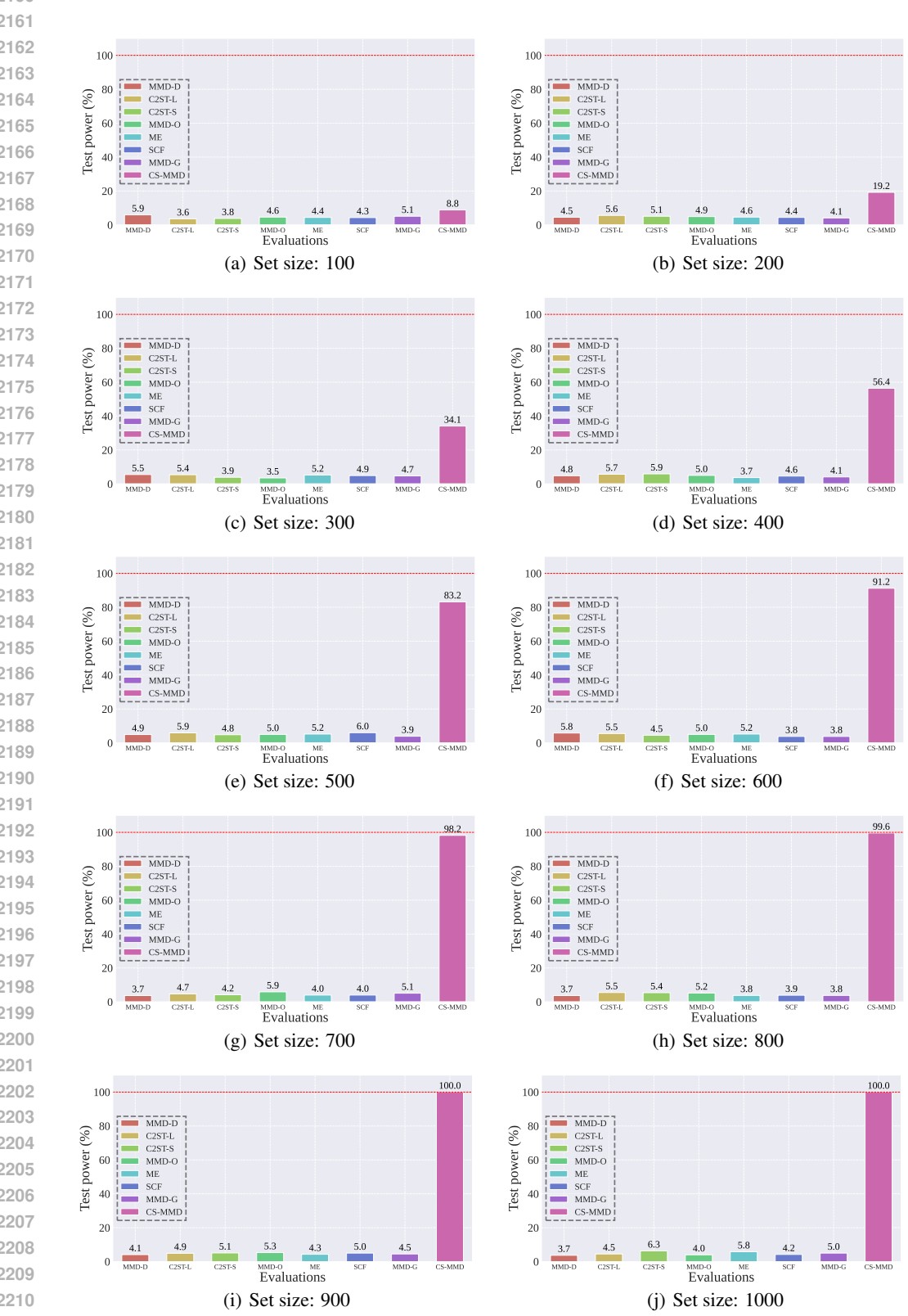

Figure 12: The test power of the HR-MMD test and baselines on different dataset sizes. As shown in the subfigures, only our HR-MMD test (referred to as CS-MMD here) achieves a test power greater than the significance level ($\alpha = 0.05$). The experiments were conducted on CIFAR-10 using ResNet-18.

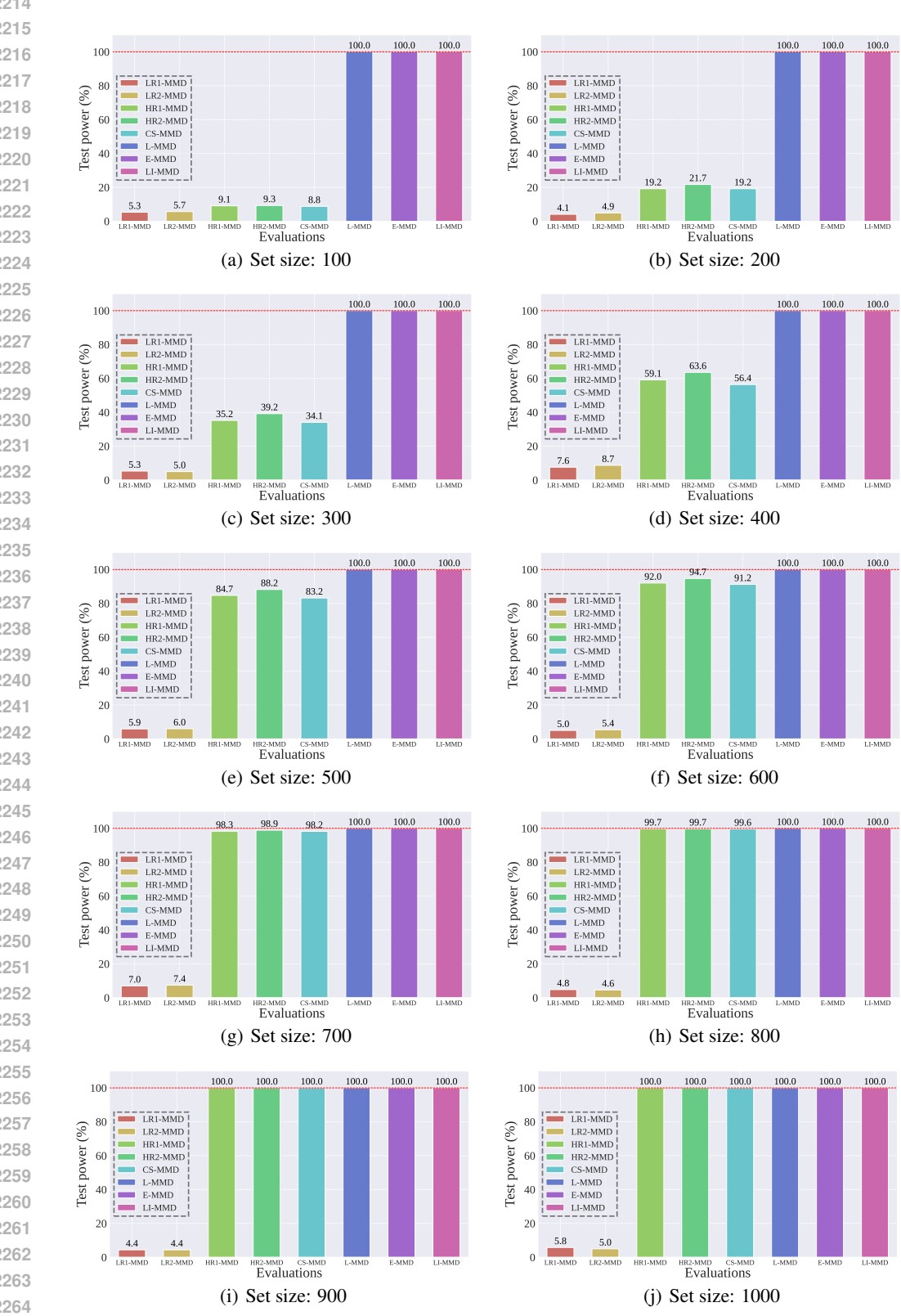

Figure 13: The test power of the HR-MMD test and baselines on different dataset sizes. As shown in the subfigures, only our HR-MMD tests (referred to as HR1-MMD, HR2-MMD, CS-MMD, L-MMD, E-MMD and LI-MMD here) achieves a test power greater than the significance level ($\alpha = 0.05$). The experiments were conducted on CIFAR-10 using ResNet-18.

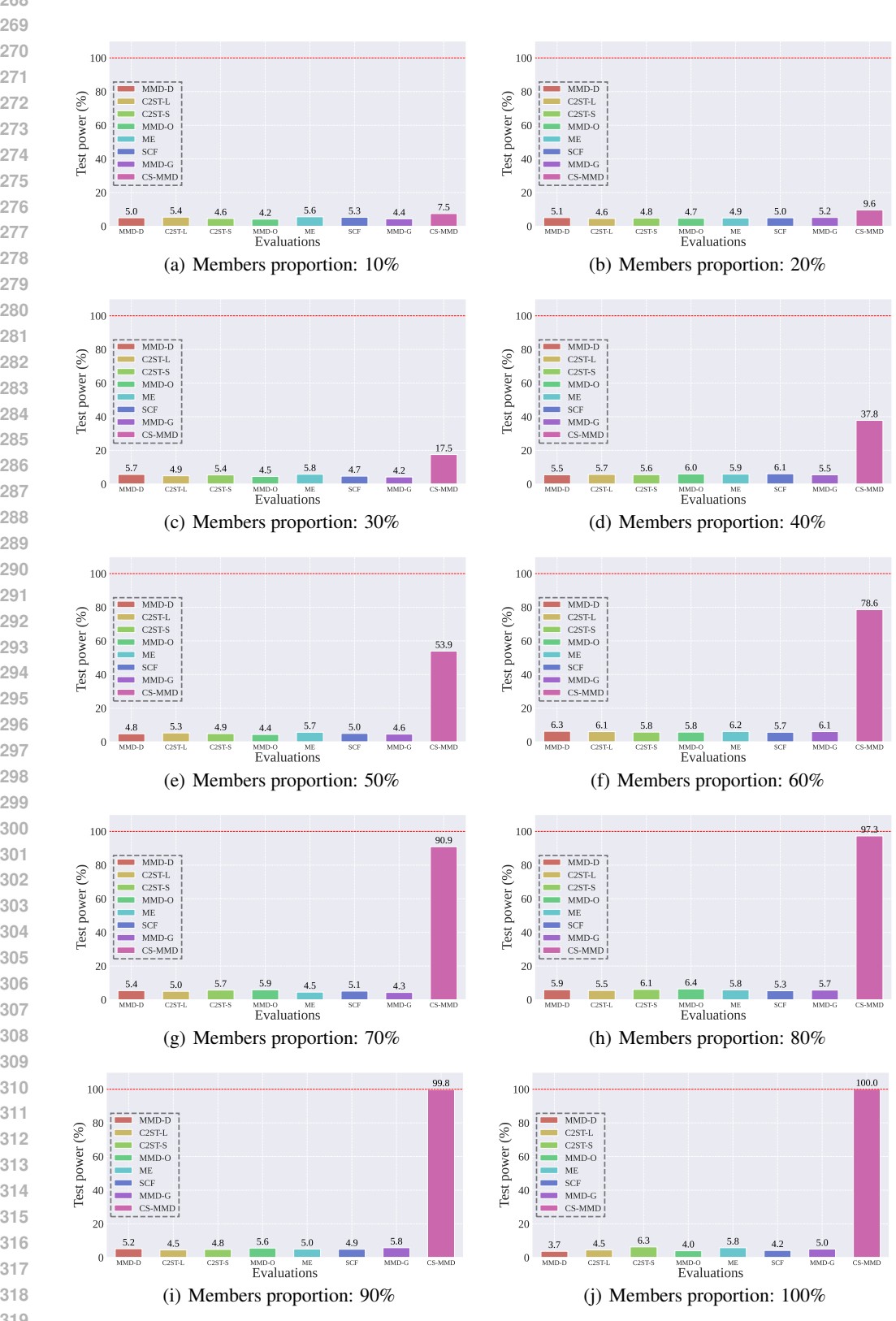

Figure 14: The test power of the HR-MMD test and baselines on different mixture proportions of members. As shown in the subfigures, only our HR-MMD test (referred to as CS-MMD here) achieves a test power greater than the significance level ($\alpha = 0.05$). The experiments were conducted on CIFAR-10 using ResNet-18.

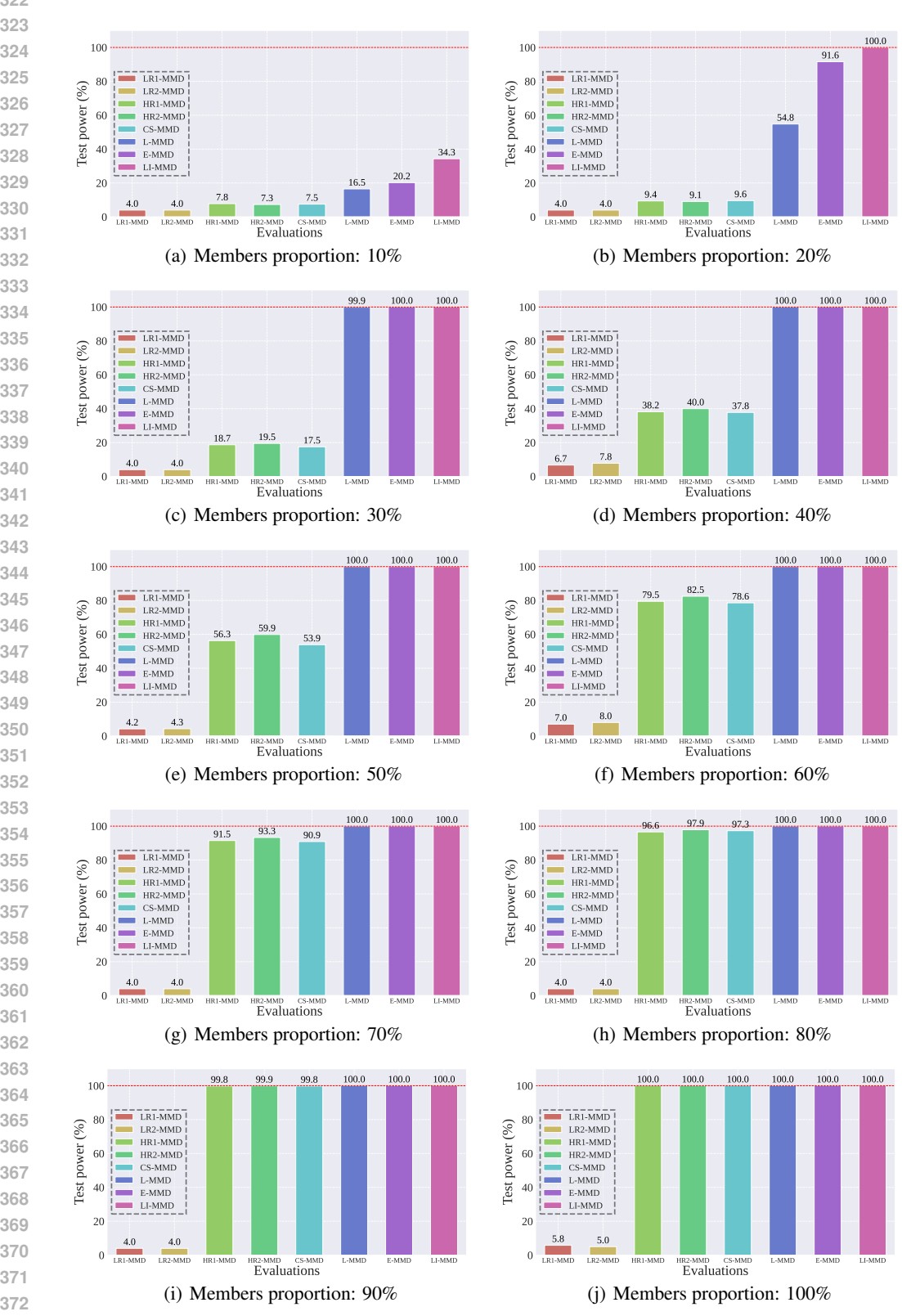

Figure 15: The test power of the HR-MMD test and baselines on different mixture proportions of members. As shown in the subfigures, only our HR-MMD tests (referred to as HR1-MMD, HR2-MMD, CS-MMD, L-MMD, E-MMD and LI-MMD here) achieves a test power greater than the significance level ($\alpha = 0.05$). The experiments were conducted on CIFAR-10 using ResNet-18.

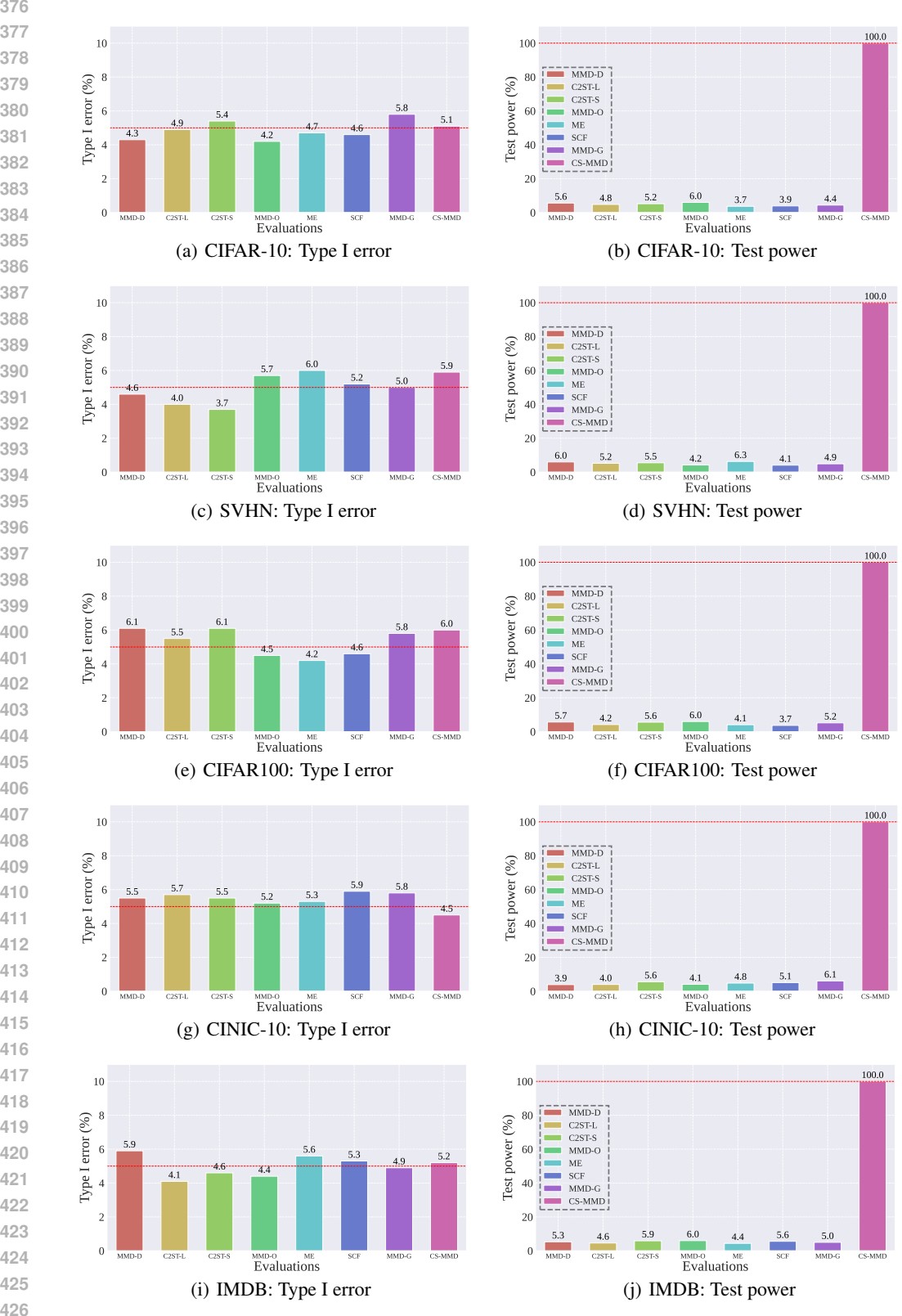

Figure 16: Test power vs. type I error comparison between our HR-MMD test and other baselines. As seen in subfigure (a,c,e,g,i), all methods achieve an ideal type I error around the significance level of 0.05. In subfigure (b,d,f,h,j), only our HR-MMD test (referred to as CS-MMD here) can achieve an ideal test power of 1. The experiments of subfigure (a-h) are conducted on WRN-34. The experiments of subfigure (i) and (j) are conducted on a 2-layer LSTM model.

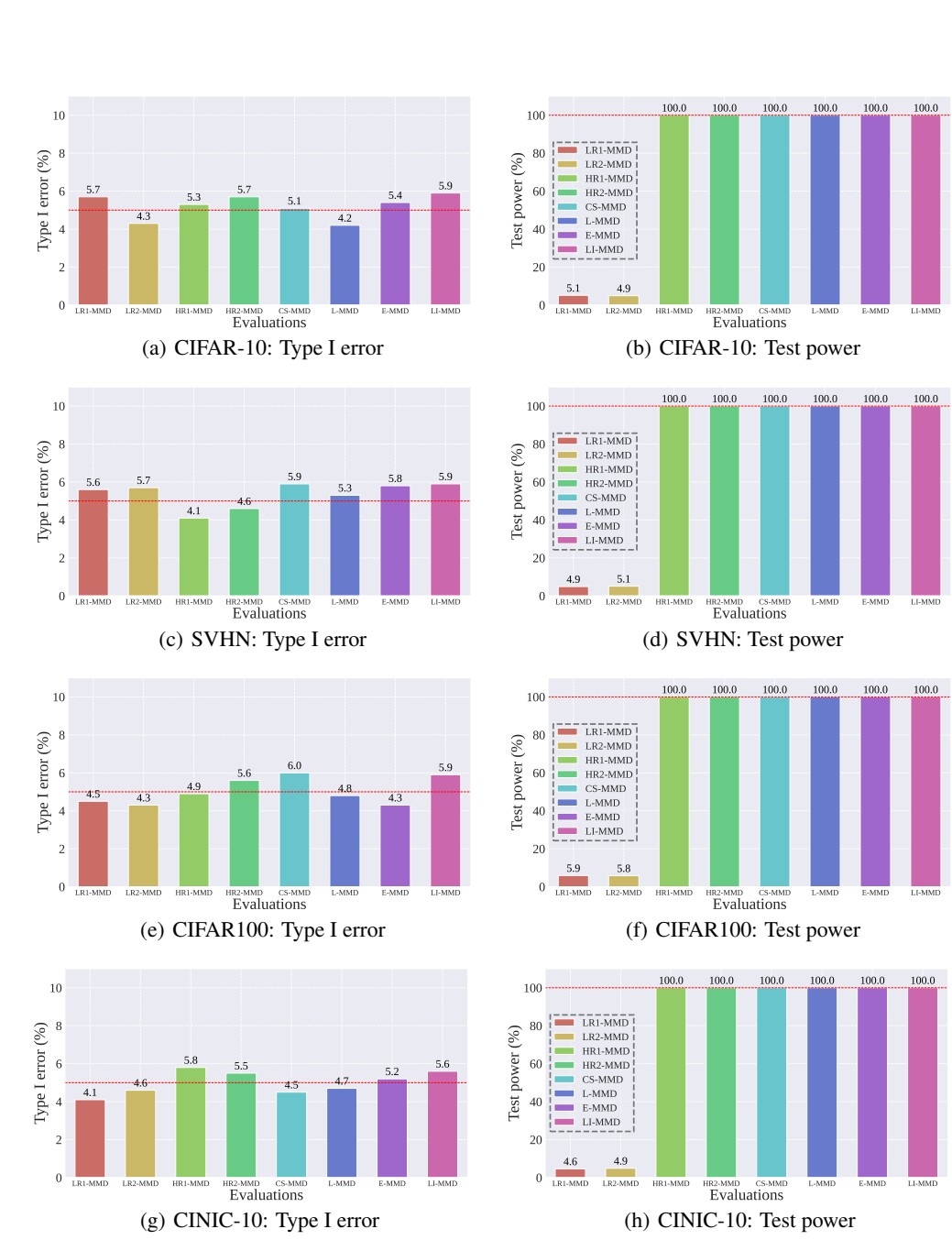

Figure 17: Test power vs. type I error comparison between our HR-MMD test and other baselines. As seen in subfigure (a,c,e,g), all methods achieve an ideal type I error around the significance level of 0.05. In subfigure (b,d,f,h), only our HR-MMD tests (referred to as HR1-MMD, HR2-MMD, CS-MMD, L-MMD, E-MMD and LI-MMD here) can achieve an ideal test power of 1. The experiments are conducted on WRN-34.

