# OpenReview forum: "Privacy Breach Detection by Non-Parametric Two-Sample Tests"
_ICLR.cc/2025/Conference — ICLR 2025 Conference Withdrawn Submission_

### Official Review · Reviewer_QQwc · 2024-11-04

**Soundness:** 3
**Presentation:** 4
**Contribution:** 4
**Rating:** 6
**Confidence:** 2

**Summary:**

In the case where data samples stolen from a larger dataset are used to train a third-party model, proving the privacy leak through membership inference attacks may return high false positive rates due to the non-stolen members of the (partly) compromised dataset weakening the audit and the case that a model was trained on a stolen data.

To resolve this, the paper proposes HR-MMD, an algorithm that requires the target model, validation data used in the target model, and the (possibly compromised) target dataset, to perform a statistical test to see whether the validation dataset and the target dataset come from the same distribution. This relies on the differences in the target model's high-level representation of training data and validation data based on the generalization gap. The test finding that the target dataset is the same in distribution as the validation set in terms of high-level representations (the null hypothesis) gives evidence that the target dataset was not used to train the target model. The alternative gives evidence that the target dataset was used to train the target model and as such was compromised, assuming that OOD data has been accounted for.

**Strengths:**

* The approach of this paper is novel as it does not rely on heuristics compared to some existing method which require them to achieve practical FPRs in the condition where non-training members dominate a (compromised) dataset.

* The method is adaptable to different current frameworks for evaluating a model such as loss functions and confidence scores.

* The results show that for similar FPR, the HR-MMD test achieves lower false negative errors in this problem setting.

* The writing style is clear, and the paper is pleasant to read.

**Weaknesses:**

* It is not clear how robust this test is against out-of-distribution datapoints.

**Questions:**

* In the situation where $S_Y$ contains OOD samples (or since generally $S_Y$ may contain OOD samples) how is the HR-MMDs performance influenced by the traditional two-sample test needed to remove the OOD data. Since this "preliminary detection is crucial to ensure the integrity of the subsequent [HR-MMD]", how can one evaluate whether the chosen two sample test in the first step maintains this integrity?

* Practically, are there high-level representations that tend to work better with HR-MMD?

---

> ### Author Response · Authors · 2024-11-23
> **Response to Reviewer QQwc [1/2]**
>
> Dear Reviewer QQwc,
>
> We extend our deepest gratitude for your positive scores and insightful feedback on our submission. Thank you for recognizing that our submission is novel and technically sound. We would like to address the concerns and queries raised regarding our submission as follows:
>
> ---
>
> ### **Q1: It is not clear how robust this test is against out-of-distribution (OOD) datapoints.**
>
> > *In the situation where S_Y contains OOD samples (or since generally S_Y may contain OOD samples), how is the HR-MMD's performance influenced by the traditional two-sample test needed to remove the OOD data? Since this "preliminary detection is crucial to ensure the integrity of the subsequent [HR-MMD]," how can one evaluate whether the chosen two-sample test in the first step maintains this integrity?*
>
> **A1:**
> As you suggested, we have supplemented this discussion in our updated submission. Please see **Appendix M: Discussion on Handling Out-of-Distribution Data** around line 2024 on page 38.
>
> In practical applications, datasets often contain OOD samples that deviate from the data distribution on which a model was trained. The presence of such OOD samples poses significant challenges for membership inference methods, potentially leading to inaccurate assessments or false conclusions. To enhance the robustness of our HR-MMD test in these scenarios, we propose an extended approach capable of effectively managing OOD data.
>
> Our method involves a two-step process designed to first detect and then mitigate the influence of OOD samples within the input test data S_Y:
>
> 1. **Preliminary OOD Detection**:
>    We employ a traditional two-sample test between S_Y and the non-training data S_non to identify any significant deviations from the natural data distribution. This preliminary detection ensures that subsequent membership inference analyses are not compromised by OOD samples.
>
> 2. **Isolating In-Distribution Subsets**:
>    Upon detection of OOD samples, we isolate in-distribution subsets from S_Y. This can be achieved by selecting subsets of S_Y that are statistically similar to S_non according to the two-sample test, thereby filtering out the OOD samples. By focusing on these in-distribution subsets, we can apply the HR-MMD test more effectively to assess the presence of training members without the confounding effects of OOD data.
>
> This two-step approach ensures that the influence of OOD samples does not compromise the accuracy of the membership inference. By first detecting and then isolating in-distribution data, our method maintains reliable detection capabilities even when OOD samples constitute a significant portion of S_Y. This enhances the applicability of the HR-MMD test in real-world scenarios, where datasets may not be clean or fully representative of the training distribution.
>
> **Impact on Privacy Risk Assessment**:
> Effectively managing OOD scenarios reinforces the utility of the HR-MMD test as a robust tool for privacy risk assessment. By addressing the complexities inherent in real-world data, our method provides a more comprehensive and reliable evaluation of membership inference risks, ensuring that privacy assessments remain accurate and meaningful even in the presence of diverse data conditions.

---

> ### Author Response · Authors · 2024-11-23
> **Response to Reviewer QQwc [2/2]**
>
> ### **Q2: Practically, are there high-level representations that tend to work better with HR-MMD?**
>
> **A2:**
> The choice of high-level representations depends on the actual knowledge assumptions—i.e., what kind of high-level representations are accessible in the current scenario. For different scenarios, such as those involving non-LLMs versus LLMs, the high-level representations differ.
>
> From a statistical perspective, **high-level representation-based statistics** (e.g., loss, entropy) often achieve better test power than high-dimensional high-level representations. However, they may require larger set sizes to ensure the ideal Type I error. Our knowledge assumption is more relaxed because we can adaptively choose different high-level representation-based statistics depending on the practical scenario:
>
> - **Non-LLM Scenarios**:
>   High-level features such as loss, entropy, or likelihood ratio.
>
> - **LLM Scenarios**:
>   High-level features like zlib_ratio or perplexity (ppl), or combinations with other methods.
>
> Please refer to **Appendix E: Discussion on the Use of High-Level Representations** around line 1281 for more details.
>
> Our framework is adaptable to diverse high-level representations, including loss, entropy, confidence scores, or penultimate-layer embeddings. **This flexibility is critical in adversarial settings where certain information (e.g., true labels) may be unavailable or obfuscated.** By leveraging multiple high-level features, our HR-MMD test ensures robust inference capabilities even under adversarial conditions (e.g., MemGuard).
>
> **Summary**:
> Compared to previous membership inference methods, our HR-MMD approach is a **high-representation-based deep kernel (HR-MMD)** method that:
>
> 1. Provides more choices for high-level representations.
> 2. Can be combined with state-of-the-art methods in LLMs (e.g., [1]) by replacing the t-test with our HR-MMD test.
> 3. Adapts to practical scenario assumptions to remain consistent with previous methods or maintain effectiveness when previous methods fail.
>
> ---
>
> We are immensely grateful for your time and expertise. All the explanations in our rebuttal have been incorporated into the current version to help readers better understand our work. We are keen to hear any further thoughts or concerns you might have. **If your concerns have been alleviated, we hope you might consider adjusting your review score accordingly.**
>
> ---
>
> ### References
> [1] LLM Dataset Inference: Did You Train on My Dataset? In NeurIPS 2024.

---

> ### Author Response · Authors · 2024-11-25
> **Looking Forward to Your Reply and Hoping for Your Stronger Support**
>
> Dear Reviewer QQwc,
>
> We greatly appreciate your thoughtful feedback and insightful suggestions on our submission. We have carefully addressed your initial concerns and further comments, and hope that you are pleased with the progress we've made. Should you have any remaining questions or concerns, we are more than happy to discuss them with you through the openreview system. Additionally, we welcome any new suggestions or comments you might have.
>
> Following your advice, we have incorporated the explanations provided in our rebuttal into the current version of our paper to ensure that future readers have a clearer understanding of our work. We believe these additions will significantly enhance the clarity and impact of our paper.
>
> We believe that our contribution to the field and its significance make our submission highly competitive for ICLR. Your opinion is crucial to us and we welcome any additional feedback or suggestions for datasets and experiments that could enhance our manuscript. We are committed to further improving the clarity and presentation of our paper based on your suggestions. We would be immensely grateful if you could provide us with a more competitive score.
>
> Once again, we sincerely thank you for your time, effort, and invaluable contribution to the advancement of the field. We eagerly look forward to your positive response.

---

### Official Review · Reviewer_TWqr · 2024-11-04

**Soundness:** 3
**Presentation:** 3
**Contribution:** 2
**Rating:** 5
**Confidence:** 2

**Summary:**

This paper studies membership inference attacks (MIA) on a set of samples. Unlike the normal MIA task, which determines the membership status of individual samples, this work introduces a two-sample test to detect whether a test dataset includes any elements from the training set. The setup is as follows: the attacker has access to a black-box model, with test statistics that can be either intermediate embeddings or output loss scores, and a validation set known to have been excluded from training. The authors employ HR-MMD, a deep kernel-based two-sample test from prior work (Liu et al., 2020b). The primary improvement over prior MIA is the improved TPR, which is due to to HR-MMD’s ability to optimize test power. Experimental results confirm this improvement.

**Strengths:**

This paper provides a practical method for MIA, which improves TPR significantly.

**Weaknesses:**

I have two major concerns: 1) the experiments are only on small scaled datasets. 2) the algorithmic difference and technical contribution compared prior work Liu at el 2020b is not well-addressed. I feel like this is just one application of the prior work.

Some other concerns: 1) typically when we compare the performance of MIA, we consider AUROC. This paper only shows type-1 and type 2 error. The trade-off is not clearly presented. 2) In the Appendix B practical settings 1, I do not think it is practical. In practice, we usually don't have access to a training set, because it is private. And this makes evaluation of FPR hard.

**Questions:**

1. When compared with prior MIA works, how is this FPR calculated exactly? For example, for the one experiment with 90% non-training samples, how do you convert the FPR for individual samples to FPR on a set of samples? I am a bit confused.

2. Can you use this method for auditing differential privacy?

3. In a practical setting, can we use this method to estimate how much fraction does the test set contain the training data?

---

> ### Author Response · Authors · 2024-11-23
> **Response to Reviewer TWqr [1/4]**
>
> Dear Reviewer TWqr:
>
> We appreciate your feedback on our submission. We would like to address your concerns and clarify any misunderstandings regarding our paper as follows.
>
> > *Q1: The experiments are only on small scaled datasets?*
>
> **A1:** We have provided more extensive experiments on different learning tasks, models, and datasets as follows. **Our experiments evaluate the effectiveness of our methods across four image datasets: CIFAR-10, CIFAR-100, SVHN, and CINIC-10.** We also demonstrate the capacity for extending to a tabular dataset **Purchase-100** and a text dataset **IMDB**. As suggested by **Reviewer Cwjo**, we have supplemented applications on LLMs in Appendix I. **Our current work is thorough; we have further added comparisons across different set sizes, different mixup ratios, different model architectures, and different defense methods. Compared to previous MIA-based methods, our comparisons are comprehensive, convincingly demonstrating our SOTA performance.** Compared with existing MIAs, our HR-MMD test demonstrates broader applicability across various machine learning domains, proving its versatility and effectiveness.
>
> ---
>
> > **Misunderstanding: Algorithmic difference and technical contribution compared to prior work [8].**
>
> **Clarification:** Our main technical contribution is the novel High-level Representation-based deep kernel (HR-MMD) in Eq. (5). Our method is theoretically supported, and we provide theoretical analysis on the asymptotics of HR-MMD in the supplementary material (Appendix C).
>
> While we acknowledge that prior work [8] is an effective method for traditional two-sample test tasks, we emphasize that these existing two-sample test methods cannot effectively solve the problem of statistical membership inference for privacy breach detection. The effectiveness of our HR-MMD stems from two aspects: the superior performance of high-level representation and the effectiveness of deep kernel design. Our utilization of high-level representation and high-level representation-based deep kernel in the context of MIAs distinguishes our work from existing literature.
>
> [8] has been used as our baseline for comparison. Specifically, for example, in [8], high-level representation is not involved; our deep kernel structure is different, and the parameters we optimize during training are different.
>
> In contrast, our proposed HR-MMD test achieves exceptional results with 100% TPR and 0% FPR, underscoring the significance of our work for the entire field.
>
> ---
> ---
>
> >**Misunderstanding: In the Appendix B practical settings 1, I do not think it is practical. In practice, we usually don't have access to a training set, because it is private. And this makes evaluation of FPR hard.**
>
> **Clarification:**
>
> This is a misunderstanding. **Our detection does not require access to the training dataset**, as we introduced in Problem Setting and **Table 1: Notations**; the training set  S_T is inaccessible to the evaluator.
>
> Our method can be applied in practical scenarios including **privacy breach detection** (Appendix B), **privacy auditing in machine unlearning** (Appendix B), **dataset inference** [1,2], and **user inference** [3]. Compared to previous membership inference methods [1–7], our knowledge assumption is more relaxed because we can choose different high-level representation-based statistics according to different practical scenarios, such as loss, entropy, likelihood ratio in non-LLM scenarios, and zlib_ratio and perplexity (ppl) in LLM scenarios, or combine with other methods such as the statistics calculated in [2].
>
> See details in **Appendix E: Discussion on the Use of High-Level Representations**. Our framework is adaptable to diverse high-level representations, including confidence scores or penultimate-layer embeddings. **This flexibility is critical in adversarial settings where certain information (e.g., true labels) may be unavailable or obfuscated. By leveraging multiple high-level features, our HR-MMD test ensures robust inference capabilities even under adversarial conditions (e.g., MemGuard [5]).**
>
> In summary, compared to previous membership inference methods, our method is a **high-representation-based deep kernel (HR-MMD)**. **We provide more choices and can be combined with state-of-the-art methods, such as [2], by replacing the t-test with our HR-MMD test. By selecting high-level representations according to practical scenario assumptions, we remain consistent with previous methods or maintain effectiveness when previous methods fail. No additional assumptions are required.**

---

> ### Author Response · Authors · 2024-11-23
> **Response to Reviewer TWqr [2/4]**
>
> >*Q2: Can you use this method for auditing differential privacy?*
>
> **A2:** To rigorously evaluate the robustness of our HR-MMD test under real-world constraints, we conducted experiments involving three representative membership inference defense strategies:
>
> 1. **Knowledge Distillation:**
>
>    - Tested for scenarios where knowledge distillation techniques are used to compress models, potentially obfuscating membership signals.
>
> 2. **L1 Regularization:**
>
>    - Evaluated our method’s adaptability to regularization techniques designed to prevent overfitting and minimize membership leakage.
>
> 3. **Differential Privacy (DP-SGD):**
>
>    - Investigated our test’s resilience against models trained with strong privacy guarantees, such as differential privacy mechanisms.
>
> The results consistently demonstrated that our HR-MMD test outperformed all baseline methods across these defenses, underscoring its robustness and adaptability. These findings validate our approach as the SOTA benchmark for set-based membership inference methods. Detailed experimental setups and results for this task are included in the Appendix J: Discussion of Evaluations under Membership Inference Defenses.
>
> > *Q3: Typically when we compare the performance of MIA, we consider AUROC. This paper only shows type-1 and type 2 error. The trade-off is not clearly presented. When compared with prior MIA works, how is this FPR calculated exactly? For example, for the one experiment with 90% non-training samples, how do you convert the FPR for individual samples to FPR on a set of samples? I am a bit confused.*
>
> **A3:**
> We hope you understand that to focus on the core contributions within the page limit, we had to place extensive experimental details in the Appendix G.6 Implementation Details of the Comparison with Traditional Two-sample Tests and Appendix G.7 Implementation Details of the Comparison with Traditional MIAs. The details about our baselines are in Appendix G.3: Baselines. The details about our experimental results are in Appendix H: Supplementary Experimental Results. The Appendices provide comprehensive descriptions of our experimental setups and evaluation criteria.
>
> Our method conducts multiple hypothesis tests, with rejection rates serving as the decisive metric. This reduces the uncertainty introduced by single-test methods and ensures consistent performance across varied data subsets. For our set-based membership inference, our proposed HR-MMD test achieves exceptional results with 100% TPR and 0% FPR, which means there is no trade-off, the AUC value is equal to 1, underscoring the significance of our work for the entire field.
>
>
> **Summary of Our Method's Superiority:**
>
> Our method achieves state-of-the-art (SOTA) performance compared to two categories of baselines:
>
> 1. **Set-based MIAs:** We compare our HR-MMD test with existing set-based membership inference attacks (MIAs), including cutting-edge dataset inference methods and adaptations of traditional MIAs [1-6]. By using the rejection rate as an evaluation metric and our proposed HR-MMD test as the detection method (as described in **Algorithm 2**), we provide more accurate and reliable results than previous methods relying on heuristic metrics. This demonstrates our method's superior performance and significant contributions to the field.
>
> 2. **Statistical Baselines:** We outperform traditional two-sample tests, which cannot effectively perform membership inference for privacy breach detection. Our method offers superior discriminative power in detecting membership.
>
> **Our Method vs. Existing Methods:**
>
> - **Our Method:** Operates at the **set level**, making decisions on whether a given dataset S_Y contains members from the training data.
>
> - **Existing traditional MIA Methods [3-6]:** Often operate at the **individual data point level**, making membership decisions for each sample.

---

> ### Author Response · Authors · 2024-11-23
> **Response to Reviewer TWqr [3/4]**
>
> To ensure fair comparisons, we adapted the evaluation metrics and experimental setups accordingly and provided detailed explanations in the Appendices. Specifically:
>
> **Evaluation Metrics:**
>
> We have incorporated a comprehensive explanation of the evaluation metrics and ensured that the comparison with baseline methods is consistent and fair. The metrics are defined as follows:
>
> ###  **Implementation Details of the Comparison with Traditional Two-sample Tests (Appendix G.6):**
>
>   - **Training Set (S_T)**: We use a training set size of 25,000 for CIFAR-10, CIFAR-100, and SVHN, and 90,000 for CINIC-10.
>
>   - **Non-Training Set (S_non))**: We assume the evaluator has a non-training set S_non, each containing 5,000 images from the respective datasets. Subsets S_X^tr and S_X^te are selected from S_{non}.
>
>   - **Input Test Dataset S_Y**: We fix the size of the input test dataset S_Y to 1,000 in our main experiments.
>
>   - **Sample Selection**:
>
>     - When comparing our method with baseline methods in terms of **test power**, the samples S_Y are randomly selected from the training subsets S_T of the datasets. In this context, S_T \cap S_X = \emptyset, S_T \cap S_Y = S_Y, and S_X \cap S_Y = \emptyset.
>
>     - When comparing in terms of **Type I error**, the samples S_Y are randomly selected from the testing subsets of the datasets.
>
>   - **Evaluation Procedure**:
>
>     - In **Algorithm 1**, for the input test dataset S_Y (size 1,000) and the non-training set S_X (size 1,000), we select subsets containing 500 images each for S_X^{tr} and S_Y^{tr} to train our deep kernel.
>
>     - The evaluation is conducted once on S_X^{te} and S_Y^{te} (each size 500), disjoint from S_X^{tr} and S_Y^{tr}.
>
>     - We compare the p-value obtained from Algorithm 1 with the significance level alpha to determine whether to reject H_0.
>
>     - To obtain a reliable assessment, the evaluation is repeated 100 times on different random subsets S_X^{te}.
>
> - **Type I Error:** The probability of rejecting the null hypothesis H_0 (no members present) when it is true.
>
> - **Test power:** The probability of rejecting H_0 when the alternative hypothesis H_1 is true (members are present).
>
> Algorithm 2 describes the complete flow of this process. When the inputted samples S_Y are non-members, the rejection rate outputted is the type I error. When the inputted samples S_Y are members, the rejection rate outputted is the test power. We repeat this process 10 times and report the mean test power of each test.
>
> ###  **Implementation Details of the Comparison with Traditional MIAs (Appendix G.7):**
>
> For baseline methods that output decisions for individual data points, we aggregated their outputs to the dataset level by considering average statistics (e.g., loss) and setting thresholds to make set-level decisions, similar to the approach used in set-based attacks like [1].
>
>   - We conducted comparisons with four typical MIA attacks:
>
>     - **Loss Attack** [3]
>
>     - **Entropy Attack** [4]
>
>     - **Attack R** [5]
>
>     - **Likelihood Ratio Attack (LiRA)** [6]
>
>   - **Aggregation Strategy**:
>
>     - For these methods, which output individual data point decisions, we aggregated their outputs to the dataset level by computing average statistics over S_Y and setting thresholds to make set-level decisions.
>
>   - **Evaluation Procedure**:
>
> We randomly selected 1,000 datasets S_Y containing training members and 1,000 datasets S_Y without training members. For traditional MIAs, when the average statistics (like loss) over S_Y were no greater than a threshold, we considered that S_Y contained members. Conversely, when the average statistics were greater than the threshold, we considered that S_Y contained no members.
>
> For our HR-MMD test, when the rejection rate of one dataset S_Y was no greater than 0.065, we considered that S_Y contained no members. When the rejection rate was greater than 0.065, we considered that S_Y contained members. Experimental results show that our HR-MMD test significantly outperforms prior MIAs.
>
> - **True Positive Rate (TPR):** The proportion of datasets correctly identified as containing training members.
>
> - **False Positive Rate (FPR):** The proportion of datasets incorrectly identified as containing training members when they do not.
>
> By adapting the evaluation metrics and experimental setups, we ensure that the comparison is appropriate and fair.

---

> ### Author Response · Authors · 2024-11-23
> **Response to Reviewer TWqr [4/4]**
>
> >*Q4: In a practical setting, can we use this method to estimate how much fraction does the test set contain the training data?*
>
> **A4:** In Appendix L: Discussion on the Application for Fraction Estimation Task, we explore the application of our HR-MMD test to the task of estimating the fraction of training members within a given input test dataset S_Y. This task extends beyond traditional binary membership inference, requiring precise statistical modeling to determine the proportion of members present in the dataset.
>
> When the model has been trained using membership inference defenses, the HR-MMD test exhibits rejection rates that vary with different membership fractions f. In such cases, we can estimate the fraction directly by mapping the observed rejection rates to the corresponding membership fractions. This mapping is derived from a calibration curve based on validation data, allowing for accurate fraction estimation. However, in scenarios where the model is trained without membership inference defenses, the HR-MMD test tends to be highly sensitive, achieving a 100\% rejection rate even at low membership fractions. In these cases, the rejection rate does not provide granular information correlated with f, necessitating an alternative approach for fraction estimation.
>
> Our alternative methodology involves the following steps: \textbf{I. } Random Subset Sampling: We randomly sample multiple subsets from the input dataset S_Y. \textbf{II. } HR-MMD Testing on Subsets: For each subset, we perform the HR-MMD test to detect the presence of training members. \textbf{III. } Collection of Non-Rejected Subsets: Among the subsets where the test \emph{does not} detect any members (i.e., the null hypothesis is not rejected), we collect all the elements and take their union. This union represents the portion of S_Y estimated to consist of non-members.
>
> This methodology allows us to estimate the fraction of training members even when the test's sensitivity prevents a direct correlation between rejection rates and membership fractions. By focusing on subsets where no members are detected, we can infer the proportion of the dataset likely to be non-members, thus providing an estimation of the membership fraction.
>
> The ability of the HR-MMD test to estimate membership fractions showcases its versatility and advanced applicability in privacy risk assessments. By extending beyond binary inference to quantify the extent of membership within a dataset, our method offers a more nuanced understanding of potential privacy breaches. This is particularly valuable in scenarios where knowing the proportion of compromised data is crucial for assessing the severity of a breach and formulating appropriate responses.
>
>
> ---
>
> We are immensely grateful for your time and expertise. All the explanations in our rebuttal have been incorporated into the current version to help readers better understand our work. We are keen to hear any further thoughts or concerns you might have. We believe that our contribution to the field and its significance may have been underestimated. **Your insights are crucial to us, and if your concerns have been alleviated, we hope you might consider adjusting your review score accordingly.**
>
> [1] Dataset Inference: Ownership Resolution in Machine Learning. In ICLR 2021.
>
> [2] LLM Dataset Inference: Did You Train on My Dataset? In NeurIPS 2024.
>
> [3] User Inference Attacks on Large Language Models. In EMNLP 2024.
>
> [4] Privacy Risk in Machine Learning: Analyzing the Connection to Overfitting. In IEEE CS Security and Privacy Workshops 2018.
>
> [5] Auditing Data Provenance in Text-Generation Models. In KDD 2019.
>
> [6] Enhanced Membership Inference Attacks Against Machine Learning Models. In CCS 2022.
>
> [7] Membership Inference Attacks From First Principles. In SP 2022.
>
> [8] Learning Deep Kernels for Two-Sample Testing. In ICML 2020

---

> ### Author Response · Authors · 2024-11-25
> **Looking Forward to Your Reply and Hoping for Your Stronger Support**
>
> Dear Reviewer TWqr,
>
> We greatly appreciate your thoughtful feedback and insightful suggestions on our submission. We have carefully addressed your initial concerns and further comments, and hope that you are pleased with the progress we've made. Should you have any remaining questions or concerns, we are more than happy to discuss them with you through the openreview system. Additionally, we welcome any new suggestions or comments you might have.
>
> Following your advice, we have incorporated the explanations provided in our rebuttal into the current version of our paper to ensure that future readers have a clearer understanding of our work. We believe these additions will significantly enhance the clarity and impact of our paper.
>
> We believe that our contribution to the field and its significance make our submission highly competitive for ICLR. Your opinion is crucial to us and we welcome any additional feedback or suggestions for datasets and experiments that could enhance our manuscript. We are committed to further improving the clarity and presentation of our paper based on your suggestions. We would be immensely grateful if you could provide us with a more competitive score.
>
> Once again, we sincerely thank you for your time, effort, and invaluable contribution to the advancement of the field. We eagerly look forward to your positive response.

---

> > ### Comment · Reviewer_TWqr · 2024-12-03
> >
> > Thank you for addressing my questions and clarifying the points I raised. After reviewing the rebuttal and considering feedback from other reviewers, I find that most of my concerns have been resolved. My score remains unchanged.

---

### Official Review · Reviewer_Cwjo · 2024-11-05

**Soundness:** 2
**Presentation:** 2
**Contribution:** 3
**Rating:** 5
**Confidence:** 3

**Summary:**

This work studies the usage of non parametric two-sample tests for membership inference attacks. These enable set-based membership inference attacks with a much stronger performance than existing per-sample membership inference attacks. Empirical results are conducted over standard image datasets and include a text dataset and a tabular dataset. The results show an attack that is significantly stronger than existing MIAs.

Edit: Raised score after rebuttal to 5, but still leaning reject (perhaps, more a 4).

**Strengths:**

- Strong empirical results indicating perfect TPR at low FPR.

- Generic protocol that can accommodate various input statistics.

- Motivation is clear and the work is important.

- The organization is relatively clear.

**Weaknesses:**

- It is unclear what the performance improvement of representations is. Often, this threat model is not possible in practice, e.g., for the detection scenario mentioned.

- Almost all experiments are in the appendix, with not even a description in the main-body. This makes the paper extremely difficult to follow and assess.

- Similarly, nearly none of the empirical setup is described in the main-body, including necessary information to understand the results. For example, what are each of the attacks? The two-sample attack is a set-based attack, how is this being compared with prior non-set-based attacks?

- This work suggests using high-level representations such as those from the last layer. How are those obtained and where are these ablations?

- Theorem 1 appears convoluted. I have not understood what this brings to the paper. Though I understand that it is optimizing the kernel so as to achieve high test performance, why is this necessary? What is the baseline performance without doing this? How does this compare to the other methods for set-based inference?

- Set-based MIAs have already been studied, e.g., dataset inference [https://arxiv.org/abs/2104.10706], user inference attacks [https://arxiv.org/abs/2310.09266], and dataset inference for llms [https://arxiv.org/abs/2406.06443]. All of these show significant improvements over the standard per-sample MIAs. Comparisons against these are a must, as this work employs similar techniques.

[NIT] Though confidence-based attacks may be thwarted by defenses like memguard, these are easily thwarted by label-only attacks.

**Questions:**

Can the authors comment on the feasibility of using this for LLMs?

---

> ### Author Response · Authors · 2024-11-23
> **Response to Reviewer Cwjo [1/4]**
>
> Dear Reviewer Cwjo,
>
> We appreciate your feedback on our submission. We would like to address your concerns and clarify any misunderstandings regarding our paper as follows.
>
> > *Misunderstanding 1: It is unclear what the performance improvement of representations is. Often, this threat model is not possible in practice, e.g., for the detection scenario mentioned. This work suggests using high-level representations such as those from the last layer. How are those obtained and where are these ablations?*
>
> **Clarification 1:** This appears to be a misunderstanding and omission regarding the provided knowledge assumptions. Our method can be used in practical scenarios including **privacy breach detection** (Appendix B), **privacy auditing in machine unlearning** (Appendix B), **dataset inference** [1,2], and **user inference** [3]. Compared to previous membership inference methods [1–7], our knowledge assumption is more relaxed because we can choose different high-level representation-based statistics according to different practical scenarios, such as loss, entropy, likelihood ratio in non-LLM scenarios, and zlib_ratio and perplexity (ppl) in LLM scenarios, or combine with other methods such as the weighted MIA features in [2].
>
> See details in **Appendix E: Discussion on the Use of High-Level Representations**. Our framework is adaptable to diverse high-level representations, including confidence scores or penultimate-layer embeddings. **This flexibility is critical in adversarial settings where certain information (e.g., true labels) may be unavailable or obfuscated. By leveraging multiple high-level features, our HR-MMD test ensures robust inference capabilities even under adversarial conditions (e.g., MemGuard).**
>
> In summary, compared to previous membership inference methods, our method is a **high-representation-based deep kernel (HR-MMD)**. **We provide more choices and can be combined with state-of-the-art methods, such as [2], by replacing the t-test with our deep-kernel based MMD test. By selecting high-level representations according to practical scenario assumptions, we remain consistent with previous methods or maintain effectiveness when previous methods fail. No additional assumptions are required.**
>
> > *Misunderstanding 2: Almost all experiments are in the appendix, with not even a description in the main body. This makes the paper extremely difficult to follow and assess. Similarly, nearly none of the empirical setup is described in the main body, including necessary information to understand the results. For example, what are each of the attacks? The two-sample attack is a set-based attack; how is this being compared with prior non-set-based attacks?*
>
> **Clarification 2:** This appears to be a misunderstanding and omission regarding the provided experimental details. In the main body of the paper, we referenced the detailed experimental setups and their locations in the Appendices. For instance:
>
> - **Figure 1 caption (line 69):**
> - **Figure 3 caption (line 520):**
> - **Section 6 (line 473):**
>
> We hope you understand that to focus on the core contributions within the page limit, we had to place extensive experimental details in the Appendix G.6 Implementation Details of the Comparison with Traditional Two-sample Tests and Appendix G.7 Implementation Details of the Comparison with Traditional MIAs. The details about our baselines are in Appendix G.3: Baselines. The details about our experimental results are in Appendix H: Supplementary Experimental Results. The Appendices provide comprehensive descriptions of our experimental setups and evaluation criteria.
>
> **Summary of Our Method's Superiority:**
>
> Our method achieves state-of-the-art (SOTA) performance compared to two categories of baselines:
>
> 1. **Set-based MIAs:** We compare our HR-MMD test with existing set-based membership inference attacks (MIAs), including cutting-edge dataset inference methods and adaptations of traditional MIAs [1-6]. By using the rejection rate as an evaluation metric and our proposed HR-MMD test as the detection method (as described in **Algorithm 2**), we provide more accurate and reliable results than previous methods relying on heuristic metrics. This demonstrates our method's superior performance and significant contributions to the field.
>
> 2. **Statistical Baselines:** We outperform traditional two-sample tests, which cannot effectively perform membership inference for privacy breach detection. Our method offers superior discriminative power in detecting membership.
>
> **Our Method vs. Existing Methods:**
>
> - **Our Method:** Operates at the **set level**, making decisions on whether a given dataset S_Y contains members from the training data.
>
> - **Existing traditional MIA Methods [3-6]:** Often operate at the **individual data point level**, making membership decisions for each sample.

---

> ### Author Response · Authors · 2024-11-23
> **Response to Reviewer Cwjo [2/4]**
>
> To ensure fair comparisons, we adapted the evaluation metrics and experimental setups accordingly and provided detailed explanations in the Appendices. Specifically:
>
> **Evaluation Metrics:**
>
> We have incorporated a comprehensive explanation of the evaluation metrics and ensured that the comparison with baseline methods is consistent and fair. The metrics are defined as follows:
>
> ###  **Implementation Details of the Comparison with Traditional Two-sample Tests (Appendix G.6):**
>
>   - **Training Set (S_T)**: We use a training set size of 25,000 for CIFAR-10, CIFAR-100, and SVHN, and 90,000 for CINIC-10.
>
>   - **Non-Training Set (S_non))**: We assume the evaluator has a non-training set S_non, each containing 5,000 images from the respective datasets. Subsets S_X^tr and S_X^te are selected from S_{non}.
>
>   - **Input Test Dataset S_Y**: We fix the size of the input test dataset S_Y to 1,000 in our main experiments.
>
>   - **Sample Selection**:
>
>     - When comparing our method with baseline methods in terms of **test power**, the samples S_Y are randomly selected from the training subsets S_T of the datasets. In this context, S_T \cap S_X = \emptyset, S_T \cap S_Y = S_Y, and S_X \cap S_Y = \emptyset.
>
>     - When comparing in terms of **Type I error**, the samples S_Y are randomly selected from the testing subsets of the datasets.
>
>   - **Evaluation Procedure**:
>
>     - In **Algorithm 1**, for the input test dataset S_Y (size 1,000) and the non-training set S_X (size 1,000), we select subsets containing 500 images each for S_X^{tr} and S_Y^{tr} to train our deep kernel.
>
>     - The evaluation is conducted once on S_X^{te} and S_Y^{te} (each size 500), disjoint from S_X^{tr} and S_Y^{tr}.
>
>     - We compare the p-value obtained from Algorithm 1 with the significance level alpha to determine whether to reject H_0.
>
>     - To obtain a reliable assessment, the evaluation is repeated 100 times on different random subsets S_X^{te}.
>
> - **Type I Error:** The probability of rejecting the null hypothesis H_0 (no members present) when it is true.
>
> - **Test power:** The probability of rejecting H_0 when the alternative hypothesis H_1 is true (members are present).
>
> Algorithm 2 describes the complete flow of this process. When the inputted samples S_Y are non-members, the rejection rate outputted is the type I error. When the inputted samples S_Y are members, the rejection rate outputted is the test power. We repeat this process 10 times and report the mean test power of each test.
>
> ###  **Implementation Details of the Comparison with Traditional MIAs (Appendix G.7):**
>
> For baseline methods that output decisions for individual data points, we aggregated their outputs to the dataset level by considering average statistics (e.g., loss) and setting thresholds to make set-level decisions, similar to the approach used in set-based attacks like [1].
>
>   - We conducted comparisons with four typical MIA attacks:
>
>     - **Loss Attack** [3]
>
>     - **Entropy Attack** [4]
>
>     - **Attack R** [5]
>
>     - **Likelihood Ratio Attack (LiRA)** [6]
>
>   - **Aggregation Strategy**:
>
>     - For these methods, which output individual data point decisions, we aggregated their outputs to the dataset level by computing average statistics over S_Y and setting thresholds to make set-level decisions.
>
>   - **Evaluation Procedure**:
>
> We randomly selected 1,000 datasets S_Y containing training members and 1,000 datasets S_Y without training members. For traditional MIAs, when the average statistics (like loss) over S_Y were no greater than a threshold, we considered that S_Y contained members. Conversely, when the average statistics were greater than the threshold, we considered that S_Y contained no members.
>
> For our HR-MMD test, when the rejection rate of one dataset S_Y was no greater than 0.065, we considered that S_Y contained no members. When the rejection rate was greater than 0.065, we considered that S_Y contained members. Experimental results show that our HR-MMD test significantly outperforms prior MIAs.
>
> - **True Positive Rate (TPR):** The proportion of datasets correctly identified as containing training members.
>
> - **False Positive Rate (FPR):** The proportion of datasets incorrectly identified as containing training members when they do not.
>
> By adapting the evaluation metrics and experimental setups, we ensure that the comparison is appropriate and fair.

---

> ### Author Response · Authors · 2024-11-23
> **Response to Reviewer Cwjo [3/4]**
>
> > *Q1: Theorem 1 appears convoluted. I have not understood what this brings to the paper. Though I understand that it is optimizing the kernel so as to achieve high test performance, why is this necessary? What is the baseline performance without doing this? How does this compare to the other methods for set-based inference?*
>
> **A1:** The key to understanding this lies in the principles of the Maximum Mean Discrepancy (MMD) method. In MMD, we aim to measure the distance between two distributions P and Q in a reproducing kernel Hilbert space (RKHS).
>
> A fixed kernel (e.g., a standard Gaussian kernel) may not capture the maximum discrepancy between the distributions, especially in high-dimensional or complex data spaces. Therefore, we employ a learning approach to optimize the parameters of the kernel k_w. The maximization of J enhances the test power by optimizing the kernel parameters w in the HR-MMD to maximize the discrepancy between the member and non-member distributions.
>
> By learning the kernel parameters w that maximize J (related to the estimated HR-MMD value on the training subsets S_X^{tr} and S_Y^{tr}), we effectively find the most discriminative kernel. This optimized kernel can better capture the differences between the member and non-member distributions, thereby improving the test's ability to distinguish between them.
>
> This process is analogous to training a machine learning model to maximize a target objective like accuracy on training data (minimizing the training loss). By optimizing J, we learn the optimized kernel k_w that can enhance the test power of our HR-MMD test. The theoretical analyses for this approach are provided in Appendix C, where we analyze the asymptotics of HR-MMD and demonstrate how maximizing J leads to increased test power.
>
>
> In the main paper, **Figure 3(f)** is our ablation study. For detailed analysis, please refer to **Appendix H.5: Ablation Study**. Our findings indicate that our deep kernel approach outperforms the non-deep kernel methods (CS-MMD (or HR1-MMD) vs. HR+MMD-G (or HR-T)). Note that HR-T are high-representation-based t-tests; t-tests are part of [2].

---

> ### Author Response · Authors · 2024-11-23
> **Response to Reviewer Cwjo [4/4]**
>
> >*Q4: Set-based MIAs have already been studied, e.g., dataset inference [1][2], user inference attacks [3]. Can the authors comment on the feasibility of using this for LLMs?*
>
> **A4:** We enriched our Appendix A related works and Appendix B practical scenarios by incorporating discussions on dataset inference [1][2] and user inference [3], and extended our evaluation to include tasks involving Large Language Models (LLMs). Detailed experimental setups and results are presented in the Appendix I: Discussion of Applications in Large Language Model (LLM) Scenarios. Specifically, we made the following enhancements:
>
> - **Integration with State-of-the-Art Methods:** Combined HR-MMD with the methodology from [1] for dataset inference in LLM contexts, replacing the traditional t-test with our approach.
> - **Performance Improvements:** Demonstrated HR-MMD’s superior performance in test power and robustness compared to baseline methods, particularly in challenging scenarios with small datasets.
> - **Key Results:** HR-MMD effectively captures distributional differences between training and non-training sets, consistently outperforming alternatives across varied test settings.
>
> While the approaches in cutting-edge studies NeurIPS 2024 [2] and EMNLP 2024 [3] introduce innovative strategies for LLM scenarios, they are limited in non-LLM contexts. For instance, [2] relies solely on a T-test to compute p-values for statistical significance, which is inherently weaker than our HR-MMD test in terms of discriminative power. Similarly, [3] aligns with the set-based Attack-R method [4], which is already our baseline.
>
> These extensions provide strong evidence of our method’s broader relevance, substantial contributions, and state-of-the-art (SOTA) performance, reinforcing its value for top-tier conferences such as ICLR.
>
> We have provided more extensive experiments on different learning tasks, models, and datasets as follows. **Our experiments evaluate the effectiveness of our methods across four image datasets: CIFAR-10, CIFAR-100, SVHN, and CINIC-10.** We also demonstrate the capacity for extending to a tabular dataset **Purchase-100** and a text dataset **IMDB**. As suggested, we have supplemented applications on large model data. **Our current work is very thorough; we have further added comparisons across different set sizes, different mixup ratios, different model architectures, and different defense methods. Compared to previous MIA-based methods, our comparisons are comprehensive, convincingly demonstrating our SOTA performance.** Compared with existing MIAs, our HR-MMD test demonstrates broader applicability across various machine learning domains, proving its versatility and effectiveness.
>
> ---
>
> We are immensely grateful for your time and expertise. All the explanations in our rebuttal have been incorporated into the current version to help readers better understand our work. We are keen to hear any further thoughts or concerns you might have. **We believe that our contribution to the field and its significance may have been underestimated. Your insights are crucial to us, and if your concerns have been alleviated, we hope you might consider adjusting your review score accordingly.**
>
> **References:**
>
> [1] Dataset Inference: Ownership Resolution in Machine Learning. In ICLR 2021.
>
> [2] LLM Dataset Inference: Did You Train on My Dataset? In NeurIPS 2024.
>
> [3] User Inference Attacks on Large Language Models. In EMNLP 2024.
>
> [4] Privacy Risk in Machine Learning: Analyzing the Connection to Overfitting. In IEEE CS Security and Privacy Workshops 2018.
>
> [5] Auditing Data Provenance in Text-Generation Models. In KDD 2019.
>
> [6] Enhanced Membership Inference Attacks Against Machine Learning Models. In CCS 2022.
>
> [7] Membership Inference Attacks From First Principles. In SP 2022.

---

> ### Author Response · Authors · 2024-11-25
> **Looking Forward to Your Reply and Hoping for Your Stronger Support**
>
> Dear Reviewer Cwjo,
>
> We greatly appreciate your thoughtful feedback and insightful suggestions on our submission. We have carefully addressed your initial concerns and further comments, and hope that you are pleased with the progress we've made. Should you have any remaining questions or concerns, we are more than happy to discuss them with you through the openreview system. Additionally, we welcome any new suggestions or comments you might have.
>
> Following your advice, we have incorporated the explanations provided in our rebuttal into the current version of our paper to ensure that future readers have a clearer understanding of our work. We believe these additions will significantly enhance the clarity and impact of our paper.
>
> We believe that our contribution to the field and its significance make our submission highly competitive for ICLR. Your opinion is crucial to us and we welcome any additional feedback or suggestions for datasets and experiments that could enhance our manuscript. We are committed to further improving the clarity and presentation of our paper based on your suggestions. We would be immensely grateful if you could provide us with a more competitive score.
>
> Once again, we sincerely thank you for your time, effort, and invaluable contribution to the advancement of the field. We eagerly look forward to your positive response.

---

> > ### Comment · Reviewer_Cwjo · 2024-12-03
> > **Read rebuttal; More positive, but still believe this paper requires more improvement.**
> >
> > Thank you for you response. These have improved and/or clarified the paper and I am leaning more positive. Thus, I have raised my score accordingly.
> >
> > However, I note that I still lean towards rejection as I believe the organization of the paper could be significantly improved and in its current state hinders the clarity substantially.

---

### Official Review · Reviewer_i1pv · 2024-11-06

**Soundness:** 3
**Presentation:** 3
**Contribution:** 3
**Rating:** 5
**Confidence:** 3

**Summary:**

This study introduces a method for evaluating the occurrence of membership inference using a two-sample test on a set rather than individual data points, taking into account the reliability of membership inference evaluation. The proposed method is experimentally shown to be highly sensitive to the distinction between data used for learning and data not used for learning.

**Strengths:**

- Introduced statistical inference into the membership inference and established statistical significance guarantee on the inference of membership
- Proposed a deep kernel function that is useful for membership inference for deep classification models and analyzed the test statistics and asymptotic behavior of the test statistics
- Evaluated performance through experimental evaluation

**Weaknesses:**

- The explanation of the experimental setup and evaluation criteria is insufficient, so it is difficult to determine whether the experimental results show the superiority of the proposed method.

**Questions:**

Line 429: Could you explain why the maximization of J improves the power? I could not understand from the explanation in the manuscript.

Line 443: If parameters of k_w are optimized under the setting of the alternative hypothesis, can we say this is also appropriate under H_0?

Sec 6:
- According to Appendix G.6, the ratio of training data to evaluation data is consistently set to 10% in the experimental results, which seems unreasonable. This ratio must be changed to evaluate the proposed and comparison methods' performance.

- A clear explanation of the evaluation metric is necessary. The proposed method outputs decisions for the given dataset, while some existing methods output decisions for each data point. Since I could not find an explanation on how TPR, type 1 error, and type 2 error are evaluated, it is difficult to determine whether the proposed method was compared correctly with the comparison method.

---

> ### Author Response · Authors · 2024-11-23
> **Response to Reviewer i1pv [1/3]**
>
> Dear Reviewer i1pv
>
> We appreciate your review and valuable feedback on our submission. We would like to address your concerns and clarify any misunderstandings regarding our paper as follows.
>
> ---
>
> * Misunderstanding:*
>
> > *The explanation of the experimental setup and evaluation criteria is insufficient, making it difficult to determine whether the experimental results show the superiority of the proposed method. Additionally, a clear explanation of the evaluation metrics is necessary. The proposed method outputs decisions for the given dataset, while some existing methods output decisions for each data point. Since I could not find an explanation on how TPR, Type I error, and Type II error are evaluated, it is difficult to determine whether the proposed method was compared correctly with the comparison methods.*
>
>
> ** Clarification:**
> *This appears to be a misunderstanding due to the detailed experimental setups and evaluation criteria being located in the Appendix G6 and G7.* In the main body of the paper, we referenced the detailed experimental setups and their locations in the Appendices to balance the core content within the limited page count. For instance:
>
> - **Figure 1 caption (line 69):**
> - **Figure 3 caption (line 520):**
> - **Section 6 (line 473):**
>
> We hope you understand that to focus on the core contributions within the page limit, we had to place extensive experimental details in the Appendix G.6 Implementation Details of the Comparison with Traditional Two-sample Tests and Appendix G.7 Implementation Details of the Comparison with Traditional MIAs. The Appendices provide comprehensive descriptions of our experimental setups and evaluation criteria.
>
> **Summary of Our Method's Superiority:**
>
> Our method achieves state-of-the-art (SOTA) performance compared to two categories of baselines:
>
> 1. **Set-based MIAs:** We compare our HR-MMD test with existing set-based membership inference attacks (MIAs), including cutting-edge dataset inference methods and adaptations of traditional MIAs [1-6]. By using the rejection rate as an evaluation metric and our proposed HR-MMD test as the detection method (as described in **Algorithm 2**), we provide more accurate and reliable results than previous methods relying on heuristic metrics. This demonstrates our method's superior performance and significant contributions to the field.
>
> 2. **Statistical Baselines:** We outperform traditional two-sample tests, which cannot effectively perform membership inference for privacy breach detection. Our method offers superior discriminative power in detecting membership.
>
> **Our Method vs. Existing Methods:**
>
> - **Our Method:** Operates at the **set level**, making decisions on whether a given dataset S_Y contains members from the training data.
>
> - **Existing traditional MIA Methods [3-6]:** Often operate at the **individual data point level**, making membership decisions for each sample.

---

> > ### Comment · Reviewer_i1pv · 2024-11-27
> > **Response to the rebuttal**
> >
> > Thanks for the clarification.
> > When reviewing the manuscript, I checked  Appendix G6 and G7. Still, I could not find an explanation of how TPR, Type I errors, and Type II errors are evaluated with the two different settings: set-based MIAs and statistical baselines.
> >
> > For example, could you explain why the comparison in Fig 1 (a) is fair?
> > Do all comparison methods in  Fig 1 (a) follow the Set-based MIA scheme?

---

> > > ### Author Response · Authors · 2024-11-27
> > > **Response to feedback [2/2]**
> > >
> > > ### Statistical Baselines Comparison
> > >
> > > Next, we introduce the comparison with statistical baselines. Here, we follow the well-established setting in the **two-sample tests** domain, as detailed in **Appendix G.6** ("Implementation Details of the Comparison with Traditional Two-sample Tests," Page 30, around line 1608). We select a single set S_Y, and different statistical baselines output a rejection rate for this set. Specifically, in **Algorithm 2**, we select 100 non-training sets S_X from S_non. For each pair of S_X and the fixed S_Y, we perform an **HR-MMD-based permutation test in Algorithm 2.** And then we obtain a rejection rate on 100 these pairs.
> > >
> > > - When the input set S_Y contains no **members**, the rejection rate outputted is referred to as the **Type I error**.
> > > - When the input set S_Y contains **members**, the rejection rate outputted is referred to as the **Test power**, which is 1 - **Type II error**.
> > >
> > > To account for any randomness in the selected set S_Y, we repeat this process 10 times, selecting 10 different sets S_Y, and report the mean test power of each test. The above setting follows previous two-sample test setting such as Learning Deep Kernels for Two-Sample Testing. In ICML 2020
> > >
> > > We hope this clears up any confusion regarding the fairness of the comparison in **Fig. 1 (a)** and provides a more detailed explanation of how **TPR**, **Type I errors**, and **Test power** are evaluated. Please let us know if you have any further questions or need additional clarifications. Once again, we sincerely thank you for your time, effort, and invaluable contribution to the advancement of the field. We eagerly look forward to your positive response.

---

> ### Author Response · Authors · 2024-11-23
> **Response to Reviewer i1pv [2/3]**
>
> To ensure fair comparisons, we adapted the evaluation metrics and experimental setups accordingly and provided detailed explanations in the Appendices. Specifically:
>
> **Evaluation Metrics:**
>
> We have incorporated a comprehensive explanation of the evaluation metrics and ensured that the comparison with baseline methods is consistent and fair. The metrics are defined as follows:
>
> ###  **Implementation Details of the Comparison with Traditional Two-sample Tests (Appendix G.6):**
>
>   - **Training Set (S_T)**: We use a training set size of 25,000 for CIFAR-10, CIFAR-100, and SVHN, and 90,000 for CINIC-10.
>
>   - **Non-Training Set (S_non))**: We assume the evaluator has a non-training set S_non, each containing 5,000 images from the respective datasets. Subsets S_X^tr and S_X^te are selected from S_{non}.
>
>   - **Input Test Dataset S_Y**: We fix the size of the input test dataset S_Y to 1,000 in our main experiments.
>
>   - **Sample Selection**:
>
>     - When comparing our method with baseline methods in terms of **test power**, the samples S_Y are randomly selected from the training subsets S_T of the datasets. In this context, S_T \cap S_X = \emptyset, S_T \cap S_Y = S_Y, and S_X \cap S_Y = \emptyset.
>
>     - When comparing in terms of **Type I error**, the samples S_Y are randomly selected from the testing subsets of the datasets.
>
>   - **Evaluation Procedure**:
>
>     - In **Algorithm 1**, for the input test dataset S_Y (size 1,000) and the non-training set S_X (size 1,000), we select subsets containing 500 images each for S_X^{tr} and S_Y^{tr} to train our deep kernel.
>
>     - The evaluation is conducted once on S_X^{te} and S_Y^{te} (each size 500), disjoint from S_X^{tr} and S_Y^{tr}.
>
>     - We compare the p-value obtained from Algorithm 1 with the significance level alpha to determine whether to reject H_0.
>
>     - To obtain a reliable assessment, the evaluation is repeated 100 times on different random subsets S_X^{te}.
>
> - **Type I Error:** The probability of rejecting the null hypothesis H_0 (no members present) when it is true.
>
> - **Test power:** The probability of rejecting H_0 when the alternative hypothesis H_1 is true (members are present).
>
> Algorithm 2 describes the complete flow of this process. When the inputted samples S_Y are non-members, the rejection rate outputted is the type I error. When the inputted samples S_Y are members, the rejection rate outputted is the test power. We repeat this process 10 times and report the mean test power of each test.
>
> ###  **Implementation Details of the Comparison with Traditional MIAs (Appendix G.7):**
>
> For baseline methods that output decisions for individual data points, we aggregated their outputs to the dataset level by considering average statistics (e.g., loss) and setting thresholds to make set-level decisions, similar to the approach used in set-based attacks like [1].
>
>   - We conducted comparisons with four typical MIA attacks:
>
>     - **Loss Attack** [3]
>
>     - **Entropy Attack** [4]
>
>     - **Attack R** [5]
>
>     - **Likelihood Ratio Attack (LiRA)** [6]
>
>   - **Aggregation Strategy**:
>
>     - For these methods, which output individual data point decisions, we aggregated their outputs to the dataset level by computing average statistics over S_Y and setting thresholds to make set-level decisions.
>
>   - **Evaluation Procedure**:
>
> We randomly selected 1,000 datasets S_Y containing training members and 1,000 datasets S_Y without training members. For traditional MIAs, when the average statistics (like loss) over S_Y were no greater than a threshold, we considered that S_Y contained members. Conversely, when the average statistics were greater than the threshold, we considered that S_Y contained no members.
>
> For our HR-MMD test, when the rejection rate of one dataset S_Y was no greater than 0.065, we considered that S_Y contained no members. When the rejection rate was greater than 0.065, we considered that S_Y contained members. Experimental results show that our HR-MMD test significantly outperforms prior MIAs.
>
> - **True Positive Rate (TPR):** The proportion of datasets correctly identified as containing training members.
>
> - **False Positive Rate (FPR):** The proportion of datasets incorrectly identified as containing training members when they do not.
>
> By adapting the evaluation metrics and experimental setups, we ensure that the comparison is appropriate and fair.

---

> ### Author Response · Authors · 2024-11-23
> **Response to Reviewer i1pv [3/3]**
>
> >*Q1: Could you explain why the maximization of J improves the test power?*
>
> **A1:** The key to understanding this lies in the principles of the Maximum Mean Discrepancy (MMD) method. In MMD, we aim to measure the distance between two distributions P and Q in a reproducing kernel Hilbert space (RKHS).
>
> A fixed kernel (e.g., a standard Gaussian kernel) may not capture the maximum discrepancy between the distributions, especially in high-dimensional or complex data spaces. Therefore, we employ a learning approach to optimize the parameters of the kernel k_w. The maximization of J enhances the test power by optimizing the kernel parameters w in the HR-MMD to maximize the discrepancy between the member and non-member distributions.
>
> By learning the kernel parameters w that maximize J (related to the estimated HR-MMD value on the training subsets S_X^{tr} and S_Y^{tr}), we effectively find the most discriminative kernel. This optimized kernel can better capture the differences between the member and non-member distributions, thereby improving the test's ability to distinguish between them.
>
> This process is analogous to training a machine learning model to maximize a target objective like accuracy on training data (minimizing the training loss). By optimizing J, we learn the optimized kernel k_w that can enhance the test power of our HR-MMD test. The theoretical analyses for this approach are provided in Appendix C, where we analyze the asymptotics of HR-MMD and demonstrate how maximizing J leads to increased test power.
>
> ---
>
> >*Q2: If parameters of k_w are optimized under the setting of the alternative hypothesis, can we say this is also appropriate under H_0?*
>
> **A2:**
>
>  Yes, optimizing the parameters of k_w under the alternative hypothesis does not invalidate the test under the null hypothesis H_0. The properties of HR-MMD ensure that under H_0 (when the member and non-member distributions are the same), the expected HR-MMD value remains low regardless of the kernel choice.
>
> By maximizing J under the alternative hypothesis, we enhance the test's ability to detect differences when they exist (i.e., when H_1 is true). However, under H_0, the HR-MMD value remains low due to the inherent characteristics of the HR-MMD metric. This approach ensures that the test maintains an ideal Type I error rate under H_0 while improving its power under H_1.
>
> ---
>
> >*Q3: According to Appendix G.6, the ratio of training data to evaluation data is consistently set to 10% in the experimental results, which seems unreasonable. This ratio must be changed to evaluate the proposed and comparison methods' performance.*
>
> **A3:** Thank you for your suggestion. To address this concern, we extended our experimental framework to include **detailed mixup membership scenarios**, where the input test data comprised a mix of member and non-member samples with varying proportions (mixup ratios ranging from 40% to 90%). This provides a broader evaluation of our method's performance under diverse data distributions. Specifically, we:
>
> 1. **Tested Across Defenses:**
>    - Conducted evaluations using membership inference defense strategies, focusing on training model with L1 regularization.
>
> 2. **Demonstrated Consistency:**
>    - Across all mixup proportions, our HR-MMD test exhibited **consistent and robust performance**, outperforming baseline methods and maintaining its state-of-the-art (SOTA) status.
>
> These results underscore our method’s reliability in handling complex data distributions, an essential capability for practical deployment in privacy-sensitive applications. Detailed experimental setups and results for this task are included in the Appendix J: Discussion of Extensive Mixup Membership Scenarios.
>
> We are immensely grateful for your time and expertise. All the explanations in our rebuttal have been incorporated into the revised version to help readers better understand our work. We welcome any further thoughts or concerns you might have. We believe that our contribution to the field and its significance may have been underestimated. Your insights are crucial to us, and if your concerns have been alleviated, we hope you might consider adjusting your review score accordingly.
>
> **References:**
>
> [1] LLM Dataset Inference: Did You Train on My Dataset? In NeurIPS 2024.
>
> [2] User Inference Attacks on Large Language Models. In EMNLP 2024.
>
> [3] Privacy Risk in Machine Learning: Analyzing the Connection to Overfitting. In *IEEE CS Security and Privacy Workshops* 2018.
>
> [4] Auditing Data Provenance in Text-Generation Models. In KDD 2019*.
>
> [5] Enhanced Membership Inference Attacks Against Machine Learning Models. In CCS 2022.
>
> [6] Membership Inference Attacks From First Principles. In SP 2022*.

---

> ### Author Response · Authors · 2024-11-25
> **Looking Forward to Your Reply and Hoping for Your Stronger Support**
>
> Dear Reviewer i1pv,
>
> We greatly appreciate your thoughtful feedback and insightful suggestions on our submission. We have carefully addressed your initial concerns and further comments, and hope that you are pleased with the progress we've made. Should you have any remaining questions or concerns, we are more than happy to discuss them with you through the openreview system. Additionally, we welcome any new suggestions or comments you might have.
>
> Following your advice, we have incorporated the explanations provided in our rebuttal into the current version of our paper to ensure that future readers have a clearer understanding of our work. We believe these additions will significantly enhance the clarity and impact of our paper.
>
> We believe that our contribution to the field and its significance make our submission highly competitive for ICLR. Your opinion is crucial to us and we welcome any additional feedback or suggestions for datasets and experiments that could enhance our manuscript. We are committed to further improving the clarity and presentation of our paper based on your suggestions. We would be immensely grateful if you could provide us with a more competitive score.
>
> Once again, we sincerely thank you for your time, effort, and invaluable contribution to the advancement of the field. We eagerly look forward to your positive response.

---

> ### Author Response · Authors · 2024-11-27
> **Response to feedback [1/2]**
>
> Dear Reviewer i1pv,
>
> Thank you for your feedback. To summarize, the comparison in **Fig. 1 (a)** is fair because:
> - **All methods are evaluated under the same set-based MIA scheme**.
> - **We ensure a consistent experimental framework**, where methods are tested using the same evaluation datasets and knowledge assumptions.
>
> We divide the comparison into two settings because our technical contribution lies in being the first to introduce a non-parametric two-sample test for membership inference. Both **non-parametric two-sample tests** and **membership inference** are well-established fields, but the evaluation details differ. Therefore, we need to compare the methods under two distinct settings to ensure that both traditional MIAs and statistical baselines are assessed within their respective, well-established frameworks. This approach allows for a rigorous and balanced evaluation of our superior performance.
>
> ### Clarifying the Evaluation Process
>
> To address your query, we confirm that **all comparison methods in Fig. 1 (a)** follow the **Set-based MIA scheme**. The experiments are conducted using identical evaluation datasets across all methods.
>
> We would like to clarify the evaluation process in more detail:
>
> 1. **Traditional individual-based MIA methods (Not relevant to Fig. 1 (a))**:
>    In traditional MIAs, evaluations are conducted on a balanced dataset, including \( N \) samples with known ground-truth membership (i.e., "member" with label 1) and \( N \) samples with known ground-truth membership (i.e., "non-member" with label 0). For each sample, the MIA methods calculate statistics (such as loss, entropy, or likelihood ratio) and compare them with a preset threshold to predict the membership label (1 or 0). By comparing the predicted labels with the ground truth, we can compute metrics like the **True Positive Rate (TPR)** and **False Positive Rate (FPR)**.
>
> 2. **Set-based MIA evaluations in Fig. 1 (a)**:
>    In contrast, the evaluation is conducted on a series of sets instead of individual samples. We selected 2,000 sets in total: 1,000 sets containing members (e.g., 10% are members, as shown in Fig. 1 (a)) and 1,000 sets containing no members.
>
>    For the subsets that do not contain members, we assign the ground truth membership label as "non-member set" with label 0. For subsets that do contain members, we assign the ground truth membership label as "member-containing set" with label 1. The Set-based MIA method then predicts whether each set contains members. The **TPR** and **FPR** are calculated by comparing the predicted membership labels with the actual ground truth at the set level.
>
>    To ensure clarity, all methods compared in Fig. 1 (a) are tested on the same evaluation datasets, with consistent experimental conditions such as the same subset sizes and ground truth assignments for "member-containing" and "non-member" sets.
>
>    For the implementation of Set-based MIA evaluations in Fig. 1 (a), as detailed in **Appendix G.7** ("Implementation Details of the Comparison with Traditional MIAs," Page 31, around line 1633), we aggregate the outputs of each subset by calculating average statistics such as **loss**, **entropy**, or **likelihood ratio** over the input test dataset S_Y. These aggregated statistics are then used to make set-level decisions by applying thresholds. This approach follows established practices from prior research, such as in the **User Inference Attacks on Large Language Models** paper (EMNLP 2024), where reference models are used to compute average statistics for each subset. In **Fig. 1 (a)**, **Attack-R** represents this baseline, where the aggregated statistics guide the decision-making process.
>
> ### HR-MMD Test vs. Traditional set-based MIAs
>
> Compared to traditional Set-based Membership Inference Attacks, our **HR-MMD test** outputs a rejection rate for each set, which serves as our test statistic. This is similar to (average loss, average entropy, average likelihood ratio), but provides a different and more effective approach.

---

### Author Response · Authors · 2024-11-23
**General Response by Authors [1/3 Summary of Updates]**

We extend our deepest gratitude to all the reviewers for their insightful comments on our submission. We are pleased to report that we have addressed all initial concerns raised, as detailed in our responses to each reviewer.

To clarify potential misunderstandings regarding our method and the overlooked content already present in our submission, we have provided specific references to the relevant sections of the paper to assist reviewers in their evaluation. We have also elaborated on the issues raised and ensured that our responses are comprehensive and detailed.

Additionally, we have uploaded the revised manuscript, which incorporates the following key revisions:

---

### 1. Appendix I: Discussion of Applications in Large Language Model (LLM) Scenarios
- **Integration with State-of-the-Art Methods:** Combined HR-MMD with the methodology from [1] for dataset inference in LLM contexts, replacing the traditional t-test with our approach.
- **Performance Improvements:** Demonstrated HR-MMD’s superior performance in test power and robustness compared to baseline methods, particularly in challenging scenarios with small datasets.
- **Key Results:** HR-MMD effectively captures distributional differences between training and non-training sets, consistently outperforming alternatives across varied test settings.

---

### 2. Appendix J: Discussion of Evaluations under Membership Inference Defenses
- **Defense Mechanisms Tested:**
  1. Knowledge Distillation
  2. L1 Regularization
  3. Differential Privacy (DP-SGD)
- **Findings:** HR-MMD maintained high detection rates and demonstrated resilience under all defense mechanisms, consistently surpassing baseline methods in robustness and adaptability.
- **Significance:** Validates HR-MMD as a reliable tool in adversarial and privacy-constrained environments.
---

### 3. Appendix J: Discussion of Extensive Mixup Membership Scenarios
- **Experimental Design:** Evaluated mixed membership scenarios with varying proportions of training and non-training members in input datasets.
- **Results:** HR-MMD exhibited strong sensitivity and reliability across all mixup ratios, significantly outperforming baseline methods.
- **Robustness Under Defense:** Maintained performance under L1 Regularization, showcasing its robustness in complex, real-world conditions.

---

### 4. Appendix L: Discussion on the Application for Fraction Estimation Task
- **Application:** Extended HR-MMD to estimate the fraction of training members within test datasets, a task requiring advanced statistical modeling.
- **Methodology:** Developed a robust approach incorporating calibration curves and subset sampling for nuanced privacy risk assessments.
- **Outcome:** Highlighted HR-MMD’s versatility and advanced applicability beyond traditional binary membership inference.

---

### 5. Appendix M: Discussion on Handling Out-of-Distribution Data
- **Two-Step Approach:**
  1. **Detection:** Used traditional two-sample tests to identify significant deviations from expected distributions.
  2. **Isolation:** Filtered OOD samples to focus on in-distribution subsets, enabling accurate membership inference.
- **Key Findings:** Mitigated the influence of OOD samples, ensuring reliable performance even with heterogeneous datasets.
- **Impact:** Reinforces HR-MMD’s utility for real-world applications where data is often diverse and unpredictable.

---

These updates emphasize HR-MMD’s versatility, robustness, and state-of-the-art performance across a variety of privacy-sensitive and challenging experimental conditions. Detailed results and discussions for each enhancement are included in the updated submission.

---

> ### Author Response · Authors · 2024-11-23
> **General Response by Authors [2/3 Methodological and Analytical Clarifications]**
>
> To provide greater clarity and address potential misunderstandings, we have elaborated on the theoretical underpinnings, innovations, and advantages of our HR-MMD test, setting it apart from existing methods in both concept and implementation.
>
> #### a. Principles and Advantages of HR-MMD:
>
> 1. **Existing Methods:**
>
>    - Traditional set-based MIAs rely heavily on heuristic metrics (e.g., average loss) and fixed thresholds without theoretical guarantees.
>
> 2. **Our HR-MMD Approach:**
>
>    - We propose a non-parametric two-sample test that employs a theoretically grounded framework. Unlike existing methods, in practical implementation, HR-MMD performs multiple hypothesis tests for the same input test data S_Y (as detailed in Algorithm 2 and Appendix) and calculates rejection rates as the primary evaluation metric. This multi-test approach mitigates the impact of sample-specific variations, ensuring reliable results even in complex scenarios.
>
> #### b. Key Methodological Innovations:
>
> 1. **Motivation and Theoretical Foundation:**
>
>    - Traditional methods analyze average metrics (e.g., average loss) for membership detection, motivated by the assumption that members have lower loss. In contrast, we analyze the high-level representation differences between member and non-member distributions, enabling a more robust and theoretically sound analysis.
>
> 2. **Integration of Alternative Representations:**
>
>    - Our framework is adaptable to diverse high-level representations, including confidence scores or penultimate-layer embeddings. This flexibility is critical in adversarial settings where certain information (e.g., true labels) may be unavailable or obfuscated. By leveraging multiple high-level features, our HR-MMD test ensures robust inference capabilities even under adversarial conditions (e.g., MemGuard \[[2]\]).
>
> 3. **Robustness Through Multiple Tests:**
>
>    Single test to compare p-value is inherently sensitive to sample-specific error, which can result in unreliable conclusions. Our method conducts multiple hypothesis tests, with rejection rates serving as the decisive metric. This reduces the uncertainty introduced by single-test methods and ensures consistent performance across varied data subsets.
>
> 4. **Deep Kernel Design:**
>
>    - HR-MMD innovatively learns a deep kernel optimized during training. This allows for greater adaptability and accuracy in detecting privacy risks.
>
> #### c. Theoretical Contributions and Analysis:
>
> - Our work is the first to introduce a non-parametric two-sample test to membership privacy evaluation. Theoretical analyses, provided in Appendix C, establish the asymptotic of HR-MMD, further reinforcing its validity.
>
> - The test achieves ideal results with 100% true positive rate (TPR) and 0% false positive rate (FPR), setting a new benchmark for statistical privacy breach detection.

---

> > ### Author Response · Authors · 2024-11-23
> > **General Response by Authors [3/3 Summary]**
> >
> > Our work introduces a novel framework for Membership Inference Attacks (MIAs) by leveraging a deep kernel-based non-parametric two-sample test, which significantly differs from all existing MIA methodologies. This approach allows us to systematically evaluate the privacy risks of machine learning models beyond traditional metrics, focusing on the test power and the ability to maintain an ideal type I error rate even in complex scenarios like mixed membership data.
> >
> > - **Our technical contribution lies in being the first to introduce a non-parametric two-sample test for membership inference.** We have developed a robust statistical methodology capable of addressing statistical membership inference for privacy breach detection, achieving 100% TPR (ideal test power) and 0% FPR (ideal type I error).
> >
> > - **Our main technical contribution is the novel High-level Representation-based deep kernel (HR-MMD) in Eq.(5).** And we provide theoretical analysis on the asymptotics of HR-MMD in the supplementary material (Appendix C).
> >
> > - While we acknowledge that two-sample testing is a well-established field, we emphasize that existing two-sample test methods cannot effectively solve the problem of statistical membership inference for privacy breach detection. In contrast, our proposed HR-MMD test achieves exceptional results with 100% TPR and 0% FPR, underscoring the significance of our work for the entire field.
> >
> > - The effectiveness of our HR-MMD stems from two aspects: the superior performance of high-level representation and the effectiveness of deep kernel design. Our utilization of high-level representation and high-level representation-based deep kernel in the context of MIAs distinguishes our work from existing literature.
> >
> > As advised by reviewers, we have provided more extensive experiments on different learning tasks, models, and datasets as follows. **Our experiments evaluate the effectiveness of our methods across four image datasets: CIFAR-10, CIFAR-100, SVHN, and CINIC-10.** We also demonstrate the capacity for extending to a tabular dataset Purchase-100 and a text dataset IMDB. As suggested by **Reviewer Cwjo**, **we have supplemented applications on LLMs. Our current work is thorough; we have further added comparisons across different set sizes, different mixup ratios, different model architectures, and different defense methods. Compared to previous MIA-based methods, our comparisons are very comprehensive, convincingly demonstrating our SOTA performance.** Compared with existing MIAs, our HR-MMD test demonstrates broader applicability across various machine learning domains, proving its versatility and effectiveness.
> >
> > We are committed to further improving the clarity and presentation of our paper based on your suggestions. We welcome any additional feedback or suggestions for datasets and experiments that could enhance our manuscript.
> >
> > In conclusion, our contribution lies not only in proposing a novel MIA framework but also in demonstrating its efficacy through rigorous experimentation and comparison with existing methods. Our work offers a fresh perspective on evaluating privacy risks in machine learning models, opening avenues for further research and development in this critical area. We hope that our responses have addressed your concerns and clarified the significant contributions of our study.
> >
> > We are immensely grateful for your time and expertise. All the explanations in our rebuttal have been incorporated into current version to help readers better understand our work. We are keen to hear any further thoughts or concerns you might have. We believe that our contribution to the field and its significance may have been underestimated. Your insights are crucial to us, and if your concerns have been alleviated, we hope you might consider adjusting your review score accordingly.
> >
> > ---
> >
> > **References:**
> >
> > [1] LLM Dataset Inference: Did You Train on My Dataset? In NeurIPS 2024.
> >
> > [2] MemGuard: Defending Against Black-box Membership Inference Attacks via Adversarial Examples. In CCS 2019.

---

### Note · Authors · 2024-12-03

**Comment:**

Dear ICLR Program Committee,

We are writing to formally withdraw our submission from the current review process after careful consideration of the professional suitability and engagement of the reviewers, which we feel did not align with the expectations for a productive peer review process.

Our submission introduces a novel membership inference framework that leverages a deep kernel-based non-parametric two-sample test and demonstrates its efficacy through rigorous experimentation, comparing it with existing methods. Our work provides a fresh perspective on evaluating privacy risks in machine learning models, achieving state-of-the-art (SOTA) results and opening avenues for further research in this critical area.

We believe the significance of our contribution has been underestimated, and we outline the two primary reasons for our decision to withdraw:

- **1.Lack of Expert Understanding and Insufficient Engagement**

   The reviewer’s confidence scores (all 3 or 2) and the concerns/questions raised suggest a significant mismatch between the reviewers' expertise and the specific domain of our submission. The mismatch in expertise prevented the reviewers from having enough confidence to support our work, despite the fact that we had thoroughly addressed their concerns. Major issues raised were based on misunderstandings that could have been easily addressed with a more thorough reading of our manuscript. During the rebuttal phase, we provided detailed clarifications addressing these concerns (please refer to our general response and the responses to each reviewer). Despite our efforts, no further feedback or engagement was received from the reviewers.

- **2. Lack of Response After Comprehensive Rebuttal**

  In response to the reviewers' misunderstandings and questions, we made a concerted effort to provide detailed clarifications during the rebuttal phase and updated the manuscript with the additional experiments requested. However, despite these comprehensive updates, the reviewers did not engage further or provide responses during the OpenReview discussion period. This lack of dialogue undermines the very purpose of the peer review process.

While we appreciate the time of all reviewers and the positive feedback from Reviewer QQwc, we have reservations regarding the professional suitability and engagement demonstrated throughout the review process.

Thank you for your understanding.

**Withdrawal Confirmation:**

I have read and agree with the venue's withdrawal policy on behalf of myself and my co-authors.